# The evaluation of the potential of global data products for snow hydrological modelling in ungauged high alpine catchments

Michael Weber[1], Franziska Koch[2], Matthias Bernhardt[2], and Karsten Schulz[2]

[1]Department of Geography, Ludwig-Maximilians-Universität, Munich, Germany
[2]Institute for Hydrology and Water Management, University of Natural Resources and Life Sciences, Vienna, Austria

**Correspondence:** Franziska Koch (franziska.koch@boku.ac.at)

**Abstract.** For many ungauged mountain regions, global datasets of different meteorological and land surface parameters are the only data sources available. However, their applicability in modelling high alpine regions has been insufficiently investigated so far. Therefore, we tested a suite of globally available datasets by applying the physically-based Cold Regions Hydrological Model (CRHM) for a 10-year (Sept 2000 – Aug 2010) period in the gauged high alpine Research Catchment Zugspitze (RCZ),

which is 12 km$^2$ and located in the European Alps. Besides meteorological data, snow depth is measured at two stations. We ran CRHM with a reference run with in situ measured meteorological data and a 2.5 m high-resolution digital elevation model (DEM) for the parameterization of the surface characteristics. Regarding different meteorological setups, we used ten different globally available datasets (including versions of ERA, GLDAS, CFSR, CHIRPS) and additionally one transferred dataset from a similar station in the vicinity. Regarding the different DEMs, we used ALOS and SRTM (both 30 m) as well as GTOPO30 (1 km). The following two main goals were investigated: a) the reliability of simulations of snow depth, specific snow hydrological parameters and runoff with global meteorological products and b) the influence of different global DEMs on snow hydrological simulations in such a topographically complex terrain. The range between all setups in mean decadal temperature is high at 3.5 °C and for the mean decadal precipitation sum at 1510 mm, which subsequently leads to large offsets in the snow hydrological results. Only three meteorological setups, the reference, the transferred in situ dataset and the CHIRPS dataset, substituting precipitation only, showed agreeable results when comparing modelled to measured snow depth. Nevertheless, those setups showed obvious differences in the catchment's runoff regime and in snow depth, snow cover, ablation period, the date and quantity of maximum snow water equivalent in the entire catchment and in specific parts. All other globally available meteorological datasets performed worse. In contrast, all globally available DEM setups reproduced snow depth, the snow hydrological parameters and runoff quite well. Differences occurred mainly due to differences in radiation model input due to different spatial realizations. Even though SRTM and ALOS have the same spatial resolution, they showed considerable differences due to their different product origin. Despite the fact that the very coarse GTOPO30 DEM performed relatively well on the catchment mean, we advise against using this product in such heterogeneous high alpine terrain since small-scale topographic characteristics cannot be captured. While global meteorological data is not suitable for sound snow hydrological modelling in the RCZ, the choice of the DEM with resolutions in decameter-level is less critical. Nevertheless, global meteorological data can be a valuable source to substitute single missing variables. For the future, however, we expect

an increasing role of global data in modelling ungauged high alpine basins due to further product improvements, spatial refinements and further steps regarding assimilation with remote sensing data.

## 1 Introduction

Worldwide, the vast majority of catchments are ungauged, which means that within these basins continuous data of in situ measured meteorological and snow hydrological parameters are not available (Abimbola et al., 2017; Blöschl et al., 2013). This is critical, as many ungauged basins need detailed information on their hydrological cycle and storages to optimize water usage schemes or hazard management (Merwade et al., 2008; Hirsch and Costa, 2004; Abimbola et al., 2017). However, the best available data for hydrological modelling are in situ hydro-meteorological station recordings like e.g. temperature, precipitation or runoff and data from snow gauges, whereof the latter is particularly important in high alpine areas (Fekete et al., 2015). A broad range of mountain catchments, which contribute snow melt as major runoff component to the rivers, are affected by the lack of sufficient instrumentation and data (Hrachowitz et al., 2013; Wortmann et al., 2018; Zhang et al., 2015; Dussaillant et al., 2012). This is problematic, as mountain catchments are highly relevant for local water supply and that of the adjacent lowlands (Bandyopadhyay et al., 1997; Viviroli et al., 2011; Meybeck et al., 2001; Wesemann et al., 2018; Mauser and Prasch, 2015; Zhang et al., 2015; Koch et al., 2011; Huggel et al., 2015).

The seasonal snow cover in mountains plays a major role for the alpine water cycle (Barnett et al., 2005; Huss et al., 2017; Brown and Mote, 2008). For example, Beniston (2012) shows for the Rhine and the Rhone basins that snow is the biggest single contributor to seasonal runoff. Worldwide, Barnett et al. (2005) estimate that one-sixth of the population lives within snowmelt-dominated catchments and in Asia or North America, the share of meltwater from snow and ice at total runoff is estimated to be more than 30% (Huss et al., 2017). Nonetheless, the instrumentation of high alpine catchments for meteorological, snow or runoff measurements is sparse. Possible reasons are the complex topography and the remoteness of mountain regions, which makes access and logistics more difficult, labor-intensive and expensive. Further issues are the lack of funds for installation and maintenance as well as too few or untrained personnel (Sabatini, 2017; Whitfield et al., 2013; Tauro et al., 2018). In particular, snow water equivalent (SWE) measurements, representing the amount of stored water are very valuable for hydrological applications. However, SWE measurements are largely underrepresented in many regions worldwide compared to snow depth measurements (Haberkorn, 2019). Nevertheless, in situ SWE measurements are increasingly facilitated by continuous improvements of existing methods like snow pillows or scales and developments of new methods like snow sensing with Global Navigation Satellite System (GNSS) signals and cosmic rays (Koch et al., 2019; Schattan et al., 2017).

Satellite-based remote sensing serves as a valuable option or additive to derive information on the snow cover and its properties. This is particularly true for snow cover extent products derived with optical sensors despite data gaps due to cloud coverage as their major drawback (Hall et al., 2002; Gascoin et al., 2019). Contrarily, the derivation of snow depth or SWE, based on satellite remote sensing products, is still not sufficiently solved, in particular for complex terrain in high alpine environments (Lettenmaier et al., 2015; Bormann et al., 2018). Active microwave products might be severely affected by relief displacements caused by layover and foreshortening and passive microwave products are too low in their spatial resolution for alpine regions

(Vuyovich et al., 2014; Shi and Dozier, 1997, 2000). However, currently the derivation of snow properties via space borne remote sensing is rapidly developing, e.g., by launching new sensors, enhancing the spatial and temporal resolution and the development of new algorithms. Those methods already show promising results in mountainous regions (Rott et al., 2014; Notarnicola, 2020; Lievens et al., 2019). Moreover, numerous promising airborne or combined in situ, airborne and space borne approaches have been developed in the last decades (Currier et al., 2019; Adams et al., 2018; Kim et al., 2018; Bühler et al., 2015), but are still not possible to perform in many mountain areas due to their remoteness and cost issues. Therefore, until now, the snow cover cannot be entirely described by solely relying on remote sensing techniques.

The sparse data availability was one of the reasons for the International Association of Hydrological Sciences (IAHS) to launch the Prediction in Ungauged Basins (PUB) initiative in 2003 (Pomeroy et al., 2013; Blöschl et al., 2013). Research foci have been various regionalization techniques such as in situ data transferred from other catchments (Liu et al., 2013), using gridded data sets based on remote sensing or atmospheric models as well as using hybrid data that combine in situ measurements, remote sensing and atmospheric modelling (Kahl et al., 2013; Pomeroy et al., 2013). Regarding the latter, two main representatives of hybrid data are reanalysis and land data assimilation products (Poli et al., 2016; Rodell et al., 2004). Both represent a consistent, multi-year description of the atmospheric state. Precipitation data is especially afflicted by errors and needs to be bias corrected, which requires additional effort (Piani et al., 2010; Smiatek et al., 2009; Christensen et al., 2008; Muerth et al., 2013) and is difficult to achieve without field measurements (Fatichi et al., 2016; Gampe et al., 2019).

However, globally available atmospheric model, hybrid or remote sensing products are still often the only available meteorological model input data source for ungauged catchments. Moreover, publicly available data is a great opportunity for hydrologists with little budget available for data acquisition. This applies to countries with already limited financial resources (Feki et al., 2017; van de Giesen et al., 2014) and affects a large number of remote mountain catchments. The same is true for land surface information such as altitude, aspect or slope which is needed for model parameterization. This information influences a wide range of model routines like the calculation of radiation or the extrapolation of station recorded temperature to other parts of the catchment. With the generation of digital elevation models (DEMs) from remote sensing products, this information became globally available since the late 20th century (Farr et al., 2007; USGS, 1996). The largest differences amongst satellite derived DEM products occur due to different viewing angles of the applied sensors and the spatial resolutions of the product. Individual high resolution topographic information acquired e.g. by LIDAR is very rare.

Given these limitations, we tested the performance of global data products for snow hydrological modelling. Two explicit research questions were guiding our analysis:

- Is it possible to use globally available meteorological data products to reliably simulate snow depth, specific snow hydrological parameters and runoff in high alpine catchments?

- What impact have DEM products with different spatial resolutions on snow hydrological simulations in complex topography?

To answer the first question, we investigated globally and publicly available meteorological driver data from the Climate Forecast System Reanalysis (CFSR), different versions of the Global Land Data Assimilation System (GLDAS), ERA-20C as

well as ERA5 and ERA5-Land datasets and precipitation information from the Climate Hazards group Infrared Precipitation with Stations (CHIRPS). Those data are referred to as hybrid data. There have already been studies in several regions to test such data products (Förster et al., 2016; Fuka et al., 2013; Essou et al., 2016, 2017; Casson et al., 2018), however, as far as we know not solely and explicitly on high-alpine catchments. Some of these studies showed good results, whilst others underlined the importance of adapting the data to regional conditions. In addition, we evaluate the performance of a transfer from in situ measured data at a similar site as suggested by the PUB initiative (Liu et al., 2013). Regarding our second question, we investigated possible differences in model performance resulting from different topographic characteristics originating from globally and publicly available DEMs with different spatial resolutions and sensing techniques. Therefore, we chose three commonly used DEMs GTOPO30, ALOS and SRTM for this study.

We conducted our study in the gauged, well-known and snow dominated high alpine Research Catchment Zugspitze (RCZ) in the European Northern Calcareous Alps (Hürkamp et al., 2019; Härer et al., 2018; Bernhardt et al., 2018) which is one of the best instrumented high alpine catchments worldwide. This allows for an evaluation with in situ measured meteorological and snow depth data as well as in situ based snow hydrologic reference simulations. To simulate the hydrological cycle, we use the physically-based Cold Regions Hydrological Model (CRHM) (Pomeroy et al., 2007) which we already tested intensively with very good results in this area (Weber et al., 2016, 2020). Moreover, CRHM was successfully applied in other snow dominated catchments (Dornes et al., 2008; Ellis et al., 2010; Essery et al., 2009; Fang et al., 2013; López-Moreno et al., 2013). We conducted all CRHM model runs for a 10-year period in the past (September 2000 – August 2010), where all input data are available in parallel. The focus of the model output analysis is on the snow cover and runoff. The snow cover simulations are compared to measured snow depth data. We first set up a reference run with in situ measured meteorological data and a 2.5 m high resolution DEM. Then, we tested eleven setups with different meteorological driver data and three setups with different DEMs and compared them to the reference run which allowed us to assess differences in quantity and temporal development as well as the overall applicability of the simulated snow hydrological situation regarding the different globally available products.

## 2   Methods

In the following, we first describe the study site and the model structure as well as the preparation of the forcing data sets and the used DEMs. Then we present a comparison of the forcing data sets and the influence of the DEMs on the characteristics of the HRU parameters.

### 2.1   The Research Catchment Zugspitze

The high alpine RCZ (center coordinates: UTM 5250416 N 653692 E, Fig. 1) is located in the Zugspitze massif in the European Northern Calcareous Alps and covers an area of 12 km$^2$. For the peak of Mount Zugspitze (2962 m a.s.l.), Germany's highest mountain, we calculated from the data recorded by the German Weather Service (DWD) an annual mean temperature of -4.5 °C (1981-2010) and an annual precipitation sum of 2080 mm a$^{-1}$ (1981-2010). Due to the geological situation and glacial processes during the Pleistocene and the Little Ice Age (1550-1850), the terrain of the RCZ is characterized by both

**Table 1.** Automatic weather stations and measured parameters (Ta = air temperature, ppt = precipitation, rh = relative humidity, u = wind-speed, Qsi = shortwave incoming radiation, Qli = longwave incoming radiation, SD = snow depth).

| station name | altitude (m a.s.l.) | measured parameters | data available since |
|---|---|---|---|
| DWD | 2964, SD at 2600 | Ta, ppt, rh, u, Qsi, Qli, sunshine duration, SD | 1900 (radiation data 2009) |
| LWD | 2420 | Ta, ppt, rh, u, Qsi, SD | 1998 |

karst and glacial features, forming a rugged surface (Hüttl, 1999; Hirtlreiter, 1992). Furthermore, the glaciers Nördlicher und Südlicher Schneeferner are located in the catchment, covering a total area of 0.24 km$^2$ in 2015 (Hagg, 2020). As the catchment's altitudinal relief is in total approx. 1600 m, and the surface's heterogeneity in slope angle, aspect and curvature is highly variable, the meteorological and topographic conditions are changing over short distances. The runoff is gauged at 1365 m a.s.l. at the Partnach Spring and shows a clear peak in spring and summer runoff, which is characteristic for the

snow-dominated runoff regime. The catchment's upper part (above 2000 m a.s.l.) is free of vegetation, except for sparse alpine meadows and pioneer plants, while areas below 2000 m a.s.l. are characterized by krummholz (Friedmann and Korch, 2010). Despite the predominant permeable karst, tracer studies have shown that the RCZ exclusively drains to the Partnach Spring (Wrobel, 1980; Rappl et al., 2010). This, in combination with the easy accessibility via ropeways and the excellent scientific infrastructure, provided by the Environmental Research Station Schneefernerhaus (UFS), makes the RCZ ideal for

(snow-) hydrological research (Härer et al., 2013, 2018; Hürkamp et al., 2019; Morche and Schmidt, 2012; Weber et al., 2020). Within the catchment, we had access to data from two automatic weather stations (AWS) providing meteorological and snow recordings for at least twenty years (Table 1). Both stations cover all meteorological parameters (Table 1) in an hourly resolution. DWD automatically records the meteorological parameters close to the summit of Mt. Zugspitze, whereas the DWD snow depth is measured daily with a snow stake at 2600 m a.s.l. on the Nördlicher Schneeferner. The second station is operated

by the Bavarian Avalanche Warning Service (LWD) and is located lower at an altitude of 2420 m a.s.l.. We use measured SWE at the LWD station, which were available in recent years to determine an under catch of 50% of snow precipitation (Weber et al., 2020). This is well within the range of snow precipitation underestimation reported in literature (WMO, 2011; Grossi et al., 2017). Therefore, we used the factor of 1.5 to correct snow precipitation measured at the DWD station which is used for model forcing. The factor was determined using 10-minute values to minimize the effect of sublimation from the snow cover.

However, the potential effect of snow redistribution by wind on this factor could not be detected by analysing the SWE in periods with temperatures below the freezing point, strong winds and without snowfall. Besides, it has to be noted that strong winds rarely occur without precipitation in RCZ, pointing to some limitations of this under catch factor determination. Table 1 gives an overview of the RCZ including the locations of the in situ stations and the measured parameters. For more detailed information on the physiography of the RCZ, it is referred to Weber et al. (2016), Friedmann and Korch (2010), Rappl et al.

(2010) and Wetzel (2004).

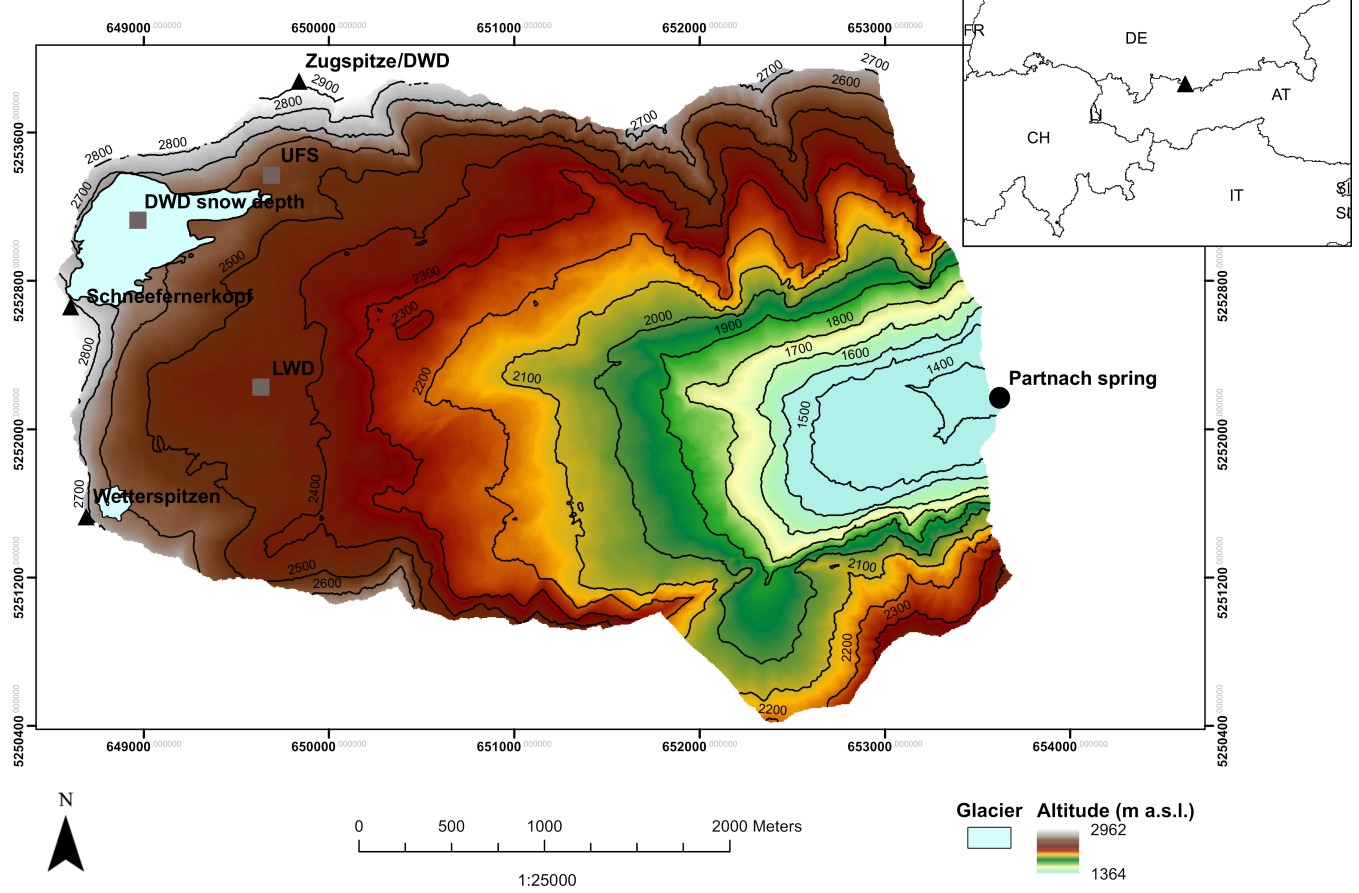

**Figure 1.** Overview of the Research Catchment Zugspitze (RCZ) and location of in situ stations. DWD = German Weather Service, LWD = Bavarian Avalanche Warning Service, UFS = Environmental Research Station Schneefernerhaus.

## 2.2 Structure of the Cold Regions Hydrological Model

For simulating the water cycle of the RCZ we used CRHM, a physically-based modelling platform which uses hydrological response units (HRUs) for spatial discretization and which was explicitly designed to simulate (snow-)hydrological processes in small-to-medium sized catchments in cold arctic and alpine environments (Pomeroy et al., 2007). Following the approach proposed by Weber et al. (2020), which is briefly explained in the following, the HRU scheme for RCZ encompasses the same HRUs also for this study (Fig. 2). Weber et al. (2020) derived HRUs via a cluster analysis of the catchment's physiographic properties aspect, slope angle, altitude, vegetation, and a wind sheltering index developed by Winstral et al. (2002). The general idea behind the concept is that spatial units that are assumed to behave hydrologically in the same way due to the natural features are assigned to one HRU. One HRU does not necessarily have to be one single area, but can contain several areas with similar topographic attributes (Flügel, 1995). To define the appropriate number of clusters, i.e. the appropriate number

of HRUs, the cluster analysis needs a reference for evaluation. Since the snow cover is the dominant component of the water cycle of the RCZ, Weber et al. (2020) chose the dominant pattern in the snow cover as reference. This was carried out under the assumptions that snow depth and its distribution are the dominant components in the water cycle of RCZ and that the snow depth distribution is a result of the interaction between meteorology and land surface (Dadic et al., 2010; Mott et al.,
2014). To identify the dominant snow cover pattern, a time series of snow depth maps, which was derived with terrestrial laser scanning, was investigated via principal component analysis (PCA). In the following, the clustering that best reflects this PCA derived pattern was chosen as the HRU scheme. For more information on this HRU derivation method, we refer to Weber et al. (2020). The advantage of this approach is that the HRUs are derived objectively and in a sufficient quantity to guarantee optimal representation of the controlling processes while at the same time ensuring computational efficiency. It turned out
that ten HRUs are optimal to represent the catchment's snow cover distribution. The areal extension, land cover as well as the main characteristics of the topographic parameters altitude, slope and aspect of these 10 HRUs are presented in Fig. 2 and for different DEM products in Table 2. The topographic characteristics for the reference setup are those of the 2.5 m resolution reference DEM setup.

Regarding the meteorological input, CRHM requires air temperature, relative humidity, precipitation, wind speed, as well as
shortwave and longwave incoming radiation for each HRU. For the reference setup, these data are provided by the DWD station. CRHM has a modular structure, which allows a model setup according to the available input data and the alignment of the research question. The model structure of CRHM is briefly depicted in Weber et al. (2016) and extensively described in Pomeroy et al. (2007). Snow melt processes are described by SnobalCRHM (Marks et al., 1999; MacDonald et al., 2010), an energy balance model specifically designed for deep alpine snow covers. Aspect and slope angle influence the radiation
budget and thus melt processes in the catchment. Therefore, radiation parameters, like direct and diffuse radiation as well as a slope correction are calculated in CRHM with a method created by Garnier and Ohmura (1970). This method uses the ratio of measured shortwave radiation (Qsi) and the calculated clear sky direct and diffuse Qsi on a horizontal plane to adjust the calculated clear sky value on the HRU specific slope. The net radiation of the whole spectrum is calculated with an approach developed by Brunt (1932). For HRU specific albedo calculations, CRHM uses a procedure developed by Essery and Etchevers
(2004) and Pomeroy and Li (2000) created the method to calculate snow transport. Regarding evapotranspiration, CRHM uses an approach developed by Granger and Pomeroy (1997). Interception and sublimation are modelled with a procedure developed by Ellis et al. (2010). All these calculations require information on catchment topography, which is characterized by the HRUs (Table 2).

In addition to the necessary information on topography, CRHM needs information on the land cover. In general, the vegetation
characteristics of the RCZ are rather simple from a snow hydrological point of view. Only the lower parts are covered with krummholz. Therefore, we do not apply specific remote sensing data or products to derive further vegetation and land cover data but stick to the good description of the land cover of the RCZ by Friedmann and Korch (2010). The extent of the glaciated areas within RCZ is derived from Hagg (2020). Unlike most hydrological models, CRHM does not require calibration. The only parameters that we adjusted for modelling are the previously named catchment/HRU specific physiographic characteristics.

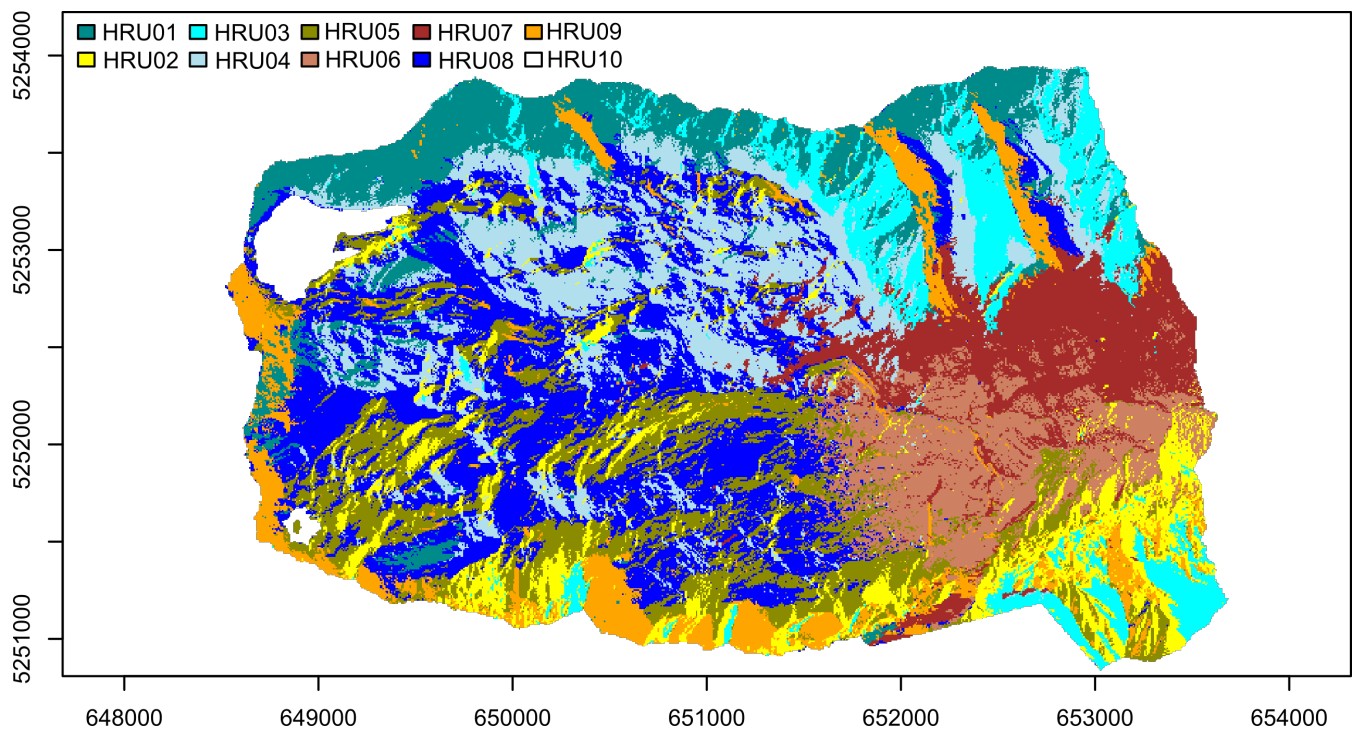

**Figure 2.** Hydrological response unit (HRU) discretization of the Research Catchment Zugspitze (Weber et al., 2020).

## 2.3 Meteorological input data and their preparation for model input

As meteorological input for the reference simulation, we use only the data measured at the DWD station on Mt. Zugspitze which represents the 'gauged basin mode'. This data is the best in terms of continuity and data quality directly measured in RCZ. With these data, we already successfully conducted snow hydrological modelling, as shown in Weber et al. (2020). Measured snow depth could be modelled with an accuracy of NSE > 0.7 (Nash-Sutcliffe-Efficiency, (Nash and Sutcliffe, 1970)). Due to some huge gaps in the LWD data during summer time as well as in spring, we refrained from using it as model input. We pre-processed the station measurements at DWD according to Weber et al. (2016). This includes the detection and removal of measurement errors and the interpolation of data gaps according to a procedure proposed by Liston and Elder (2006). The corrected data were then transferred to the HRUs following the method of Liston and Elder (2006) as well. This method uses monthly variable lapse rates for temperature (Kunkel, 1989) and a monthly variable precipitation adjustment factor (Thornton et al., 1997) for the altitude-dependent adjustment of temperature and precipitation. Relative humidity was adjusted via dew point temperature which has a relatively linear dependence on elevation (Liston and Elder, 2006). The same method is also applied for the non-reference meteorological and DEM input data, which is presented in the following.

To investigate the potential performance of non in situ measured data in the RCZ, we set up the CRHM simulations also for

an 'ungauged basin mode', in which we assume to have no measured model forcing data, with a selection of in total eleven different meteorological setups using the following data categories:

- hybrid data, which are a combination of satellite products, ground based data and atmospheric models, for all meteorological driver data

- hybrid data using a site-specific downscaling with temperature correction

- hybrid data for the supplementation of precipitation data only

- data transferred from a similar catchment

In order to guarantee for comparability, we chose data that overlaps for at least ten years which is between September 2000 and August 2010. Moreover, the same HRU delineation as for the reference was applied. For our study, we came up with the data sets listed in Table 3.

Regarding the hybrid data, we forced CRHM with the well-known Climate Forecast System Reanalysis (CFSR) dataset provided by the National Centers for Environmental Prediction (NCEP) (Saha et al., 2010). Moreover, we used NASA's Global Land Data Assimilation Systems (GLDAS) (Rodell et al., 2004), ERA-20C (Poli et al., 2016) and the new ERA5 (Hersbach et al., 2020) and ERA5-Land (Muñoz Sabater, 2019). CFSR, GLDAS, ERA-20C, ERA5 and ERA5-Land products combine in situ measurements, remote sensing and atmospheric modelling. ERA5 and ERA5-Land are the latest members of the ERA family and ERA5-Land has a relatively high spatial resolution with 9 km. We use four different versions of GLDAS to test potential differences. The differences between the GLDAS versions occur in the atmospheric models and the forcing data. Forcing data of GLDAS1 and GLDAS 2.1 simulations are a combination of NCEP's Global Data Assimilation System (GDAS), disaggregated CMAP (CPC Merged Analysis of Precipitation) precipitation (Arkin et al., 2018), and Air Force Weather Agency radiation data sets (Eylander et al.). GLDAS2, is forced with the Princeton Global Meteorological Forcing Dataset (Sheffield et al., 2006) and GLDAS2_repro with an updated version of it. Simulations of GLDAS1 are conducted with the Noah 2.7.1 model (Niu et al., 2011), whereas the other GLDAS simulations are conducted with the Noah 3.3 model. The reason for choosing the relatively coarse (125 km) ERA-20C data is that we wanted to examine the effect of the temperature correction approach by Gao et al. (2012) which was developed with ERA-20C data and in situ data from RCZ (ERA-20C_corr).

Regarding the category of hybrid products providing precipitation data only, we chose data from the Climate Hazards group Infrared Precipitation with Stations (CHIRPS) (Funk et al., 2015; Love et al.), which provides daily precipitation at a spatial resolution of 0.05°. The idea of choosing CHIRPS data was to test the model performance in case only precipitation needs to be substituted. For the remaining meteorological driver data, we used the measured meteorological reference data. The motivation behind this specific setup was that precipitation gauges are costly to maintain and precipitation measurements in mountain environments are often highly error-prone (WMO, 2011; Grossi et al., 2017; Weber et al., 2020; Germann and Joss, 2004; Hrachowitz and Weiler, 2011), whilst the other input parameters are easier to acquire and more reliable, especially in winter. Thus, there is a frequent interest and need to compensate the lack of precipitation data in high alpine areas.

In addition to the globally and publicly available data, we transferred data from another alpine DWD station situated on the

top of Mt. Wendelstein (1832 m a.s.l.) (DWD_Wendelstein) to our study catchment. This station is located approximately 84 km to the northeast of RCZ in the same alpine climate surrounding at the northern rim of the Alps. As proposed by Liu et al. (2013) in the PUB initiative, we borrowed the recorded data for all meteorological parameters and transferred it to the RCZ. This station had the best available data set, as we had no access to other alpine data sets in the vicinity at an altitude comparable to the meteorological station at Mt. Zugspitze and for the investigated time period. As no longwave radiation was recorded at Mt. Wendelstein, we used the available cloud cover and temperature data to calculate it following a combination of the approaches from Konzelmann et al. (1994) for clear sky conditions and König-Langlo and Augstein (1994) and Sedlar and Hock (2009) for overcast conditions. Prior to this application, we tested the procedure with measured data from Mt. Zugspitze and could achieve good accordance with a NSE of 0.81. Table 3 provides an overview of all meteorological input data sets used for this study. Additionally, the mean altitude of the datasets is shown – an information that is needed for the transfer of each meteorological parameter to each HRU according to the method by Liston and Elder (2006), which has been described in Section 3.2. Regarding all the globally and publicly available data providing the meteorological input, the spatial resolution is coarser than the areal extent of the whole RCZ, which is therefore represented by only one pixel (Table 3). Thus, this data can be considered as a point measurement at the given altitude. The CHIRPS data is slightly finer in its spatial resolution with four pixels covering the entire RCZ. However, the resolution is still too coarse for a precise attribution to the individual HRUs. Therefore, we took the mean of the four pixels as the altitude differences between the four pixels are negligible and treated it like a point measurement.

## 2.4    Applied DEMs to describe the land surface parameters

Besides the meteorological input data, land surface information, such as altitude, slope and aspect, are needed to parameterize each HRU (see Section 2.2). This HRU specific information is needed by CRHM for a number of routines which calculate albedo, snow transport, the share of snow in precipitation or sublimation (see also Section 2.2). Due to remoteness and high costs, an individual high-resolution DEM that we used it in our reference setup is not available for the majority of high alpine catchments in the world. Thus, we aim to examine how well land surface information, derived from globally and publicly available DEMs, performs for snow hydrological CRHM simulations in the RCZ. For this purpose, we chose the ALOS (Advanced Land Observing Satellite) DEM (Tadono et al., 2014) and the SRTM (Shuttle Radar Topography Mission) DEM (Farr et al., 2007), both with a spatial resolution of 30 m. This makes it interesting to investigate if there are big differences despite the same spatial resolution. Moreover, we chose the GTOPO30 DEM with a spatial resolution of 30 arc seconds (approx. 600 x 900 m in RCZ) (USGS, 1996; Gesch et al., 1999). The SRTM and the GTOPO30 DEMs are especially well-known and have been frequently used for hydrologic modelling (Ludwig and Schneider, 2006; Nagaveni et al., 2019; Seyler et al., 2009). In 2010, the Global Multi-resolution Terrain Elevation Data 2010 (GMTED2010) was published to replace GTOPO30. GMTED2010 consists of SRTM data, if available, and of other DEMs with variable resolution outside the SRTM coverage (Danielson and Gesch, 2011). It is obvious that in more recent years, the DEM products with higher spatial accuracy increased in their usage compared to older and less accurate product like GTOPO30. Nevertheless, we want to examine their performance in snow hydrological modelling since it used to be a standard product for hydrological applications for many

years. An additional globally and publicly available DEM with a 30 m resolution that was updated in 2019 would be the ASTER DEM (Abrams and Crippen, 2019).

## 3 Results

We first compared the meteorological input data as well as the influence of different global DEMs on meteorological conditions. In a second step we simulated snow depth for all setups and performed further snow hydrological investigations with the most plausible setups. The results in comparison to measured in situ data and the reference simulation are presented in the following.

### 3.1 Comparison of the different meteorological input data sets

We compared the air temperature and precipitation of all setups with the in situ measured meteorological reference data set. Although we consider the measured data to be the reference, it is not sure at this point of the paper that it performs best in modelling the snow cover. Therefore, the presented results are relative. Table 4 provides the 10-year mean values of the individual data sets for the RCZ. The differences in the mean decadal annual air temperature between the warmest and coldest meteorological setup is 3.5 °C. The mean annual air temperature of the reference is 0.9 °C, which is warmer than all others. As the CHIRPS setup only supplements precipitation, temperature is identical to the reference. Reasons for the pronounced higher temperatures of the measured reference could be inversion or Föhn conditions which are not represented properly in global products. With -2.6 °C, the coldest data set regarding the mean annual temperature is represented by ERA-20C. In contrast, the temperature of the ERA-20C_corr setup is closer to the reference. The newer and higher resolution ERA5 and ERA5-Land setups are in a similar range with -1.8 °C and -2.5 °C. The mean annual air temperature of all other driver data is in-between those rather cold ERA setups and the reference. Interestingly, the differences among the GLDAS versions are quite high. As Fig. 3a shows, the differences between all data sets are almost constant over the years. In addition, Fig. 3 presents a climatology information with the 30-year (1991 - 2020) median, the interquartile range (IQR), expressing a quantile range of 25% to 75%, and the total range of the reference data. For the annual mean temperature, it is obvious that no dataset except some years of DWD_Wendelstein as well as the GLDAS2 and GLDAS2_repro is within the range that was measured at DWD during the last 30 years.

According to the reference data set, the catchment receives a mean annual precipitation sum of 1830 mm in the study period. The range in the mean decadal annual precipitation sum between the wettest and driest meteorological setup is very high with 1510 mm (Table 4). The smallest values are 1651 (GLDAS1) and 1660 mm a$^{-1}$ (GLDAS2.1), being the driest setups, and the maximum value is 3161 mm a$^{-1}$ (CFSR), corresponding to 70% higher mean annual precipitation than in the reference. All other data sets indicate wetter conditions than the reference. In contrast, the differences in annual precipitation show no constant offset (Fig. 3b). Although in the same climatic region, the precipitation measured at the DWD station Wendelstein shows a contrary development to the reference data between 2005 and 2010 which might be due to small scale local effects. Only precipitation data from ERA5 and CFSR are close to the reference data over the last 30 years. An overview on the decadal mean monthly precipitation sums provides information on the annual precipitation regime of the individual setups (Fig. 4). In

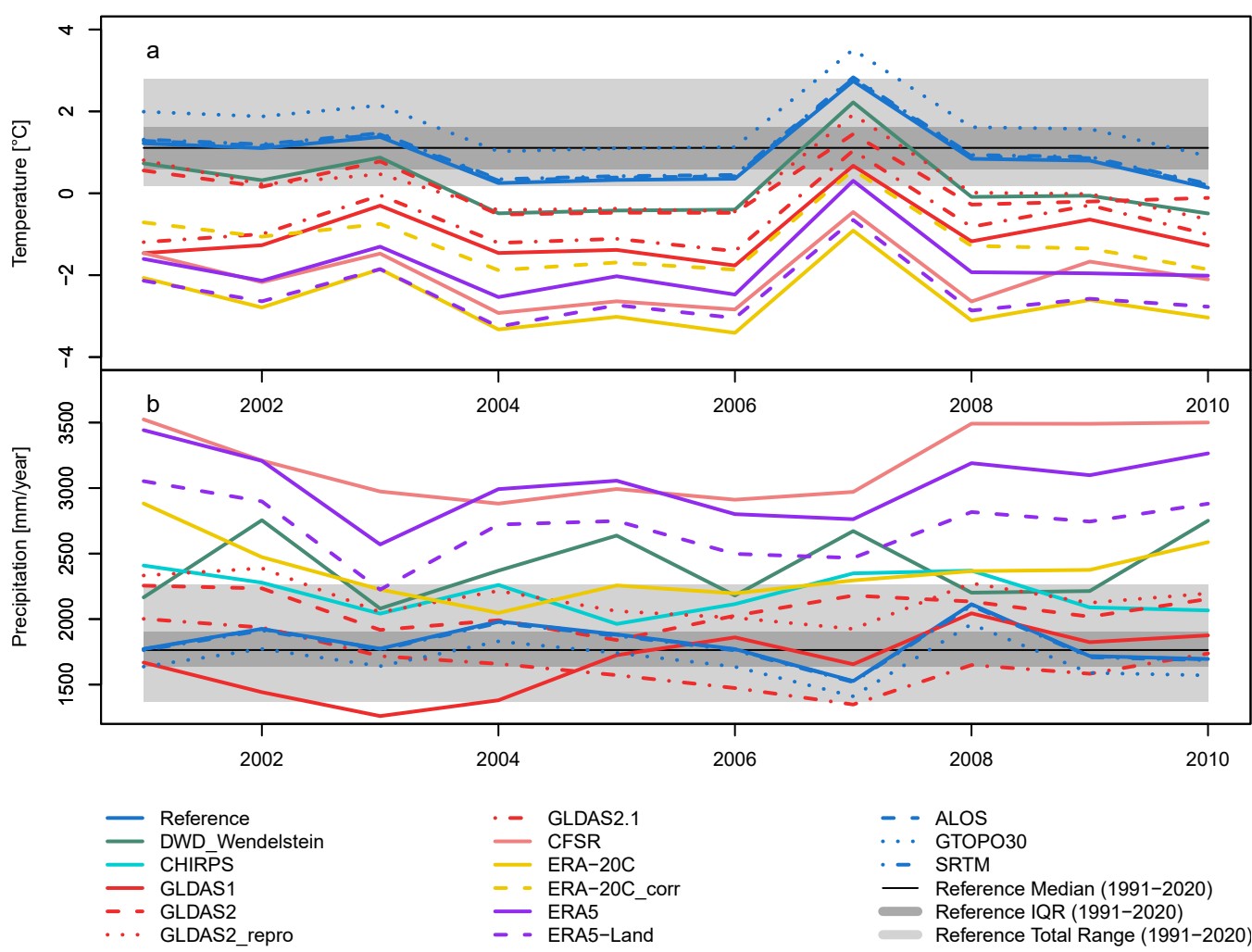

**Figure 3.** Annual Research Catchment Zugspitze mean of (a) air temperature and (b) precipitation sum of the different meteorological setups and of the different DEM setups.

general, not all precipitation regimes follow the same pattern. The reference and DWD_Wendelstein data sets follow a regime with predominant precipitation during the snow accumulation period. All other meteorological datasets reach their maximum precipitation in summer but also show a local maximum in March. The CHIRPS precipitation is similar to the reference, however, with a more pronounced maximum in August. As we focus on the seasonal snow cover, we assume that precipitation amount and regime is of major importance for snow cover simulations.

The 10-year monthly mean of incoming shortwave radiation (Qsi) for the RCZ, relevant for snow cover depletion, is presented in Fig. 6. All setups show very similar data. However, some data, e.g. the ones of ERA5, ERA5-Land and DWD_Wendelstein

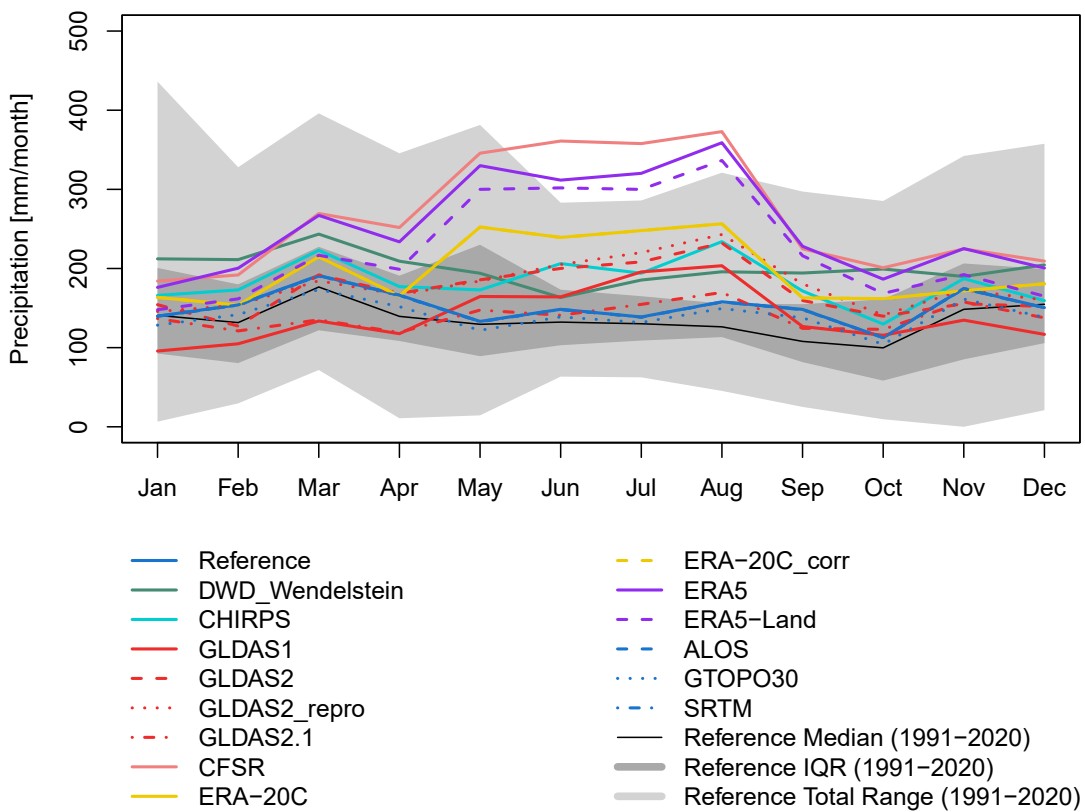

**Figure 4.** 10-year average of monthly precipitation sums of the different meteorological setups and of the different DEM setup.

show almost a plateau from spring to summer in contrast to the reference (Fig. 6a. A possible reason might be that convective clouds form during this period while Mt. Zugspitze is frequently above the condensation level.

To sum it up, the majority of the global products is not in the climatological range of the in situ data regarding the mean annual temperature and mean annual precipitation and indicates large offsets. This is also true for the interannual variability of monthly precipitation sums; however, they are for the most setups within the total range of the climatology. All ERA products and the CFSR datasets are significantly colder and wetter compared to the reference dataset. The DWD_Wendelstein, GLDAS2 and GLDAS2_repro datasets are slightly colder and wetter, and the GLDAS1 and GLDAS2.1 datasets are colder and drier than the reference. The CHIRPS dataset only differs in precipitation with the reference; however, showing a similar precipitation regime. The effects of these deviations on snow-hydrological processes will be shown in Section 3.3 and 3.4.

### 3.2 Influence of the DEMs and associated land surface parameters on meteorological conditions

The HRU elevation, slope and aspect obtained with the 30 m resolution DEMs ALOS and SRTM are similar to the ones obtained with the 2.5 m high-resolution reference DEM (Table 2). The variance of slope and aspect angles in the coarse GTOPO30 DEM setup is very low. This means that almost all HRUs are represented quite similarly from a topographical

perspective. Compared to the reference, for some HRUs parametrized with GTOPO30, the aspect is even almost 180°different. The slope angles vary up to 45°, which does not give a realistic topographic picture of the HRUs and influences the radiation budget. To demonstrate the effect of different topographic characteristics, we used the reference meteorological dataset from DWD and adjusted it to the ALOS, SRTM and GTOPO30 DEM specific HRU altitude, slope and aspect (Table 2) as explained in Section 2.3. Differences in the meteorological variables due to variations in topography are summarized in Fig. 3. The 0.9 °C higher decadal mean temperature of the GTOPO30 setup when compared to the reference is caused by the 105 m lower catchment mean altitude (Table 2). The mean catchment altitudes of the ALOS and the SRTM setups are 2233 m a.s.l. and 2258 m a.s.l, respectively, which is very close to the mean catchment altitude of the reference DEM with 2234 m a.s.l.. The catchment mean temperature deviates only 0.1 °C for the two global DEMs from the reference data set. Since the precipitation adjustment is – like the temperature adjustment – also carried out according to an altitude dependent principal (Section 2.3), the 10-year mean annual precipitation sum listed in Table 4 shows a similar picture like the temperature. The GTOPO30 setup differs considerably also regarding the mean annual precipitation sum. It is 139 mm less than the reference caused by the lower mean catchment altitude. It is worth noticing that the initial meteorological data, and therefor the precipitation regime, is the same in each DEM setup (Fig. 4).

Pronounced differences also occur for Qsi (Table 4), since this parameter is largely influenced by slope and aspect. Fig. 5 presents the 10-year mean of Qsi in each HRU. The greatest variation in radiation can be seen in the reference setup due to the high resolution of the DEM which represents even very small scale topographic differences. In contrast, the lowest variation can be seen for the coarse GTOPO30 setup. For both DEMs with 30 m spatial resolution, variations in the ALOS setup are slightly larger compared to the SRTM setup. This is because the SRTM DEM is smoother due to a specific post-processing algorithm used for this product which becomes obvious in the steep terrain (Guth, 2006). The representation of Qsi using the ALOS DEM is closer to the reference DEM. Differences between SRTM and ALOS might also occur due to different sensor viewing angles. In addition to the spatial distribution of Qsi in the catchments, Fig. 6b shows the 10-year mean annual cycle of Qsi for the RCZ and illustrates the same mean annual radiation curve of the three DEM setups with an almost constant bias.

To summarise, the resolution or processing of the DEM does not only influence aspect and slope angles, and thus Qsi, but also elevation. As a consequence, this affects the calculation of the catchment- and HRU mean temperature and precipitation values which is especially significantly different for the very coarse GTOPO30 DEM setup. Despite the ALOS and SRTM DEM having the same spatial resolution, the different smoothing algorithms which have been applied to generate these products, result in considerable differences in Qsi due to differences in aspect and slope angle.

## 3.3 Evaluation of simulated snow depth with measured data

In a next step, the different meteorological and DEM setups for HRU parameterization were used as data input for snow hydrological simulations with CRHM. To evaluate the quality of the simulated results of all setups including the reference, we compared at first, as presented in this section, the modelled snow depth on a daily basis with in situ measurements that are available in two HRUs. The DWD snow depth measurement site is located in HRU10 and is only 15 m lower than HRU

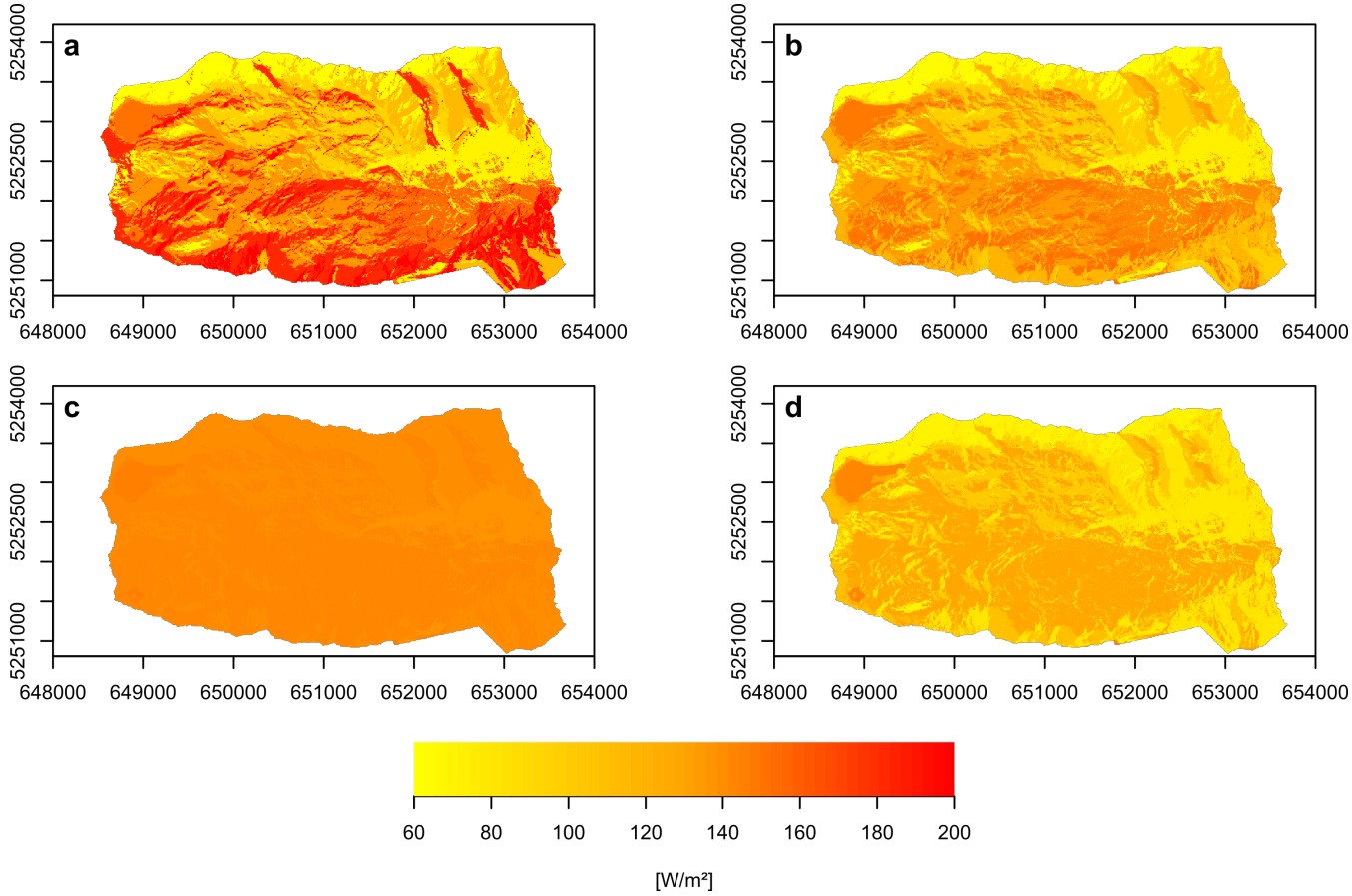

**Figure 5.** 10-year mean of short wave incoming radiation (Qsi) in the (a) reference, (b) ALOS, (c) GTOPO30 and (d) SRTM setup.

altitude. The LWD station is located in HRU4 but is situated 99 m higher than the HRU altitude. Table 5 provides the quality measures NSE, coefficient of determination ($R^2$) and the mean absolute error (MAE).

Figure 7 compares the simulated daily snow depth of all different meteorological and DEM model setups including the measured snow depth at the DWD and the LWD stations. It is obvious that the simulated snow depth with the reference model

setup fits best with the measured data for both stations (Fig. 7a, b). This is also reflected in the high statistical values regarding a NSE of 0.79, $R^2$ values of 0.81 and 0.86, and the lowest mean absolute error (MAE) as shown in Table 5. The results of the reference run and the majority of all setups are better for the LWD station than for the DWD station. One reason for this could be that the DWD snow depth is measured directly on top of the glacier. The ice has a cooling effect on the snow cover which leads to a delayed melt out of a thin snow cover compared to a faster melt of snow directly on top of rocks. This effect

is particularly strong in spring and autumn, and is not considered in the model. However, this effect is negligible in the case where larger amounts of snow are piled up on top of the glacier, as it is the case for most of the time during the winter season. Thus, modelled snow cover at the DWD station lasts on average 20 days less than the measured one.

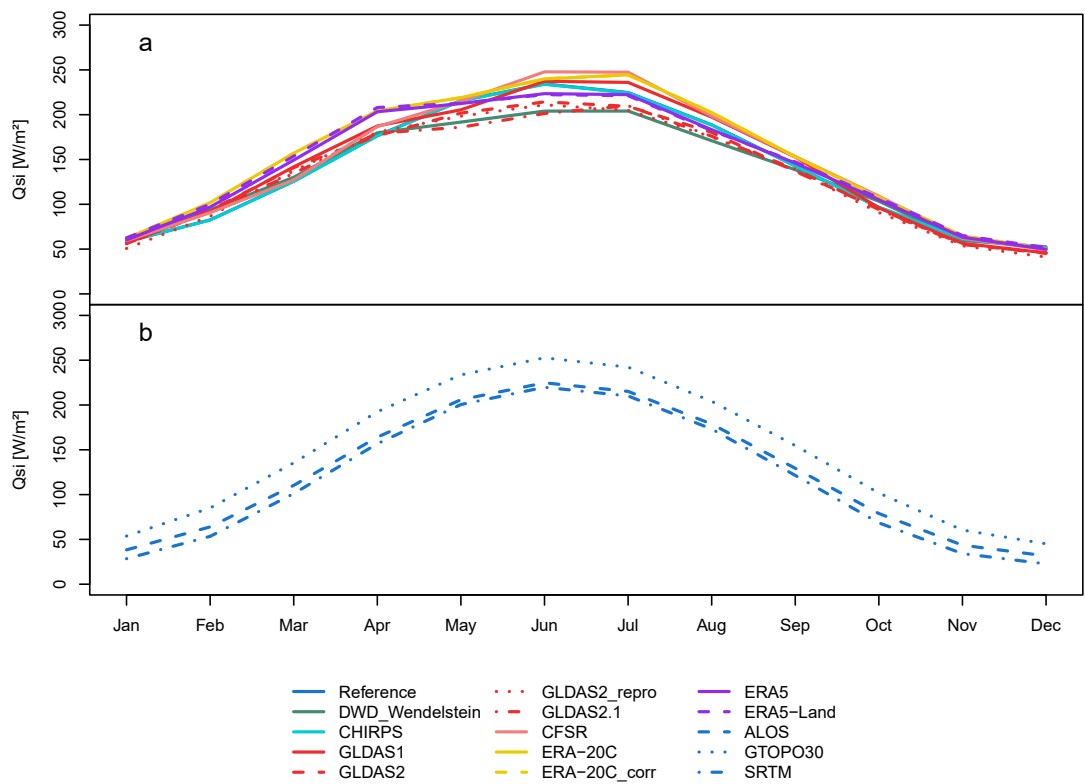

**Figure 6.** 10-year mean monthly short wave incoming radiation (Qsi) of the Research Catchment Zugspitze of (a) the different meteorological setups and of (b) the different DEM setups.

The comparison of the snow depth results simulated by the all setups with different meteorological input data, including the reference with the measurements at the two stations, can be summarized as follows: The simulations, using the transferred data from the DWD AWS at Mt. Wendelstein (Fig. 7a, b) and the CHIRPS dataset (Fig. 7c, d) both produce plausible results at each measurement station (Table 5). The NSE values of the CHIRPS setup range between 0.54 and 0.55; the $R^2$ values range between 0.66 and 0.79 (Table 5). The reason for the good performance of the CHIRPS setup is that the forcing data is the same as in the reference setup except from precipitation. The overestimation of snow depth in some years, especially at the DWD site, is due to the higher precipitation values as described in Section 3.1. For the snow depth simulations based on the transferred data from the alpine DWD AWS at Mt. Wendelstein, we also obtained reasonable but weaker results, e.g., with an $R^2$ of 0.59 and 0.66 at LWD and DWD, respectively but higher MAE values. The NSE values at the stations are 0.33 and 0.11, respectively. The statistical values are rather moderate, but still hint at a plausible representation of the temporal development compared to the results of the other global data. The low NSE values can be traced back to the years 2002, 2005 and 2010 in which the peaks are strongly overestimated in the DWD_Wendelstein run at both stations. Both, the CHIRPS and DWD_Wendelstein show temperature and precipitation data which are for most years within the total range of climatology. All four GLDAS setups (Fig. 7d) and the ERA-20_corr (Fig. 7f) setup, except for the years 2001 to 2003 that indicate too

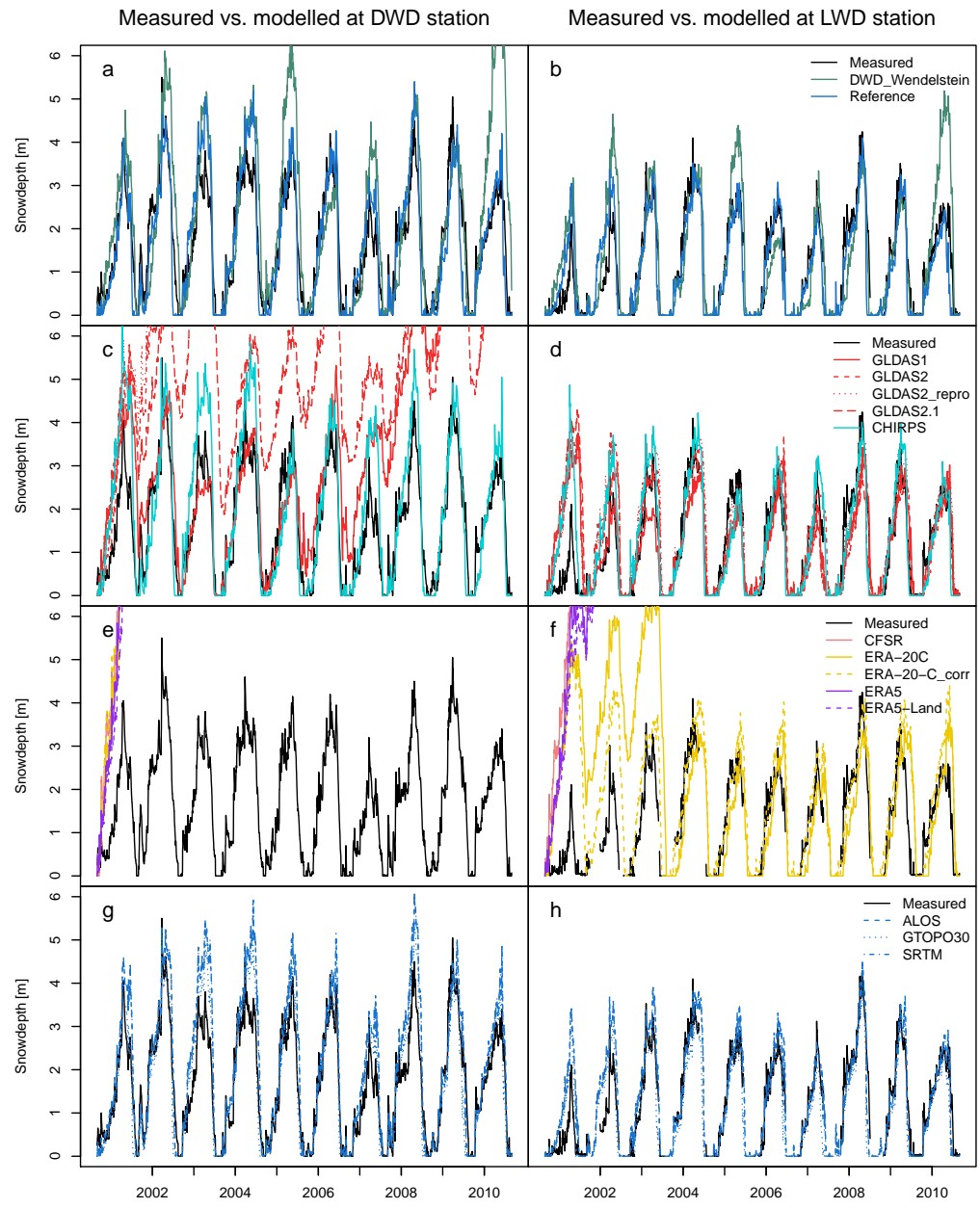

**Figure 7.** Comparison of modelled snow depth results with measured snow depth at the measurement sites of Bavarian Avalanche Warning Service (LWD) and German Weather Service (DWD). The subplots (a) to (f) are related to different meteorological input setups and subplots (g) and (h) to different hydrological response unit (HRU) parameterizations using different DEMs.

high snow depths, reflect quite well the quantity and the temporal dynamics of the snow cover development at the LWD station. However, this is not the case for the simulations at the DWD station (Fig. 7c, e), where so called 'snow towers' pile up resulting in very low $R^2$ and NSE values as well as high MAEs (Table 5). Snow towers are an effect in snow hydrological

modelling that occurs mainly at higher altitudes and describes the unrealistically high accumulation of snow over several years. Reasons can be the insufficient description of redistribution processes in the model or unrealistic meteorological forcing data (Freudiger et al., 2017) as the latter will be true in our case. These simulated snow towers are caused by much lower temperatures and much higher precipitation compared to the reference (Section 3.1). The same can be observed for the CFSR, ERA5 and ERA5-Land setups, even at both measurement stations. The relatively better performance of the ERA-20C_cor

setup compared to the ERA-20C setup originates from the local temperature downscaling which has already been tested for the high-alpine Mt. Zugspitze by Gao et al. (2012). Moreover, the previously described effect of snow towers also occurred in the two highest located HRUs for all GLDAS and ERA-20C_corr setups and also at further HRUs within the catchment for GLDAS2, GLDAS2.1 and GLDAS2_repro and the ERA-20C data sets. The location of the highest HRU can be one reason for the lower accordance of snow depth at the DWD station compared to the LWD station regarding the simulated results of

all global meteorological setups. This favours temperatures and precipitation amounts, which can lead to the development of snow towers.

Regarding the snow depth results simulated by using the model setups with different topographic characteristics on basis of the 2.5 m DEM for the reference run and the three different globally and publicly available DEMs ALOS, SRTM and GTOPO30, the snow depth development was simulated realistically at both measurement sites (Fig. 7g, h, Table 5). This

results in $R^2$ values above 0.8, NSE values above 0.77 and MAE values below 0.33 for all three global data setups at the LWD site being close to the statistical values of the reference. For GTOPO30, the quality measures for the DWD site are even slightly better, whereas the ALOS and SRTM setups show slightly weaker NSE, $R^2$ and MAE values at the DWD site (Table 5). The two 30 m DEMs SRTM and ALOS show slight differences in the NSE values at the measurement stations, which can be attributed to differences in the sensing angles and post processing algorithms leading to topographic differences and as a

consequence also to differences especially in Qsi as mentioned in Section 3.2.

In summary, besides the simulation of snow depth with the reference setup, only the setups with the transferred data set from the DWD station at Mt. Wendelstein as well as the simulations with precipitation substituted by CHIRPS gave plausible results for the two measurement stations and showed no snow towers in any HRU. However, the validity of the simulated snow depth is largely limited by using all other global products, which means that for these products, the substitution of all meteorological

parameters using atmospheric model or hybrid or reanalysis data as model input for snow depth simulations is unfortunately not useful. Although it was possible to simulate plausible snow depth for some years and for some HRUs, e.g., at the LWD station, with the GLDAS setups as well as with ERA-20C and ERA-20C_corr, they are also classified as non-reliable as in several HRUs simulated snow towers piled up. In contrast to the global meteorological setups, the simulations of snow depth with all different DEM setups produced reasonable results.

## 3.4 Comparison of snow hydrologically relevant indices and the simulated runoff regime

In the following, we present the effect of different meteorological and DEM model input data sets on specific snow hydrological indices calculated for the entire RCZ and single HRUs as well as the catchment mean runoff. As the majority of products was already classified as unreliable in Section 3.3, we only chose the model runs that produced plausible snow depth results within all HRUs of the RCZ for the investigation in this section (Table 6). This encompasses besides the reference setup, the DWD_Wendelstein and CHIRPs setups as well as all DEM setups.

As specific snow hydrological indices, we chose the maximum snow water equivalent (MSWE), the day of MSWE (DMSWE), the snow cover duration and the number of ablation days. The latter is defined as the days between the DMSWE and the last day with snow cover. Since no SWE measurements are available in the investigated time period, we compare the simulation results to the results obtained with the reference setup. We refrained from evaluating the snow cover duration and the number of ablation days with measured snow depth data for two reasons. First, these values cannot be determined in most years at the LWD station, because of data gaps in snow depth measurements due to lightning strikes, which occur mainly in spring and autumn. Second, as already mentioned, the DWD snow depth measurements which are performed directly on the glacier surface might bias the parameters on snow cover duration and ablation periods due to a delayed melt out. As with SWE, we do not compare the simulated runoff with measured data. One reason for this is the existence of large gaps in the measured runoff data (months to years). Another reason is the generally poor data quality of measured runoff during the analyzed period which is due to maintenance issues in this harsh high alpine environment.

For the catchment mean and varying meteorological products, Table 6 shows an overestimation of MSWE in the CHRIPS and DWD_Wendelstein setup of 15% and 20%, respectively, compared to the reference setup. This overestimation is is related to the 378 mm and 534 mm higher mean annual precipitation sum in the CHIRPS and the DWD_Wendelstein setups. For the DWD_Wendelstein setup the combination with the lower mean temperature leads to a 12 days later occurrence of the DMSWE, a 22 days longer snow cover duration and a two weeks longer ablation period on average for the entire catchment. In contrast, the later DMSWE, the longer snow cover duration and the longer melting period of the CHIRPS setup is solely due to higher winter precipitation and the resulting higher MSWE. As a consequence, the higher MSWE values in these two meteorologically different setups result in a higher runoff of 29 and 30%, respectively (Table 6). The runoff regimes of both setups show considerable differences compared to the reference with its peak in June as presented in Fig. 8. The peak runoff of the CHIRPS setup occurs in June and July, whilst the peak runoff of the DWD_Wendelstein setup is in total one month later, which is due to lower spring temperatures that lead to a slower ablation and the longer availability of snow for melting in summer.

Regarding the simulated snow hydrological results using the setups with different DEMs, the mean catchment MSWE corresponding to the GTOPO30 DEM is closest to the reference with a negligible deviation of 1% (Table 6). This good performance goes in line with the comparison of the measured and modelled snow depth data in Section 3.3 (Table 5). Both 30 m resolution DEMs result in a considerably higher catchment mean MSWE than the reference setup at +18% in the ALOS setup and +17% in the SRTM setup. The GTOPO30 setup shows for the DMSWE and the length of the snow cover as well as the ablation periods also the smallest deviations compared to the reference setup. The mean catchment DMSWE of the ALOS and SRTM

setups occur eleven and 12 days later, respectively, when compared with the reference setup. Moreover, the snow cover dura-
tion in these setups is almost one month longer, which is reflected in their higher number of ablation days, too. This is due to
considerably less received Qsi in the ALOS and SRTM setups.

The variations in the modelled snowpack development are also evident in the runoff regime (Fig. 8). The higher MSWE and
its delayed occurrence in the ALOS and SRTM setups (Table 6) lead to a shift in peak runoff from June to July compared to
the reference. This is favoured by snow that is still available for melt in the radiation intense summer months. The fact that the
snow cover indices of the GTOPO30 setup are so close to the reference entails a very similar runoff behavior, despite it having
the largest warm bias, precipitation underestimation and insolation overestimation (Table 4). To explain this, the differences in
the process of discharge formation have to be examined. Although the GTOPO30 DEM is lower in altitude than the reference
DEM, it is still high enough that monthly mean temperature is below freezing until March and only at 0.1 °C for April while
it is at -0.7 °C April in the reference. Both, the reference and GTOPO30 setup are clearly above freezing with 4.2 °C and 5.1
°C, respectively, in May. The main melt, which is temperature induced in spring starts almost at the same time in both setups
illustrated by the SWE decline in Fig. 8. Melt induced runoff starts slightly earlier in the reference setup (Fig. 8), because there
are HRUs which receive considerably more Qsi than in the GTOPO30 setup (Fig. 4). Nonetheless, the major part of the RCZ
receives more Qsi in the GTOPO30 setup. The longer the days last in spring, the stronger becomes the radiation induced melt
effect which is why in the results of the GTOPO30 setup, melt induced runoff becomes stronger in June than in the reference
setup. In summer the lower elevation has a greater effect on the temperature than in winter which leads to higher melt induced
runoff in the GTOPO30 setup in June and July. This is illustrated in the faster decline of SWE in June and July in Fig. 8. This
effect decreases in August since less snow is left to melt. The higher peak runoff induced by the CHIRPS setup is due to a
higher snow mass availability for melt in general. The lower temperatures of the DWD_Wendelstein setup along with lower
radiation in the ablation period (Fig. 6) leads to a lower runoff in June, although more snow for melt is available. The peak
runoff subsequently occurs in July, when temperatures are higher. Both, the CHIRPS and DWD_Wendelstein summer runoff
peaks are increased by the higher amount of precipitation compared to the reference (Fig. 3).

In addition to the results shown for the catchment's average, Fig. 9 presents the 10-year mean MSWE, DMSWE, snow cover
duration and ablation periods for each single HRU, sorted according to their mean altitude (see Table 2). It is obvious that
overall all four snow hydrological parameters increase with altitude in every setup, indicating that altitude is the dominant factor
for snow cover development. Most considerable differences for all setups can be observed especially for MSWE. In the CHIRPS
and DWD_Wendelstein setups, the differences can mainly be explained with temperature and precipitation differences, as
temperature changes are mainly linear with altitude. Differences amongst the DEM-based setups mainly occur due to different
Qsi values, which are in turn dependent on slope, aspect and altitude. As previously mentioned the GTOPO30 causes quite
homogeneous HRU conditions. The same is true, with only slight losses in variability for the DEM setups ALOS and SRTM.
Nonetheless, the two 30 m resolution setups show considerable differences in aspect and slope compared to the reference
originating from the lower resolution. Compared to the reference, the simulated snow hydrological parameters of all tested
setups indicate higher values. Although the GTOPO30 results of the catchment's average is quite similar to the reference
setup, this picture is not true regarding the individual HRUs. The differences in all snow hydrological parameters between the

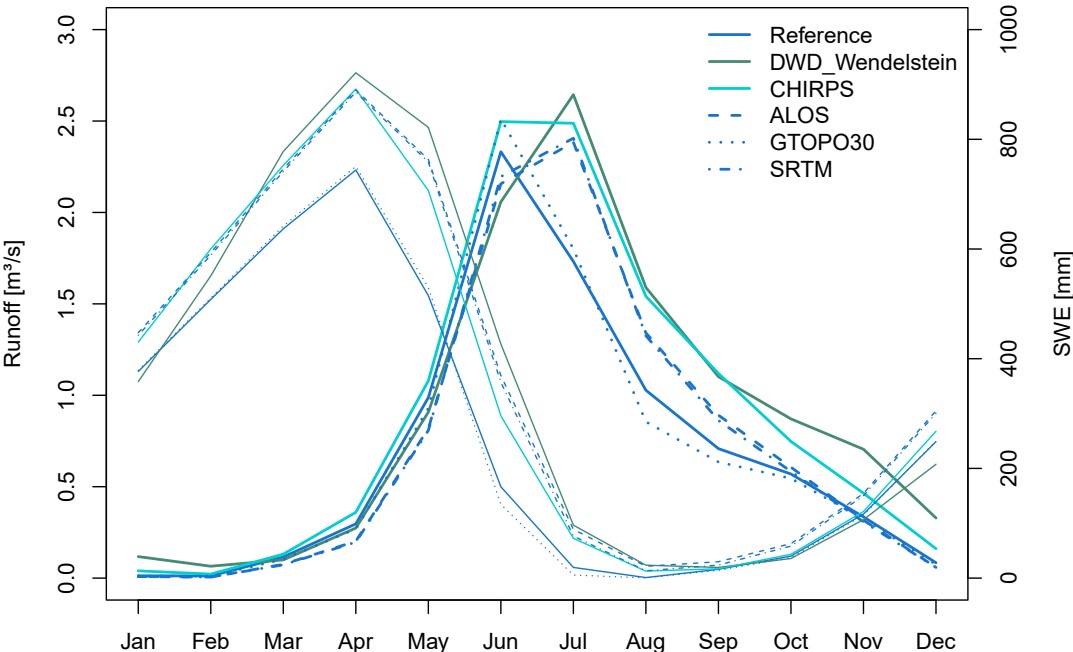

**Figure 8.** 10-year mean monthly catchment runoff (bold lines and refers to the left axis) and snow water equivalent (SWE, thin lines and refers to the right axis) of the Research Catchment Zugspitze for the five most plausible model setups in comparison to the reference simulations.

GTOPO30 and all other setups, which are particularly noticeable in HRU1 for MSWE, can mainly be attributed to the coarse
spatial resolution resulting in an almost 200 m lower mean altitude of HRU1 (see Table 2). Regarding the ablation period, HRU1, HRU6, and HRU10 show considerable differences between the reference and the GTOPO30 setup despite their very similar catchment mean values (see Table 6). The other seven HRUs covering 80% of RCZ are very similar to the results of the other setups, which indicates that this effect is averaged out when calculating catchment means. It could be argued that the evaluation of the simulation results on an HRU basis is insufficient due to the large heterogeneity of the catchment.
However, the chosen HRU delineation partly accounts for this heterogeneity since it directly relies on the LIDAR measured dominant snow depth distribution (Weber et al., 2020). Moreover, dominant snow depth patterns in high alpine regions are largely persistent over the years as e.g. Grünewald et al. (2013) show for various high alpine catchments. We are thereafter confident that the spatial distribution of the snow cover is quite well simulated.

In summary, although the five investigated setups, including the DWD_Wendelstein and CHIRPS meteorological setups and
all three global DEM setups, represented the snow depth quite well (Section 3.3). At a closer look, considerable differences in MSWE, DMSWE as well as the entire snow covered and the ablation periods were found compared to the reference. In addition, compared to the reference and the GTOPO30 setup, all other DEM setups indicate a higher mean monthly SWE. The differences in the snow cover, results also in obvious differences in the runoff regime.

## 4   Discussion on potential applications in ungauged basins

In the following, we discuss the potential applicability of global products for snow hydrological modelling in high alpine regions as exemplarily demonstrated for the RCZ in this study. In this regard, we focus on our two initial research questions on the reliability of globally available meteorological data products and on the influence of different globally available DEM products in such complex topography. In addition we discuss potential future developments.

### 4.1   Application of global meteorological data

Garen (2013) as well as Kundzewicz and Stakhiv (2010) state that the usage of remote sensing and atmospheric modelling data is not yet common or not even applicable in hydrological modelling of mountain regions of high spatial variability and steep topographies. Due to the rather low spatial resolution of these data products, their representation of atmospheric parameters is often too coarse to describe the complex high alpine situation. In this study, we confirm this statement for all investigated globally and publicly available meteorological data sets, from which we used all necessary meteorological parameters as

forcing data. According to the knowledge of the authors, studies comparing different global meteorological input data explicitly for high-alpine regions are very sparse and not available in the extent of this study. Therefore, a framework-based comparison of different studies in this regard is not possible at current stage. In the following, however, we discuss further studies, which point to a certain extent in this direction.

The replacement of precipitation only by using the CHIRPS data set produced valid model results for snow cover and runoff
for the RCZ. This quite good performance of the CHIRPS data set was also found by Satgé et al. (2019), who used it for hydrological modelling of the Lake Titicaca region. In general, the literature findings according to the overall quality of model results using hybrid meteorological input data are not uniform. However, it seems that better results can be achieved for study sites which are located in flatter terrain or which cover only a small proportion of alpine or high alpine areas. In contrast to our study results, e.g. Fuka et al. (2013) came to the conclusion that meteorological reanalysis data perform at least as well
as measured data at the catchment scale and sometimes even outperform in situ point measurements. Their test sites include a mountain watershed twice as large as RCZ and they also tested the CFSR data. Fuka et al. (2013) explain the good model performance with CFSR's ability to represent the average catchment meteorology well in contrast to point measurements. Thus, the weak performance we obtained with CFSR does not confirm their findings. In a study by Förster et al. (2016) focusing on the Gepatsch catchment in the Central European Alps, good results in hydrological modelling could be achieved
with Climate Forecast System Version 2 (CFSV2) driver data which represent a further development of CFSR (Saha et al., 2014). However, they modelled on a monthly basis and used in situ measurements to downscale CFSV2 data in advance — a step which cannot be done for an ungauged basin. Essou et al. (2016) also demonstrates a good performance of reanalysis data as hydrological model drivers including the CFSR and ERA-Interim. Although their study includes mountainous watersheds, the watersheds are much larger than the RCZ and do not include a high alpine headwater catchment. Therefore, their findings
cannot be compared one by one to our high alpine case. In a follow up study including the same watersheds, Essou et al. (2017) propose to use reanalysis data in combination with observation data for regional hydrological modelling to increase

performance particularly in snow-dominated catchments. Casson et al. (2018) found that the application of reanalysis data in modelling SWE of subarctic catchments is problematic but can be improved by integrating observed SWE data. In contrast to our study, they used a conceptual model to simulate snow cover development, which could also lead to differences in results

compared to the physically-based CRHM we used. The applied temperature correction scheme from Gao et al. (2012) for ERA-20C increased the performance of this hybrid data set. This is a strong indication that local downscaling of meteorological data in complex terrain is very valuable. Similar conclusions on such corrections are drawn by Buytaert et al. (2010); Teutschbein et al. (2011); Hay and Clark (2003). Thus, we would recommend applying the scheme to other global data, too. Regarding the quite new and higher resolved ERA5 and ERA5-Land data, we were quite surprised that both data sets are not able to

better capture the meteorological situation in the RCZ. In particular from the 9 km resolution ERA5-Land data, we expected better results. One reason for the bad performance is the direct vicinity of the RCZ to the surrounding much lower forelands. Therefore, even ERA5-Land might be too coarse to capture this situation. The comparably low pixel altitude of this product also supports this assumption. We assume that ERA5 and ERA5-Land could be applied with much more success in a central alpine catchment where the investigated catchment is surrounded by high-alpine regions, which might be more representative

for the landscape domain. This aspect could also be a reason for the bad performance of the other global data products in RCZ. As already mentioned in Section 3.1 another reason could be that inversion or Föhn conditions are insufficiently represented in global products in contrast to in situ measured data. Besides the good results with the hybrid CHIRPS precipitation product, we also achieved plausible model results for the data set DWD_Wendelstein, by "borrowing" meteorological driver data from another alpine AWS, as suggested by the PUB initiative (Liu et al., 2013). Regarding our results for different meteorological

setups, we can conclude that meteorological data transferred from stations in the closer vicinity in the same mountain range and with similar climatology still provide the best results, when all meteorological input data have to be substituted. According to Girons Lopez et al. (2020), who investigated the effect of different model structure choices of the HBV model in several catchments in mountainous areas in Central Europe, there may be pronounced differences in the model response on structure changes depending on the catchment. In this context, a potential limitation of our study is that only one high alpine catchment

has been tested for all the global meteorological products but so far we had no access to similar data for other high-alpine catchments.

## 4.2  Application of global DEM products

To the authors' knowledge, a comprehensive comparison of different globally available DEM products for various high alpine sites with very complex topography is not yet available. However, several studies which were carried out in other than explicitly

high alpine regions with either global or commercial products report that the spatial resolution of DEMs considerably affects the output of hydrological models. The reason is the influence of DEM resolution on the calculation pf topographic parameters or indices (Sørensen and Seibert, 2007; Vaze et al., 2010; Nagaveni et al., 2019). In general, the coarser the DEM, the larger the uncertainty of the simulations. In addition, Hopkinson et al. (2010) exemplarily investigated a glaciated area and described that the DEM resolution has also a direct impact on spatio-temporal variation in simulated meld due to the scale dependent

representation of the topographic characteristics. In our study we can confirm that the simulated SWE and runoff generation

is affected by the DEM resolution. However, at a first glance it seems that no matter which DEM resolution between 2.5 m and up to 1 km is used, the modelled results for the snow cover remain plausible to a certain extent. This is especially true for the two snow depth measurement sites LWD and DWD in the upper parts of the RCZ. As the GTOPO30 DEM is approx. 300 times coarser than the reference and 25 times coarser in resolution than ALOS and SRTM, this is particularly remarkable. However, on closer inspection, the GTOPO30 setup is not able to describe the topographic conditions for the individual HRUs as shown in Table 2. Slope angles are far too low and the aspect is up to 180 degrees different, which results in large Qsi offsets compared to the reference. However, the almost 100 m lower DEM-dependent average catchment altitude of GTOPO30 plays a minor role, since it is still high enough and temperatures are low enough that the snow cover can develop in the model as it was measured. Regarding the catchment scale, the differences among HRUs are averaged out. The GTOPO30 DEM was not created for applications in small basins with highly heterogeneous terrain and we generally would refrain from using it for such applications. Further investigations on the usage of coarser DEMs would be necessary for applications where they are required since large mountain domains should be modelled and computational power is limited. In contrast, the two 30 m resolution DEMs ALOS and SRTM are well applicable and we assume that they can be also applied with good success in other high alpine regions. , although some DEMs have the same spatial resolution, like ALOS and SRTM, they still might perform differently. One reason can be due to different techniques of DEM creation using stereo satellite imagery, interferometry or altimetry. Further differences can be due to different satellite viewing angles as well as post-processing techniques.

### 4.3   General remarks and potential future developments

The presented results show that the right choice of the meteorological forcing data is still the biggest challenge for snow hydrological modelling in ungauged alpine headwater catchments. The choice of the DEM has far less impact on the snow cover modelling, but can still result in considerable differences in snow cover as well as runoff generation. In particular for global meteorological products, a sound quality assessment of the results is inevitable, when the model is forced with non in situ measured meteorological data. However, our comfortable situation of having in situ data for evaluation is not the case in most high alpine catchments. In this regard, further tests with a broad range of different global data sets in the globally widely distributed alpine catchments of the International Network for Alpine Research Catchment Hydrology (INARCH) (http://words.usask.ca/inarch), a GEWEX project, would be a very helpful and logical next step. This would enable the creation of a framework of how to use globally available data for model forcing in ungauged high alpine basins. Regarding globally available meteorological setups, a framework categorization could include statements on catchment size and homogeneity, climatological variability as well as its topographic characteristics. Regarding the usage of global DEM products, an investigation especially on the degree of topographic complexity and especially the steepness of terrain would be most interesting.

In the next decades, model inputs could be improved by high resolution remote sensing products for measuring snow cover properties like snow cover extent, snow depth or even SWE. This includes using unmanned aerial vehicles (Bühler et al., 2015), satellite-based altimetry (Kwok and Markus, 2018), aerial LIDAR (Broxton et al., 2019) and optical tristereoscopic remote sensing (Shaw et al., 2020) for snow depth measurements, high resolution Synthetic Aperture Radar sensors for SWE derivation (Lievens et al., 2019), and satellite-based optical sensors (Notarnicola, 2020; Marti et al., 2016) or webcam data (Härer et al.,

2016) for spatially and temporally improved snow cover detection. This could enhance the evaluation or initialization of snow hydrological models. Although the new Sentinel satellite series has a high temporal resolution of up to 2.5 days, the 'realistic' temporal resolution especially for optical sensors can be far less due to cloud cover issues. Another point of improvement can be seen for the meteorological reanalysis and land data assimilation methods, as they are also constantly evolved (Hersbach et al., 2019). Products such as SNODAS (Barrett, 2003) and products based on it (Lv et al., 2019) indicate a good way forward

in this direction, including ground observations, satellite data and snow hydrological models, which are driven by globally available data to close the temporal and spatial gap between remotely sensed and ground based snow cover data. Another step in this direction is the direct usage of snow cover data from reanalysis products like ERA5 or ERA5-Land (Hersbach et al., 2020) that might provide information for model evaluation or initialization purposes. However, its current spatial resolution of 31 km and 9 km, respectively, is most probably too coarse for heterogeneous high alpine terrain as also seen in this study

regarding the meteorological data. However, we assume further spatial refinement of such products in the coming years. As with the global meteorological products, we also expect a constant improvement in spatial resolution for DEMs. Considering all these developments, we see great potential that snow cover products with high spatial and temporal resolution could be available with large improvements for many alpine headwater catchments in the future.

## 5   Conclusions

In this study, we evaluated the applicability of global data products regarding their application for snow hydrological modelling in a high alpine region for the potential use in ungauged basins. For the snow hydrological simulations, we set up the physically-based CRHM in the high-alpine RCZ of 12 km$^2$ in the Northern European Calcareous Alps for a study period of 10 years (September 2000 – August 2010). In the RCZ, we were able to evaluate 'ungauged basin' mode simulations with measured in situ data and reference simulations. We compared snow depth as well as snow hydrological parameters like MSWE, DMSWE,

snow cover duration and ablation period and runoff. To answer our first research question on the potential applicability of globally available meteorological data, we examined data from CFSR, different versions of GLDAS, CHIRPS, ERA5, ERA5-Land and ERA-20C including a specific downscaling approach, as well as a data transfer from another alpine station. All meteorological input data as well as the simulated snow hydrological results of the in total eleven different meteorological setups were compared to a reference relying on in situ measured data. To answer our second research question on the influence

of different global DEM products on snow hydrological modelling in complex terrain, we tested the impact of the GTOPO30 (approx. 1 km) as well as ALOS and SRTM DEMs (both 30 m). We compared these results to results produced by a setup using a 2.5 m high-resolution reference DEM.

Regarding all eleven meteorological setups we tested in comparison to the reference, we could only obtain plausible results for two setups for the catchment mean as well as for all HRUs. This is true for the snow depth development which was compared

to measurements at two sites and for other snow hydrological indices and the catchments runoff. Those two plausible setups are the CHIRPS setup in which precipitation only is substituted whilst taking all other meteorological data from the reference in situ data set, and the meteorological dataset which was 'borrowed' from another alpine AWS in the catchment's greater

vicinity at Mt. Wendelstein. In contrast, all other meteorological setups which are globally and publicly available data sets substituting all meteorological variables produced a totally unrealistic snow cover development for at least some parts of the catchment. This is especially true for the upper measurement station and the upper HRUs. To some extent, with the setups based on ERA-20C reanalysis data and different versions of GLDAS, we could simulate the snow cover development at one measurement site quite reasonably. The application of temperature downscaling improved the quality of the ERA-20C setup. The CFSR as well as ERA5 and ERA5-Land setups performed worst and produced so called 'snow towers' in all parts of the catchment due to too low temperatures, leading to the fact that the snowpack could not melt completely, even in summer.

The simulated results performed with the three setups relying on different topographic characteristics based on the different globally available DEMs, performed overall quite well in comparison to the references – even though they differ widely in their spatial resolution. As a consequence of their different spatial resolutions, product origin and product post-processing, variations in altitude, slope and aspect especially influence the radiation balance but also temperature and precipitation calculated for each HRU. Although, the coarsest DEM performed considerably well, this is due to the wrong reasons as small scale topographic effects were neglected. Despite the overall good performance of the three globally available DEM setups compared to all global meteorological setups, they also showed some considerable differences in MSWE, DMSWE, snow cover duration and ablation period at the catchment scale and for individual HRUs. This results in a shift in peak runoff of up to one month in some setups. Based on our results, the answer to our first research question is that it is not possible to exclusively use the tested global meteorological data products to reliably simulate snow depth and further snow hydrological parameters as well as runoff in the high alpine RCZ. One reason is that up to now, these global products are neither able to describe the meteorological heterogeneity of such complex catchments with steep terrain, nor its average conditions. This is reflected in a range of 3.5 °C in the catchment mean decadal temperature and 1510 mm in the catchment mean decadal annual precipitation sum over all input data for the RCZ. However, we assume that results could be different if the investigated catchment was not a topographic outlier in the landscape region as it is the case with RCZ with its adjacent lower forelands.

The answer to our second question is that compared to the influence of the different meteorological forcings on simulated snow hydrological parameters, the influence due to different characteristics of topographic parameters like slope, aspect and altitude due to different DEM products is smaller, even in complex terrain. Nonetheless, there are considerable differences mainly due to DEM product dependent variations in the radiation balance and due to mean HRU altitude induced variations in temperature and precipitation. Despite the weak performance of the global meteorological products, we assume that they might produce better results if the analyzed catchment is more representative for the surrounding larger scale landscape region. Furthermore, we expect a growing importance of such data in future snow hydrological modelling, also in ungauged basins, due to their constant and rapid evolution including temporal and spatial refinements. In order to generalize the findings from our study and to intensively test newly developed meteorological and snow hydrological products, we suggest to conduct further investigations in the well monitored catchments from INARCH.

*Data availability.* All global meteorological model input data as well as the global DEM data used for HRU parameterization are publicly available as described in Section 2. The URLs are:

- GLDAS: https://ldas.gsfc.nasa.gov/gldas

- ERA-20C: https://apps.ecmwf.int/datasets

- ERA5 and ERA5-Land: https://cds.climate.copernicus.eu

- CHIRPS: https://www.chc.ucsb.edu/data/chirps

- CFSR: https://cfs.ncep.noaa.gov

- ALOS: https://www.eorc.jaxa.jp/ALOS/en/aw3d30/index.htm

- GTOPO30 and SRTM: https://www.usgs.gov/centers/eros/science

The meterorological and snow depth station data from the German Weather Service (DWD) is also publicly available at https://www.dwd.de.
A second set of snow depth station data used for validation was provided by the Bavarian Avalanche Service (LWD) and can be made available upon request and approval by the LWD. The used reference DEM was provided by David Morche (Martin-Luther-University Halle-Wittenberg) and might also be made available upon request and upon approval by the copyright owner.

*Author contributions.* MW and MB conceptualized the study. MW was responsible for the methodology, carried out the investigations and the formal analysis and wrote the original draft. FK reviewed & edited the manuscript, acquired funds for publication and supervised the
work. KS reviewed & edited the manuscript and supervised the work.

*Competing interests.* The authors declare that they have no conflict of interest.

*Acknowledgements.* The authors want to thank David Morche (Martin-Luther-University Halle-Wittenberg) for providing the high resolution DEM of the RCZ. We gratefully obtained meteorological and snow height station data from the Bavarian Avalanche Service (LWD) and the Umweltforschungsstation Schneefernerhaus. Open access funding was provided by BOKU Vienna Open Access Publishing Fund.

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

**Table 2.** Main land cover and topographic parameters of the 10 hydrological response units (HRUs) of the Research Catchment Zugspitze derived from the reference, ALOS, GTOPO30 and SRTM DEM.

| HRU | altitude [m a.s.l.] | | | | aspect [°] | | | | slope [°] | | | | land cover | area [km²] |
|---|---|---|---|---|---|---|---|---|---|---|---|---|---|---|
| | ref | ALOS | GTOPO30 | SRTM | ref | ALOS | GTOPO30 | SRTM | ref | ALOS | GTOPO30 | SRTM | | |
| 1 | 2598 | 2596 | 2403 | 2585 | 160 | 163 | 119 | 162 | 44 | 11 | 7 | 39 | rock | 1.30 |
| 2 | 2104 | 2104 | 2109 | 2102 | 327 | 235 | 118 | 197 | 36 | 33 | 7 | 32 | rock | 0.96 |
| 3 | 2312 | 2307 | 2195 | 2288 | 243 | 230 | 123 | 214 | 46 | 12 | 6 | 39 | rock | 0.97 |
| 4 | 2321 | 2320 | 2215 | 2320 | 175 | 163 | 116 | 160 | 24 | 23 | 7 | 23 | rock | 1.81 |
| 5 | 2228 | 2227 | 2137 | 2230 | 29 | 83 | 112 | 114 | 22 | 21 | 7 | 23 | rock | 0.71 |
| 6 | 1687 | 1683 | 1591 | 1690 | 81 | 94 | 108 | 110 | 26 | 25 | 6 | 26 | krummholz | 1.03 |
| 7 | 1803 | 1802 | 1737 | 1806 | 155 | 153 | 118 | 151 | 40 | 39 | 10 | 37 | krummholz | 1.11 |
| 8 | 2329 | 2328 | 2213 | 2327 | 102 | 109 | 113 | 117 | 19 | 19 | 8 | 20 | rock | 2.71 |
| 9 | 2376 | 2375 | 2246 | 2360 | 59 | 109 | 117 | 118 | 51 | 45 | 6 | 40 | rock | 0.90 |
| 10 | 2615 | 2610 | 2506 | 2631 | 81 | 88 | 100 | 98 | 14 | 14 | 5 | 16 | ice | 0.23 |
| mean/sum | 2234 | 2233 | 2129 | 2258 | 136 | 140 | 115 | 143 | 31 | 24 | 7 | 28 | | 12.73 |

**Table 3.** Used meteorological data sets (Ta = temperature, rh = relative humidity, ppt = precipitation, Qsi = shortwave incoming radiation, Qli = longwave incoming radiation, u = wind speed, AWS = automatic weather station).

| data set | temporal resolution | type | spatial resolution | altitude [m a.s.l.] | parameters |
|---|---|---|---|---|---|
| reference | h | AWS | point measurement | 2964 | Ta, rh, ppt, Qsi, Qli, u |
| DWD_Wendelstein | h | AWS | point measurement | 1832 | Ta, rh, ppt, Qsi, cloud cover, u |
| GLDAS1 | 3h | hybrid | 0.25° | 1530 | Ta, rh, ppt, Qsi, Qli, u |
| GLDAS2 | 3h | hybrid | 0.25° | 1530 | Ta, rh, ppt, Qsi, Qli, u |
| GLDAS2_repro | 3h | hybrid | 0.25° | 1530 | T, rh, ppt, Qsi, Qli, u |
| GLDAS2.1 | 3h | hybrid | 0.25° | 1530 | Ta, rh, ppt, Qsi, Qli, u |
| ERA-20C | 6h | hybrid | 125km | 1287 | Ta, rh, ppt, Qsi, Qli, u |
| ERA-20C_corr | 6h | hybrid | 125km | 1287 | Ta, rh, ppt, Qsi, Qli, u |
| ERA5 | h | hybrid | 31km | 1157 | Ta, rh, ppt, Qsi, Qli, u |
| ERA5-Land | h | hybrid | 9km | 1427 | Ta, rh, ppt, Qsi, Qli, u |
| CFSR | 6h | hybrid | 0.20° | 1540 | Ta, rh, ppt, Qsi, Qli, u |
| CHIRPS | d | hybrid | 0.05° | 1909 | ppt |

**Table 4.** 10-year Research Catchment Zugspitze mean of annual air temperature and annual precipitation sum of the input data sets. Area weighted 10-year catchment mean of temperature, annual precipitation sum and short wave incoming radiation (Qsi) of the reference setup and the setups global meteorological data and land surface paremterization on basis of global DEMs.

| data set | temperature mean [°C] | precipitation [mm a$^{-1}$] | Qsi [W m$^{-2}$] |
|---|---|---|---|
| reference | 0.9 | 1830 | 139 |
| DWD_Wendelstein | 0.2 | 2364 | 133 |
| GLDAS1 | -1.0 | 1651 | 142 |
| GLDAS2 | 0.1 | 2066 | 134 |
| GLDAS2_repro | 0.2 | 2154 | 131 |
| GLDAS2.1 | -0.7 | 1660 | 131 |
| ERA-20C | -2.6 | 2347 | 151 |
| ERA-20C_corr | -1.2 | 2347 | 151 |
| ERA5 | -1.8 | 3038 | 143 |
| ERA5-Land | -2.5 | 2705 | 145 |
| CFSR | -2.0 | 3161 | 146 |
| CHIRPS | 0.9 | 2208 | 139 |
| GTOPO30 | 1.8 | 1691 | 147 |
| ALOS | 1.0 | 1814 | 124 |
| SRTM | 1.0 | 1809 | 116 |

**Table 5.** Statistical overview of simulation results of snow depth by using different meteorological input data and hydrological response unit parameterizations according to different DEMs and measurements performed at the measurement sites of the German Weather Service (DWD) and the Bavarian Avalanche Warning Service (LWD). NSE = Nash-Sutcliffe-Efficiency, $R^2$ = coefficient of determination, MAE = mean absolute error.

| data set | NSE LWD | NSE DWD | $R^2$ LWD | $R^2$ DWD | MAE LWD [m] | MAE DWD [m] |
|---|---|---|---|---|---|---|
| reference | 0.79 | 0.79 | 0.81 | 0.86 | 0.32 | 0.41 |
| DWD_Wendelstein | 0.33 | 0.11 | 0.59 | 0.66 | 0.61 | 0.84 |
| GLDAS1 | 0.46 | -6.25 | 0.49 | 0.12 | 0.51 | 2.28 |
| GLDAS2 | 0.31 | -195.84 | 0.47 | 0.01 | 0.57 | 15.15 |
| GLDAS2_repro | 0.36 | -266.87 | 0.55 | 0.01 | 0.55 | 17.94 |
| GLDAS2.1 | 0.14 | -8.11 | 0.33 | 0.39 | 0.46 | 3.59 |
| ERA-20C | -2.57 | -743.77 | 0.09 | 0.00 | 1.41 | 30.19 |
| ERA-20C_corr | -0.55 | -690.06 | 0.37 | 0.00 | 0.84 | 29.25 |
| ERA5 | -593.07 | -1648.16 | 0.07 | 0.00 | 22.13 | 42.18 |
| ERA5-Land | -669.14 | -1138.21 | 0.07 | 0.00 | 23.35 | 36.98 |
| CFSR | -25.53 | -37.22 | 0.07 | 0.00 | 24.50 | 42.08 |
| CHIRPS | 0.55 | 0.54 | 0.66 | 0.79 | 0.49 | 0.63 |
| GTOPO30 | 0.79 | 0.84 | 0.81 | 0.89 | 0.33 | 0.37 |
| ALOS | 0.77 | 0.68 | 0.83 | 0.89 | 0.33 | 0.50 |
| SRTM | 0.79 | 0.66 | 0.83 | 0.89 | 0.31 | 0.51 |

**Table 6.** Catchment 10-year means of maximum snow water equivalent (MSWE), day of maximum snow water equivalent (DMSWE), snow cover duration, ablation days and runoff in the Research Catchment Zugspitze.

| data set | MSWE [mm] | DMSWE [DoY] | snow cover duration [days] | ablation days | runoff [$m^3 s^{-1}$] |
|---|---|---|---|---|---|
| reference | 854 | 108 | 223 | 40 | 0.69 |
| DWD_Wendelstein | 1028 (+20%) | 120 | 255 | 54 | 0.90 |
| CHRIPS | 984 (+15%) | 113 | 239 | 52 | 0.89 |
| GTOPO30 | 849 (-1%) | 111 | 226 | 42 | 0.67 |
| ALOS | 1007 (+18%) | 119 | 247 | 50 | 0.74 |
| SRTM | 999 (+17%) | 120 | 248 | 51 | 0.74 |

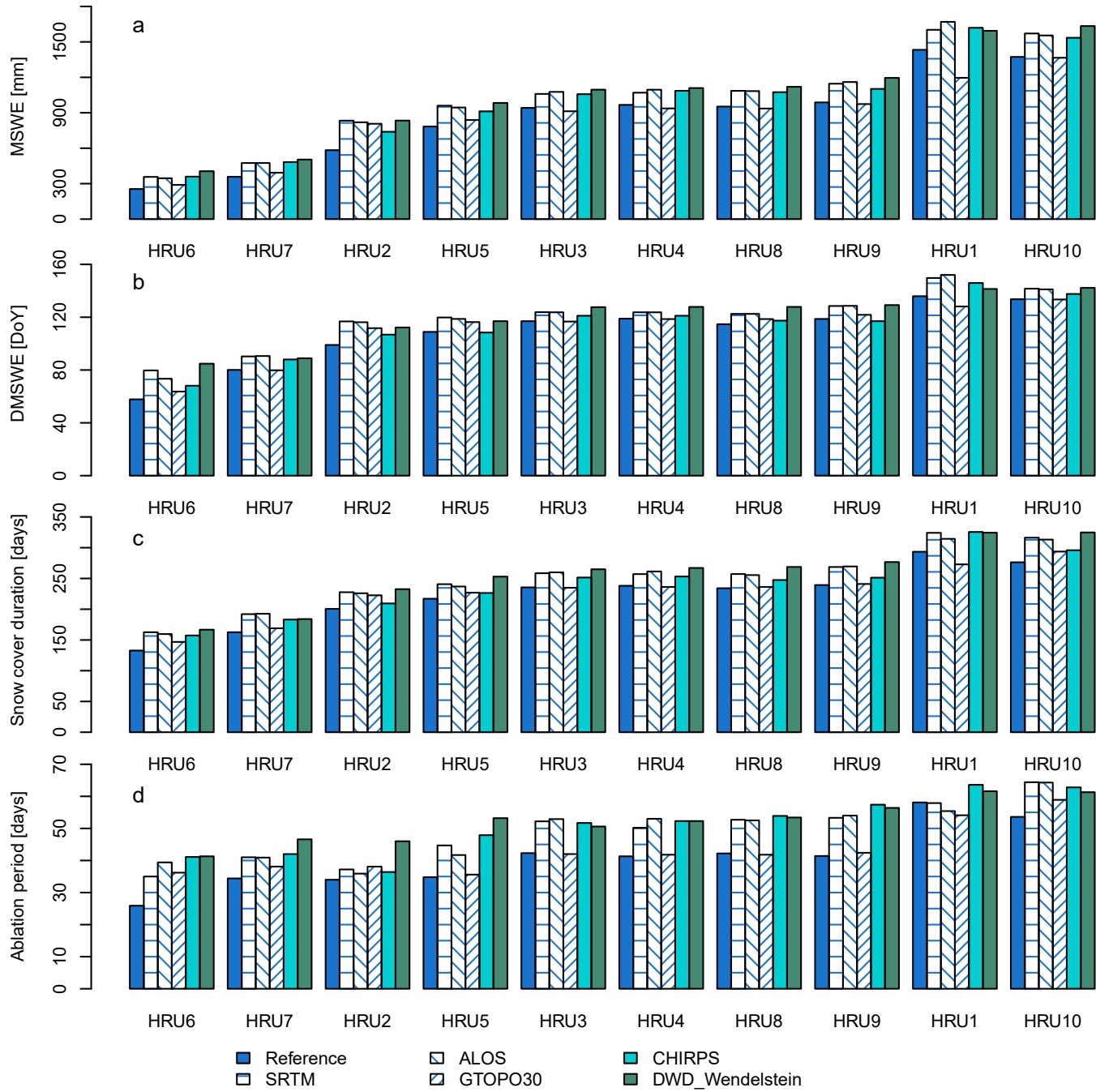

**Figure 9.** 10-year mean (a) maximum snow water equivalent (MSWE), (b) day of maximum snow water equivalent (DMSWE), (c) snow cover duration, and (d) ablation period simulated for individual hydrological response units (HRUs) within the Research Catchment Zugspitze for the five most plausible model setups in comparison to the reference simulations. The HRUs are sorted from the lowest to the highest altitude (see Table 2).