# Peer review of "The evaluation of the potential of global data products for snow hydrological modelling in ungauged high alpine catchments"

_Hydrology and Earth System Sciences, 2020_

## Referee Comment (RC1) · Anonymous Referee #1 · 3 Sep 2020

This is a review of "The evaluation of the potential of global data products for snow hydrological modelling in ungauged high alpine catchments" by Weber et al. The paper investigates the impacts of using a series of climate data products to force a hydrological model and it's advanced snow modules and compare the behavior in terms of snow process representation and runoff. The authors also compare the impact of modifying the DEM resolution to investigate the impact of using coarser (but free) DEMS compared to more refined (but expensive) ones such as LiDAR. The study takes place on a small, 12km2 catchment in Bavaria near the summit of Zugspitze. The authors find that the choice of DEM is not as critical as first thought, and that the use of global climate products can yield reasonable results in hydrological modelling but that there is

still improvements to be made. I have read the paper and found it very interesting and complete. The text flows generally well, although some expressions and sentences don't "sound" right and should be corrected by a native speaker. Scientifically, I have some issues with a few aspects of the work and I also have some suggestions to improve the work and make it more useful to the community. I will start by mentioning the more general points and end with smaller, more technical points.

General comments:

1- The authors use a variety of climate data products to drive the hydrological model to simulate the snow accumulation and melt processes. There are two station datasets (local and from a somewhat distant but similar catchment), satellite products and the ERA20C product. I have a problem with the latter. It can be argued that the $0.05°$, and to some extent the $0.2°$ / $0.25°$ products can be "reasonable" in terms of spatial resolution to represent the 12km2 catchment. However, the 125km resolution ERA-20C has a resolution of 125x125 = 15625km2, or more than three orders of magnitude difference. The catchment represents less than 0.08% of the tile size. It seems unreasonable to me to include it in the analysis. I think no researcher would use this product for such a small catchment in real world applications. The authors talk about using ERA-20C because of the correction of Gao et al. 2012, but I am positive that using a product such as ERA5-Land (With a $0.1°$ resolution) would be a better proposition.

2- The GTOPO30 product seems to give reasonable results and the authors state this in several places in the paper. However, it seems that it performs well because it is biased and it is "counteracting" the bias of the meteorological products. Therefore it is better, but for the wrong reasons. I think it would be warranted to add a section (or sentence) in the discussion to clarify this to prevent readers from getting the wrong impression of the quality of GTOPO30. Again, I think users working on very small catchments with high gradients would never use such a coarse product, it was not designed for this.

3- I think the authors should have compared the snow modelling results they obtain with those from a reanalysis directly, such as ERA5. This could be a much simpler way than using reanalysis meteorological data to drive a hydrological model. It seems to me that this would be a much simpler alternative than using these convoluted methods? I think it could be appropriate to at least mention the possibility here as it fits the bill perfectly: using publicly available global datasets to model snow hydrological modelling in alpine catchments. Furthermore, it could be used to force the initial states of the hydrological model to simulate runoff.

4- I notice that there is no section on model calibration, as this model does not require calibration but is instead "parameterized" to the environment. I suggest adding this information as it is atypical for a model to not require calibration.

Specific comments:

Lines 155-160: Do I understand correctly that all precipitations were multiplied by 1.5? The SWE technically also includes the effects of ablation/sublimation/transport, so I think it is dangerous to correct precipitation in this manner as the actual real factor is probably different. Perhaps add some limitations in the text here.

Lines 296-301: This section is a bit confusing. Also, there are 2 peaks of runoff caused by snow accumulation periods? There are two in the year?

Line 322: missing year for reference "Danielson and Gesch".

Line 336: "Adjusted it to the ALOS..." : Should this be altitude-corrected? Please clarify

Line 378: "shorter" should be "less"

Line 394: "snow towers" needs to be defined better.

Lines 413-422: This section is not clear upon reading. I needed to read more of the paper before coming back and understanding this section. Please simplfy and/or clarify.

---

## Referee Comment (RC2) · Anonymous Referee #2 · 7 Sep 2020

Review of "The evaluation of the potential of global data products for snow hydrological modelling in ungauged high alpine catchments" Weber et al.

General overview

The authors present an evaluation of snowpack and hydrological modeling outcomes in a well-characterized and gauged catchment using parameterizations derived from regional and global datasets. They also evaluate the influence of three different global digital elevation models on the derivation of model inputs such as slope, aspect, and solar insolation as well as the impact on model results. The goal was to illustrate and quantify the impacts of using these products to estimate snowpack and runoff in

ungauged catchments in snow-dominated mountainous regions.

General comments

The purpose and need for the study are reasonably clear and such work is important for the advancement of snow and mountain hydrology, in general. While it appears that the work and results are technically sound, I found the manuscript very difficult to follow and as such am unable to fully evaluate their results. I recommend a major revision. Specifically, I recommend: 1) reorganizing the methods and results to more concisely lay out the study (methods then results), including the use of additional tables or figures as recommended in the "Specific comments" section, and 2) rewriting the manuscript with an eye toward brevity and clarity, including working with an English-speaking editor. The manuscript is overly long, repetitive, and full of awkward and difficult to follow sentences.

Specific comments

Introduction: At some point in the introduction, there needs to be a clear statement or bulleted list of the research questions. These questions should then guide the organization of the rest of the paper.

Sections 2,3,4: These appear to be all methods, yet there are partial results mixed in. It would be much clearer, if it were a single methods section with subsections on the catchment, the model method, and the input datasets. Then the first subsection under results could compare the inputs of the various datasets, followed by the rest of Section 5. Otherwise, it is very difficult to follow.

Page 5, line 138. The term "gradient" should be replaced with "relief". "Relief" refers to the absolute difference in elevation of a region, whereas "gradient" refers to the slope.

Page 5, line 142. The term "knee wood" should be replaced by "krummholz", which is the more internationally recognized term for dwarf woody alpine vegetation.

Introduction, lines 112-124. Much of this should be moved to the methods section as it

is repetitive.

Page 5, line 160-199. A figure or table would greatly help the reader understand the model structure and underlying component calculations.

Page 8, line 201-202. Why is the reference referred to as representing the "gauged basin mode"? It would help if the methods section started out with statement about how the study is organized, such as "The approach we use is to model snow and runoff in the well-characterized RCZ using the CRHM and in-situ measurements. Then we repeat the modeling process using alternative datasets. In order to do this, we first describe the watershed, the model, the derivation of the datasets. . . ." Moreover, the derivation of the reference model needs to be more clearly explained. Are you using a single meteorological station's data (Mt. Zugspitze DWD station) to drive the catchment model and compare the results at two other stations in the catchment (LWD, DWD)? Use of standard three-letter station abbreviations for all stations in the study would help clarify this, too.

Page 9, lines 214-219. Place these in a table along with the specific dataset and reference

Page 10, line 257. What does PUB stand for?

Figures 3 and 4. These figures are introduced in the text and appear before section 4.3 which describes the DEM parameterizations. It was unclear, at first, why the DEM datasets were part of the graphs. It would be clearer to present methods in one full section and results in the next. As it is, some results are mixed in with each method section.

Page 18, line 432. What does 'thunderstruck' mean? Do you mean that there are data gaps to power outages caused by lightning strikes?

---

## Referee Comment (RC3) · Anonymous Referee #3 · 15 Sep 2020

Comments on the manuscript: "The evaluation of the potential of global data products for snow hydrological modeling in ungauged high alpine catchments" by Weber et al.

This manuscript examines the sensitivity of the simulated snow characteristics in a small alpine basin related to meteorological input data and DEMs using a hydrological model with a sophisticated snow module. Studies like this are of a great importance as high elevation snowpack critically affects alpine ecosystems and water resources in a number of regions in the world. The experiment is comprehensively designed and well executed. The paper is informative and interesting.

The main concerns on this manuscript include:

[Figure]

(1) The analysis of the experimental results and the organization as well as the presentation of the analysis output need improvements,

(2) Insufficient evaluation of the reference simulation across entire HRUs is also problematic. Because of the large elevation range within the test basin (RCZ), the snow field within the basin is expected to vary widely. Without the evaluation of the elevation-dependent model performance with the reference data, the accuracy of the reference run cannot be well established.

(3) The lack of the analysis and evaluation on monthly time scales (i.e., annual-cycle resolving analysis). This is important because the forcing and snow fields undergo strong seasonal cycle, hence, seasonal cycle-resolving analysis & evaluation can be useful in assessing key sources of the simulation errors. This will also help to link the seasonal cycle of the runoff to monthly snow ablation.

(4) The terminology "topography parameterization" is confusing. The true meaning of "topography parameterization" in the manuscript is "characteristics of topographic parameters" such as slope and azimuth that vary according to the DEM resolutions. "Parameterization" typically means representing unresolved properties using resolved values or representing a property using a related variable(s).

(5) The writing is OK, but I found occasional awkward/unfamiliar sentences. As a speaker of American English as the second language, I don't like to suggest any specific changes in grammar and writing. I strongly recommend the authors to consult native English speakers to go through the entire writing.

Specific comments and suggestions are presented in the below.

(1) The authors present results from elaborate model runs and analyses, but the key messages are not clearly presented and are sometimes confusing.

(2) Please improve Abstract so that the key findings in the experiment are presented more concisely clearly. Separating the sensitivity to DEMs and associated orographic

parameters from the sensitivity to the forcing data may help organizing with more clarity.

(3) The statement in Abstract, L23:24, contradicts the statement in Conclusion L625-634.

(4) L149:151: How the mean snow field (e.g., SWE, SCA) over the entire RCZ is calculated? This is among the key evaluation variables.

(5) L156:157: The met data at LWD are not used as the forcing. Why the snow precipitation at LWD is corrected for undercatch?

(6) Spell out HRU at its first appearance.

(7) L227: Check the spatial resolution of the CFSv2 data. The finest resolution I could find is T382 that correspond to approximately 0.313degree.

(8) L265: Section 3 → Section 4

(9) L288-291: I don't understand what "contrary development" indicates. This sentence is ambiguous. Please provide more explanations.

(10) L293-295: This sentence is ambiguous. Please rewrite.

(11) L288:289: This sentence cannot explain the peak in March.

(12) May change the title of Section 4.3. I suggest "Landsurface parameterization on basis of DEMs" → "Sensitivity of the land-surface parameters to DEMs (or DEM resolutions)"

(13) Also suggest a new title for Section 4.4: "Influence of the DEMs and associated land-surface parameters on meteorological conditions"

(14) L339: 100 m → 200 m (195 m). (200m elevation difference also corresponds to a lapse rate of 5K/km, slightly more stable than the standard atmosphere which is understandable over a cold surface like snow/ice.)

(15) Section 5: The evaluation based only on NSE and R2 is insufficient. Need more

metrics, at least the 'mean bias' and RMSE. Also provide additional evaluations for each month (i.e., annual-cycle resolving model evaluations)

(16) Section 5.1: If there are problems in evaluating the snow simulations at DWD and DLW as stated in Section 5.2, how can you justify evaluating the daily snowdepth against the observations at these sites?

(17) L395: "snow towers pile up" → Is this due to the forcing errors or model errors or combined focing-model errors?

(18) L395: 'downscaled temperature' → 'temperature downscaling'

(19) L406:421: This paragraph repeats the statements in the previous paragraph. May be removed and present a summary of this paragraph in Conclusion.

(20) Section 5.2

A. Statements in this section is too qualitative without solid supports from acceptable level of evaluations.

B. If missing data and site characteristics at LWD and DWD, respectively, prevent using these data for evaluation, how can we trust the reference data are accurate enough for evaluating model data?

C. If DMSWE cannot be validated, how can MSWE can be validated? MSWE is supposed to occur on DMSWE.

(21) L457:470: How can the data of largest warm bias, precipitation underestimation, and insolation overestimation perform best in simulating the amount and timing of runoff? This may indicate major flaw in the model physics and/or forcing combinations. Need further discussions. This also indicates the need for a more rigorous evaluations of the reference data and other model data.

(22) L475: Monthly Qsi needs to be evaluated as it is directly involved in snowmelt.

[Figure]

(23) L506:507: This may be an overstatement for the CHRPS data. CHRPS yields good results for snow cover and runoff, but not in NSE and R2 of the daily snow depth.

(24) 511-530: Can the inter-model difference also be related to their snow models? Do all of these models use the same snow model?

(25) Please discuss the poor NSEs and R2s with the transferred and CHRP data.

(26) L550-551: This statement ignores the substantial differences in runoff between GTOPO30 and ALOS/SRTM.

(27) L571: "the choice of the DEM has far less impact" Incorrect. DEMs have large impacts on the simulated streamflow annual cycle

(28) L600:604: Misleading. The 'borrowed' data performed poorly in terms of NSE and R2.

(29) L510-512: There is a mystery. How can the forcing data sets of such a wide variation can produce such similar simulations? This needs answers from the authors.

---

## Author Comment (AC1) · 6 Nov 2020

Response to Anonymous Referee #1

This is a review of "The evaluation of the potential of global data products for snow hydrological modelling in ungauged high alpine catchments" by Weber et al. The paper investigates the impacts of using a series of climate data products to force a hydrological model and it's advanced snow modules and compare the behavior in terms of snow process representation and runoff. The authors also compare the impact of modifying the DEM resolution to investigate the impact of using coarser (but free) DEMS compared to more refined (but expensive) ones such as LiDAR. The study takes place

on a small, 12km2 catchment in Bavaria near the summit of Zugspitze. The authors find that the choice of DEM is not as critical as first thought, and that the use of global climate products can yield reasonable results in hydrological modelling but that there is still improvements to be made. I have read the paper and found it very interesting and complete. The text flows generally well, although some expressions and sentences don't "sound" right and should be corrected by a native speaker. Scientifically, I have some issues with a few aspects of the work and I also have some suggestions to improve the work and make it more useful to the community. I will start by mentioning the more general points and end with smaller, more technical points.

Author's answer: We would like to thank the reviewer for carefully reading the manuscript and providing us with a very constructive and overall positive feedback. We have thoroughly considered all of his/her comments and address them point by point in the following.

General comments:

1 - The authors use a variety of climate data products to drive the hydrological model to simulate the snow accumulation and melt processes. There are two station datasets (local and from a somewhat distant but similar catchment), satellite products and the ERA20C product. I have a problem with the latter. It can be argued that the 0.05, and to some extent the 0.2 / 0.25 products can be "reasonable" in terms of spatial resolution to represent the 12km2 catchment. However, the 125km resolution ERA-20C has a resolution of 125x125 = 15625km2, or more than three orders of magnitude difference. The catchment represents less than 0.08% of the tile size. It seems unreasonable to me to include it in the analysis. I think no researcher would use this product for such a small catchment in real world applications. The authors talk about using ERA-20C because of the correction of Gao et al. 2012, but I am positive that using a product such as ERA5-Land (With a 0.1 resolution) would be a better proposition.

Author's answer: This is a very important comment and we are also convinced that

showing results of the quite new ERA5-Land product will improve our work. There-after, we absolutely agree with your suggestion, and already performed model runs with ERA5-Land (approx. 9 km) data for the updated version. With ERA5-Land we now were already able to achieve considerably better model results ($R^2$: 0.49 (LWD), 0.86 (DWD); NSE: 0.19 (LWD), 0.54 (DWD)) compared to all presented setups so far driven with global meteorological data. We will present and discuss these results in the revised version of the manuscript. Moreover, as we are also interested in the impact of different products and scales of global meteorological inputs, we will include ERA5 (ap-prox. 31 km) as well. The ERA5 product of the ERA family bridges the gap in terms of spatial scales to the very coarse product ERA-20C. We also already conducted model runs with ERA5 and could achieve even better results than with ERA5-Land ($R^2$: 0.53 (LWD), 0.77 (DWD); NSE: 0.52 (LWD), 0.75 (DWD)). We will also discuss shortly the fact that although ERA5 is coarser than ERA5-Land, it can lead to better results. More-over, we absolutely agree with you that the ERA-20C data set is actually too coarse for the application in such a small high alpine catchment. However, we still would like to include this product as it was widely used before ERA-5 came up and we also want to show the results in comparison to products with finer spatial resolution. In the text we will add that nowadays in real world applications (and especially for such a small high alpine catchment) this product will not be chosen (similar to the GTOPO30), but that we added it in terms of showing a broad variety of different products, including also 'older' and coarser global products. In addition, we would like to keep ERA-20C as we test with this product also the performance of the 'Gao et al. 2012' - correction method for ERA-20C temperature data, which was developed with data from RCZ. As already presented, these results showed a better performance compared to the uncorrected data. However, the new ERA5 and ERA5-Land setup performed considerably better.

2 - The GTOPO30 product seems to give reasonable results and the authors state this in several places in the paper. However, it seems that it performs well because it is biased and it is "counteracting" the bias of the meteorological products. Therefore it is better, but for the wrong reasons. I think it would be warranted to add a section (or

sentence) in the discussion to clarify this to prevent readers from getting the wrong impression of the quality of GTOPO30. Again, I think users working on very small catchments with high gradients would never use such a coarse product, it was not designed for this.

Author's answer: We agree with you that the GTOPO30 product was not designed for applications in small and highly heterogeneous catchments. We will point this out in the discussion section. Nonetheless, such a product might be useful in large mountain domains and if computational power is limited. Moreover, we will clarify the reason why the GTOPO30 product performed so well at a first glance. The reasonable results derived with the GTOPO30 DEM are, however, just specious due to the coarse resolution, where small scale topographic effects are averaged out. This leads to the fact that the meteorological products are interpolated in a too coarse topographic description of the catchment, which might level out the meteorological data. Therefore, for some parts of the catchment the values are too high, while they are too low in other parts. Nevertheless, it might well capture the mean topographic/meteorological situation of the catchment as it is the case in our study.

3 - I think the authors should have compared the snow modelling results they obtain with those from a reanalysis directly, such as ERA5. This could be a much simpler way than using reanalysis meteorological data to drive a hydrological model. It seems to me that this would be a much simpler alternative than using these convoluted methods? I think it could be appropriate to at least mention the possibility here as it fits the bill perfectly: using publicly available global datasets to model snow hydrological modelling in alpine catchments. Furthermore, it could be used to force the initial states of the hydrological model to simulate runoff. Author's answer: Indeed, the quite new ERA5-Land reanalysis product includes many very interesting variables such as snow height and SWE as an average over the entire pixel size of the product. We also agree with your statement that integrating ERA5-Land snow would be an interesting addition. In this regard, we will focus on the ERA5-Land SWE product and will compare

it to the specific snow hydrologic variables MSWE, DMSWE, snow cover duration, ablation shown in Table 6 for the entire RZC. In our opinion, it does not make sense to compare this ERA5-Land snow product to single HRU outputs or to compare it to the measurements at the DWD and LWD stations due to the very different spatial scales. In addition, we like your idea that such products could also be used to force initial states of the hydrological models in ungauged basins at certain time steps. However, this is out of the scope for this manuscript, but we will definitely mention that this would be very interesting to integrate in further studies.

4 - I notice that there is no section on model calibration, as this model does not require calibration but is instead "parameterized" to the environment. I suggest adding this information as it is atypical for a model to not require calibration.

Author's answer: We will follow your recommendation and will add the following sentences: "Unlike most hydrological models, the physically based CRHM does not require calibration. The only parameters that we adjusted for modelling are the previously named catchment/HRU specific physiographic parameters."

Specific comments:

Lines 155-160: Do I understand correctly that all precipitations were multiplied by 1.5? The SWE technically also includes the effects of ablation/sublimation/transport, so I think it is dangerous to correct precipitation in this manner as the actual real factor is probably different. Perhaps add some limitations in the text here.

Author's answer: We only corrected snow precipitation with this factor and agree with you that the real factor might be different for the named effects. However, the factor was determined in a way that minimizes these effects. To minimize the influence of sublimation, data with a 10 minute temporal resolution were used for the comparison of SWE and precipitation. Before the comparison, the effect of wind on the measured snow water equivalent was investigated. Therefore, time periods without snow fall, temperatures below freezing and strong wind were checked. This revealed no wind

induced snow redistribution that could be captured by the snow gauge. Nevertheless, there are effects, which could not be taken into account such as sublimation from the snowflakes during the snow fall event due to strong winds. As you suggested, we will add some limitations and clarifications to the text.

Lines 296-301: This section is a bit confusing. Also, there are 2 peaks of runoff caused by snow accumulation periods? There are two in the year?

Author's answer: We agree with you that this section is a bit confusing and thus will be rewritten. We just wanted to hint at the fact, that there are datasets (the two with in situ measured data, reference and Mt. Wendelstein) which have their precipitation maximum during the snow accumulation period, while the other data sets have their maximum in summer. There are not 2 peaks of runoff within a year on a monthly basis in each setup. The reference data shows a maximum in precipitation in March and November, which is during the snow accumulation season in the RCZ. The regime of Mt. Wendelstein is similar having also a maximum in March, however, without the second maximum in November. All other meteorological datasets indicate their maximum precipitation in summer although they show a local maximum in March. In this comparison, the CHIRPS precipitation data set is in-between. Its precipitation regime is similar to the reference; however, with a more pronounced maximum in August.

Line 322: missing year for reference "Danielson and Gesch".

Author's answer: We will add the year.

Line 336: "Adjusted it to the ALOS...:" : Should this be altitude-corrected? Please clarify

Author's answer: We will clarify the sentence: "...adjusted it to the ALOS, SRTM and GTOPO30 specific HRU altitude, slope and aspect (Table 2, for the methods refer to Section 4.1)..."

Line 378: "shorter" should be "less"

Author's answer: Will be corrected.

Line 394: "snow towers" needs to be defined better.

Author's answer: We will add the following definition: "Snow towers are an effect in snow hydrological modelling that occurs mainly at higher altitudes and describes the unrealistically high accumulation of snow over several years. Reasons can be the insufficient description of redistribution processes in the model or unrealistic meteorological driver data (Freudiger et al., 2017)."

Line 413-422: This section is not clear upon reading. I needed to read more of the paper before coming back and understanding this section. Please simplfy and/or clarify.

Author's answer: We will make some changes to the text to clarify this section: "Regarding the model setups with different topographic parameterizations on basis of the three different globally and publicly available DEMs ALOS, SRTM and GTOPO30, the snow cover development at both measurement was simulated realistically (Fig. 6g, h, Table 5). This results in $R^2$ - values above 0.8, and NSE values above 0.77, for all three setups at the LWD site. The results are very similar to the reference at the measurement stations (Table 2). The two 30 m DEMs SRTM and ALOS, show slight differences in the NSE values at the measurement stations, which can be attributed to the differences in their topography despite having the same resolution. Such an effect was also mentioned for Qsi in Section 4.2."

References:

Freudiger, D., Kohn, I., Seibert, J., Stahl, K., and Weiler, M.: Snow redistribution for the hydrological modeling of alpine catchments, WIREs Water, 4, e1232, doi:10.1002/wat2.1232, 2017.
* * *

---

## Author Comment (AC2) · 6 Nov 2020

Response to Anonymous Referee #2

Review of "The evaluation of the potential of global data products for snow hydrological modelling in ungauged high alpine catchments" Weber et al.

General overview The authors present an evaluation of snowpack and hydrological modeling outcomes in a well-characterized and gauged catchment using parameterizations derived from regional and global datasets. They also evaluate the influence of three different global digital elevation models on the derivation of model inputs such as

slope, aspect, and solar insolation as well as the impact on model results. The goal was to illustrate and quantify the impacts of using these products to estimate snowpack and runoff in ungauged catchments in snow-dominated mountainous regions. The purpose and need for the study are reasonably clear and such work is important for the advancement of snow and mountain hydrology, in general. While it appears that the work and results are technically sound, I found the manuscript very difficult to follow and as such am unable to fully evaluate their results. I recommend a major revision. Specifically, I recommend: . . .

Author's answer: We would like to thank the reviewer for taking his/her time to read the manuscript and providing us with a very constructive feedback. Furthermore, we thank him/her for mentioning the importance of this work regarding advancements in snow/mountain hydrology. We have thoroughly considered all of his/her comments and address them point by point in the following.

General comments:

1 – I recommend reorganizing the methods and results to more concisely lay out the study (methods then results), including the use of additional tables or figures as recommended in the "Specific comments" section

Author's answer: We will follow your recommendation and will restructure the manuscript. More details on this point are given in the answers to the relevant specific comments.

2 – I recommend rewriting the manuscript with an eye toward brevity and clarity, including working with an English speaking editor. The manuscript is overly long, repetitive, and full of awkward and difficult to follow sentences.

Author's answer: Following your suggestions, the updated version will be better streamlined and we will particularly take care that repetitions will be eliminated where possible. Regarding the language issue, a native speaker will perform a proof read on the

updated manuscript. Please find also more details in the respective answers to the specific comments.

Specific comments:

Introduction: At some point in the introduction, there needs to be a clear statement or bulleted list of the research questions. These questions should then guide the organization of the rest of the paper.

Author's answer: In the last paragraph of the introduction, we will present shortly but in more detail the structure of the paper, as we agree that this will improve better readability. Moreover, we will formulate our two main goals as research questions with bullet points in the introduction, both regarding the overall question how far it is possible to use globally available input data for snow-hydrologic modelling in ungauged high alpine catchments: • To which extent can we use different global meteorological data products varying in product type and spatial scale to simulate snow depth and further snow hydrological parameters as well as runoff in a high alpine catchment? • What is the influence of different characteristics of topographic parameters like slope, aspect and altitude due to different DEM products with different resolutions on snow hydrological simulations in a heterogeneous high alpine catchment?

Section 2, 3, 4: These appear to be all methods, yet there are partial results mixed in. It would be much clearer, if it were a single methods section with subsections on the catchment, the model method, and the input datasets. Then the first subsection under results could compare the inputs of the various datasets, followed by the rest of Section 5. Otherwise, it is very difficult to follow.

Author's answer: As noted in the general comments, we will restructure the manuscript according to your suggestions. The method section will now contain descriptions of the catchment, the model, and the input datasets. The Sections 4.2 and 4.4 will be moved as subsections to the results section.

[Figure]

Page 5, line 138: The term "gradient" should be replaced with "relief". "Relief" refers to the absolute difference in elevation of a region, whereas "gradient" refers to the slope.

Author's answer: Thank you for pointing out this linguistic subtlety to us. We will change "gradient" to "relief".

Page 5, line 142: The term "knee wood" should be replaced by "krummholz", which is the more internationally recognized term for dwarf woody alpine vegetation.

Author's answer: We will replace "knee wood" by "krummholz".

Introduction, lines 112-124: Much of this should be moved to the methods section as it is repetitive.

Author's answer: We will shorten this paragraph. In the updated version this paragraph will be in the methods part.

Page 5, line 160-199: A figure or table would greatly help the reader understand the model structure and underlying component calculations.

Author's answer: CHRM is a widely used model for investigating snow hydrological questions. A very general structure is presented in Weber et al. (2016). The modules and underlying component calculations are described in detail in Pomeroy et al. (2007) and the other papers cited in the paragraph. We will reference this and point out more clearly the structure and description of the model in the updated version.

Page 8, line 201-202: Why is the reference referred to as representing the "gauged basin mode"? It would help if the methods section started out with statement about how the study is organized, such as "The approach we use is to model snow and runoff in the well-characterized RCZ using the CRHM and in-situ measurements. Then we repeat the modeling process using alternative datasets. In order to do this, we first describe the watershed, the model, the derivation of the datasets...." Moreover, the derivation of the reference model needs to be more clearly explained. Are you using a single meteorological station's data (Mt. Zugspitze DWD station) to drive the catchment

model and compare the results at two other stations in the catchment (LWD, DWD)? Use of standard three-letter station abbreviations for all stations in the study would help clarify this, too.

Author's answer: We will follow your suggestion to shortly outline the method steps undertaken at the beginning of the methods section, which will increase readability. The reference setup is referred to as 'gauged basin mode' since it uses in situ measured driver data directly measured within the gauged RCZ. Indeed, we are using data from a single meteorological station for the reference meteorological data input. We will point this out more clearly in the updated version along with a better description of the reference data. The reason why we 'just' use the DWD data as meteorological input is the following, which will also be clarified in more detail in the manuscript: For the reference setup, we used meteorological data measured at the DWD station on Mt. Zugspitze since it is the best dataset in terms of continuity and data quality measured directly within the catchment. Due to some longer data gaps, especially at the end of the winter season and the summer time, we did not use the meteorological LWD data as driver data. We will add in line 145 that besides the gaps in summer, there are also some gaps during winter, mainly at the end of season. In general, we agree that three-letter abbreviations for all stations would be good. For the two in situ stations LWD and DWD in the RCZ we used three-letter abbreviations. To avoid confusion, we name the DWD station situated at Mt. Wendelstein 'DWD_Wendelstein'. We think that for this study it is ok to extend the latter station name, as we also do not use three letters to describe the results of the global meteorological and DEM datasets.

Page 10, line 257: What does PUB stand for?

Author's answer: The acronym PUB stands for predictions in ungauged basins and is defined on page 3, line 74.

Figures 3 and 4: These figures are introduced in the text and appear before section 4.3 which describes the DEM parameterizations. It was unclear, at first, why the DEM

datasets were part of the graphs. It would be clearer to present methods in one full section and results in the next. As it is, some results are mixed in with each method section.

Author's answer: We will follow your recommendation and will restructure the manuscript. Regarding the graphs you mentioned (Fig. 3 & 4), we think that in order to exceed a not too high number of graphs in the manuscript in total, and not to be repetitive, that the additional information showing the information related to the DEMs is ok. However, we will mention more clearly in the text that the graphs consist of more information than the meteorological input data sets described in the section, where the graphs are already presented and that the further datasets shown in this graphs, will be described in the following (sub)section(s).

Page 18, line 432: What does 'thunderstruck' mean? Do you mean that there are data gaps to power outages caused by lightning strikes?

Author's answer: We will change it to lightning strikes.

References

Pomeroy, J. W., Gray, D. M., Brown, T., Hedstrom, N. R., Quinton, W. L., Granger, R. J., and Carey, S. K.: The cold regions hydrological model: a platform for basing process representation and model structure on physical evidence, Hydrol. Process., 21, 2650–2667, doi:10.1002/hyp.6787, 2007.

Weber, M., Bernhardt, M., Pomeroy, J. W., Fang, X., Härer, S., and Schulz, K.: Description of current and future snow processes in a small basin in the Bavarian Alps, Environ Earth Sci, 75, 962, doi:10.1007/s12665-016-6027-1, 2016.

---

## Author Comment (AC3) · 6 Nov 2020

Response to Anonymous Referee #3

Comments on the manuscript: "The evaluation of the potential of global data products for snow hydrological modeling in ungauged high alpine catchments" by Weber et al. This manuscript examines the sensitivity of the simulated snow characteristics in a small alpine basin related to meteorological input data and DEMs using a hydrological model with a sophisticated snow module. Studies like this are of a great importance as high elevation snowpack critically affects alpine ecosystems and water resources in a number of regions in the world. The experiment is comprehensively designed and well

executed. The paper is informative and interesting.

Author's answer: We would like to thank the reviewer for thoroughly reading the manuscript and for providing us with very valuable suggestions for improvement. Moreover, we thank him/her for mentioning the importance of this work regarding improvements in the investigation of high alpine water resources which is relevant for numerous (ungauged) catchments worldwide. We have carefully considered all of his/her comments and address them point by point in the following.

General comments:

(1) The analysis of the experimental results and the organization as well as the presentation of the analysis output need improvements,

Author's answer: In general, we will reorganize the paper structure, which was also mentioned as an issue by reviewer 2 and we will better streamline and improve the presentation and analysis of the results. The paper will also be improved with an eye on brevity. As you make some clear suggestions on these points in the specific comments below, please also consider our answers to those points in the following.

(2) Insufficient evaluation of the reference simulation across entire HRUs is also problematic. Because of the large elevation range within the test basin (RCZ), the snow field within the basin is expected to vary widely. Without the evaluation of the elevation dependent model performance with the reference data, the accuracy of the reference run cannot be well established.

Author's answer: We agree with you that the snow cover in the basin can vary widely as it is the case in any high alpine region with complex topography and that it would be valuable if it was possible to evaluate the accuracy of the elevation dependent snow cover variation in more detail. We will discuss this point in more detail in the updated version. For the study time period 2000 to 2010, however, we unfortunately have no LiDAR data to compare the modelled with the measured snow depth distribution. However, as we found out in our former study for the RCZ (Weber et al., 2020) and which is also known from other studies (e.g. Grünewald et al. 2013), the spatial distribution in various years and also during the most of the season does not change much (except at the very beginning and the very end of the season). In the study of Weber et al. (2020), we were able to include information of several LiDAR snow depth data sets derived in the years 2015 and 2016 which were conducted for the RCZ to delineate HRUs that guarantee for optimal representation of the snow cover distribution in RCZ. This HRU delineation was, as mentioned in the manuscript, also used for this study. We are thereafter confident that the spatial distribution of the snow cover is quite well simulated in the reference setup. Moreover, we think that the availability of two snow depth gauges in such a small catchment is already a lot and we are quite lucky with the situation in this well gauged high alpine catchment. Of course, it would be nice if each of the ten HRUs was represented with a snow depth gauge; however, this is not available and will most probably not be found in such a potential station density anywhere in the world. Thereafter, the only possible accuracy analysis for this study period is the investigation of snow depth with data measured at the DWD and LWD station which are situated at least at different altitudes. Moreover the locations of these stations were chosen from the two operating institutions to be as representative as possible for the respective altitude, e.g. no shadowing against wind but also no particular exposure, relatively flat terrain, etc.

(3) The lack of the analysis and evaluation on monthly time scales (i.e., annual-cycle resolving analysis). This is important because the forcing and snow fields undergo strong seasonal cycle, hence, seasonal cycle-resolving analysis & evaluation can be useful in assessing key sources of the simulation errors. This will also help to link the seasonal cycle of the runoff to monthly snow ablation.

Author's answer: We agree with you and will integrate an analysis of monthly ablation in the revised version. This will also complete the monthly analysis we already performed for precipitation and runoff. Please also consider our answers to your specific

comments on this point.

(4) The terminology "topography parameterization" is confusing. The true meaning of "topography parameterization" in the manuscript is "characteristics of topographic parameters" such as slope and azimuth that vary according to the DEM resolutions. "Parameterization" typically means representing unresolved properties using resolved values or representing a property using a related variable(s).

Author's answer: You are right, this was misleading. We will change the wording in the text where necessary.

(5) The writing is OK, but I found occasional awkward/unfamiliar sentences. As a speaker of American English as the second language, I don't like to suggest any specific changes in grammar and writing. I strongly recommend the authors to consult native English speakers to go through the entire writing.

Author's answer: Before submitting the reviewed version of the manuscript, we will give it to a native speaker for prove reading.

Specific comments:

(1) The authors present results from elaborate model runs and analyses, but the key messages are not clearly presented and are sometimes confusing.

Author's answer: We will revise the entire paper and streamline the text so that the main goals of our study and the key messages become clearer. Therefore, in the introduction, we will add our two main research questions as bullet points, which are both regarding the overall question how far it is possible to use globally available input data for snow-hydrologic modelling in ungauged high alpine catchments: • To which extent can we use different global meteorological data products varying in product type and spatial scale to simulate snow depth and further snow hydrological parameters as well as runoff in a high alpine catchment? • What is the influence of different characteristics of topographic parameters like slope, aspect and altitude due to different

DEM products with different resolutions on snow hydrological simulations in a heterogeneous high alpine catchment? We will provide the answers to these questions more clearly in the relevant discussion and conclusion parts as well in the abstract, so that the key message of our results is better visible. Since we already performed additional model runs with the quite new global products ERA5 and ERA5-Land data as suggested by reviewer 1, our updated key messages are regarding: a. the investigated global meteorological data sets: In total, we investigated 12 different meteorological setups considering a variety of different data products (ERA, GLDAS, CFSR, CHIRPS) covering different scales and versions of the products and an additional specific downscaling method for one product. However, only five meteorological setups including the reference in situ data setup showed agreeable results regarding the simulations of snow depth and other snow hydrological parameters, like total snow cover and ablation period, and the date and quantity of maximum SWE. However, in the results of these five setups considerable differences in the runoff regimes were found with a delay of peak runoff of up to one month. The applied globally available CHIRPS data performed in total agreeably well and gave the best results regarding all other meteorological setups. However, this setup covers the specific case that only precipitation was substituted and other meteorological driver data are still taken from the measured in situ data. Our investigations on the assumption that the entire data set of meteorological forcing is required, the newly added ERA5 and ERA5-Land products, that have a comparably high spatial resolution regarding all other applied global meteorological products, produced the best results. This especially applies to the results of snow depth simulated at the two meteorological stations as well as the catchment mean regarding the further hydrologic relevant snow cover parameters. In total, ERA5 products performed even better than the data transferred from a similar catchment. The other tested globally available meteorological data showed very weak performance in simulating the snow depth at the two stations and lead to absolutely unrealistic snow cover developments and were thereafter excluded for the more detailed analysis of temporal and quantitative comparisons regarding the catchment mean and on HRU

basis of the further snow hydrological parameters and the runoff. b. the investigated different globally available DEMs: In total, we used four different products with a wide span width of spatial resolutions from 2.5 m (LiDAR derived reference DEM setup) up to 1 km (GTOPO30). Two products we used have the same resolution with 30 m (SRTM and ALOS) to check the potential differences in results, although the spatial resolution is the same. Regarding the three globally available DEMs and the reference setup used for the parameterization of the surface characteristics, all DEM setups reproduced the measured snow depth and the further snow hydrological parameters on HRU and catchment scale quite well. However, they show considerable differences in the runoff regime, which is especially the case for the two DEMs of 30 m resolution in comparison to the reference. Despite the fact that the very coarse GTOPO30 DEM performed relatively well on the catchment mean, we advise against using this product in such heterogeneous high alpine terrain since the small-scale differences cannot be captured. The key messages of our results will be also better connected to our following overall suggestions regarding the applicability of such global products in ungauged basins in general. The formulation of the key messages will also be improved in the abstract (as also mentioned in the next comment).

(2) Please improve Abstract so that the key findings in the experiment are presented more concisely clearly. Separating the sensitivity to DEMs and associated orographic parameters from the sensitivity to the forcing data may help organizing with more clarity.

Author's answer: As mentioned in the answer before, we will define the main questions and key messages much clearer and will also restructure the abstract according to your suggestions.

(3) The statement in Abstract, L23:24, contradicts the statement in Conclusion L625-634.

Author's answer: We will rewrite the statement in the abstract to avoid this contradiction.

(4) L149:151: How the mean snow field (e.g., SWE, SCA) over the entire RCZ is calculated? This is among the key evaluation variables.

Author's answer: CRHM calculates for each HRU and each time step values of SWE and snow depth. For these calculations, CRHM requires a meteorological input for each HRU. Since such data is not automatically available for each HRU, we used the method developed by Liston and Elder (2006) to generate it for each HRU. We use the modelled values of SWE and snow depth to calculate the area weighted mean value for the entire RCZ. We are confident, that the chosen HRU delineation is suitable for realistic snow cover simulations since it is the same as applied in Weber et al. (2020), where we established these HRUs and evaluated them with spatially distributed LIDAR snow depth measurements. We will describe this better in the revised manuscript and hope that this is the answer to your question since L149:151 do not deal with the calculation of the mean snow field over the entire RCZ.

(5) L156:157: The met data at LWD are not used as the forcing. Why the snow precipitation at LWD is corrected for undercatch?

Author's answer: The meteorological data at the LWD station has, as already stated in the manuscript, some longer data gaps (especially at the end of the winter seasons and during summer time in the study period of 2000-2010), which is why we refrained from using it as meteorological forcing data. Since 2014, a snow scale was installed at the LWD station, which enabled us to use these measured SWE for more recent years as a good possibility to investigate the precipitation under catch. Similar to the literature reported under catch of snow precipitation of up to 50% (WMO, 2011; Grossi et al., 2017), we also found out in our former study (Weber et al., 2020) that we can expect an under catch of 50% for the RCZ. Assuming that under catch variations in precipitation over the entire catchment are rather negligible, the DWD snow precipitation has been corrected with the factor derived at the LWD station. We will clarify this in the revised manuscript.

[Figure]

(6) Spell out HRU at its first appearance.

Author's answer: We will spell it out at its first appearance. We can also offer to provide a list of acronyms.

(7) L227: Check the spatial resolution of the CFSv2 data. The finest resolution I could find is T382 that correspond to approximately 0.313degree.

Author's answer: We checked the spatial resolution again and thank you for the hint! In course of this, we realized that we misunderstood the data description of the database where we downloaded the data. As you wrote, the spatial resolution of CFSv2 is approx. 0.313 degree. However, CFSv2 only exists from 2011 onwards. All data before and thus the data we used is data from the Climate Forecast System Reanalysis (CFSR) which has a 0.2 degree resolution. We will change this throughout the manuscript and rename this setup CFSR.

(8) L265: Section 3 → Section 4

Author's answer: We will change that.

(9) L288-291: I don't understand what "contrary development" indicates. This sentence is ambiguous. Please provide more explanations.

Author's answer: We referred to the wrong Figure and will correct that. It is Figure 3b instead of 3a. This figure shows that precipitation in the reference and at the DWD_Wendelstein setup is contrary over the years 2005 – 2010.

(10) L293-295: This sentence is ambiguous. Please rewrite.

Author's answer: We will rewrite the sentence.

(11) L288:289: This sentence cannot explain the peak in March.

Author's answer: We guess you mean L.298-298 and thank you for pointing this out. This sentence makes indeed no sense and will be deleted.

(12) May change the title of Section 4.3. I suggest "Landsurface parameterization on basis of DEMs" → "Sensitivity of the land-surface parameters to DEMs (or DEM resolutions)"

Author's answer: We will change the section sub title according to the reviewer's suggestion.

(13) Also suggest a new title for Section 4.4: "Influence of the DEMs and associated land-surface parameters on meteorological conditions"

Author's answer: We will change the section sub title according the reviewer's suggestion.

(14) L339: 100 m → 200 m (195 m). (200m elevation difference also corresponds to a lapse rate of 5K/km, slightly more stable than the standard atmosphere which is understandable over a cold surface like snow/ice.)

Author's answer: The difference in catchment mean altitude is 105 m which is roughly 100 m. We will write the exact value instead of roughly 100 m to avoid confusion and also add the catchment mean values in Table 2.

(15) Section 5: The evaluation based only on NSE and R2 is insufficient. Need more metrics, at least the 'mean bias' and RMSE. Also provide additional evaluations for each month (i.e., annual-cycle resolving model evaluations)

Author's answer: We will additionally provide the mean average error (MAE) and, as previously suggested, we will also include an analysis on a monthly basis of the ablation and will discuss this also in the context of the already provided analysis on seasonal precipitation and runoff.

(16) Section 5.1: If there are problems in evaluating the snow simulations at DWD and DLW as stated in Section 5.2, how can you justify evaluating the daily snowdepth against the observations at these sites?

Author's answer: In section 5.2 we state that we do not validate the snow cover duration and the number of ablation days with measured values, mainly due to data gaps at the LWD station in spring, which make it impossible to determine the melt out day as well as the exact length of snow cover duration for some years. In addition, at the DWD station, the melt out day may be delayed due to the fact that the snow stake is installed directly on the small glacier 'Nördlicher Schneeferner'. This might bias the definition of snow cover duration and ablation days. Nonetheless, we can very well quantitatively validate the daily modelled with the measurements snow depth for the entire season when data is available (this is true for most days except the days, where we had data gaps at the LWD station at the end of the season). Of course, there might be years regarding the DWD station, in which the model is not able to capture the exact melt out date of the snow cover due to the mentioned effect (ice is conserving the snow cover). However this can almost be neglected in a day based comparison since this snow cover is usually very thin and hardly affects the total seasonally accumulated snow volume. We will clarify the above mentioned points in more detail in the updated version.

(17) L395: "snow towers pile up" → Is this due to the forcing errors or model errors or combined focing-model errors?

Author's answer: This is a well-known effect, which might occur if the meteorological input is not realistic, which is the case for some input setups we showed. The model structure, however, is always the same in each of our setups and such snow towers could never be observed when forcing the model with in situ measured meteorological data. The reason for this snow towers is mentioned in the following sentences: "These simulated snow towers are mainly caused by lower mean annual temperatures and higher amounts of precipitation compared to the meteorological values of the reference data set (Section 4.2)." To clarify this in more detail we also added the following definition: "Snow towers are an effect in snow hydrological modelling that occurs mainly at higher altitudes and describes the unrealistically high accumulation of snow over

several years. Reasons can be the insufficient description of redistribution processes in the model or unrealistic meteorological driver data (Freudiger et al., 2017)."

(18) L395: 'downscaled temperature' → 'temperature downscaling'

Author's answer: We guess the reviewer means L399. We will change it.

(19) L406:421: This paragraph repeats the statements in the previous paragraph. May be removed and present a summary of this paragraph in Conclusion.

Author's answer: We will remove this paragraph and we will also revise the manuscript with an eye on deleting repetitions in general.

(20) Section 5.2

A. Statements in this section is too qualitative without solid supports from acceptable level of evaluations.

Author's answer: We partly agree and partly disagree with you. On the one hand, as suggested, we see that it is necessary to also include an analysis of ablation on a monthly basis to better link the snow cover to the presented runoff simulation results. On the other hand, our statements in this section were already supported by "hard" facts and statistical measures as presented in the Tables 2, 4, and 6. Of course, it would be good to perform further statistical analysis with the presented indices. However, we have only ten years regarding values per index and model run and would need many more years to have enough data for a more in depth statistical analysis. In addition, we do not consider this as necessary since our objective is to demonstrate if it is possible to simulate the snow cover and runoff of our catchment with globally available data and subsequently to show the differences between the model runs.

B. If missing data and site characteristics at LWD and DWD, respectively, prevent using these data for evaluation, how can we trust the reference data are accurate enough for evaluating model data?

[Figure]

Author's answer: This concern is reasonable, probably for all measurement stations worldwide. DWD station: this data is quality-controlled by the DWD, the German Weather Service. Measurement errors are filtered and there are virtually no gaps. LWD station: The instruments of this station are regularly maintained by the LWD, the Bavarian Avalanche Warning service since they rely on good quality data as well. All, the large gaps at this station used to occur in late spring when the avalanche season was over and the LWD no longer produces an avalanche warning bulletin. So they simply had no need for data in this time of the year. So the data recorded at that location is the most accurate one can get. Regarding the location of the measurement stations, both institutions chose a location which is as representative as possible for the altitude. Further restrictions particular concerning the use of DWD data for snow cover duration evaluations are explained in the answer to comment 16.

C. If DMSWE cannot be validated, how can MSWE can be validated? MSWE is supposed to occur on DMSWE.

Author's answer: We never validated the MSWE with measurements and never stated that in our manuscript, as we do not have measured SWE during the investigated time period. Regarding measurements, we only compared the measured snow depth at the LWD and DWD to validate our model runs. Regarding all SWE related snow hydrological parameters (MSWE, DMSWE, snow cover duration, ablation period), we compared the simulated results of all investigated setups to the results of the reference setup, which is forced with in situ measured driver data at the DWD station and which was parameterized with the high resolution DEM.

(21) L457:470: How can the data of largest warm bias, precipitation underestimation and insolation overestimation perform best in simulating the amount and timing of runoff? This may indicate major flaw in the model physics and/or forcing combinations. Need further discussions. This also indicates the need for a more rigorous evaluations of the reference data and other model data.

Author's answer: This is a good point and will be discussed in more detail in the updated version. To answer this, we first try to explain a bit more in detail what actually happens during discharge concentration in our model setups and what are the local conditions. In reality, there is no surface runoff in RCZ due to its karstic situation. However, CRHM has no routine to route karst runoff. Moreover, there is not enough knowledge of the hydrological response of the karst system until now. Therefore it is almost impossible to appropriately parameterize the routing in CRHM, which is why the runoff presented in our study can be regarded as direct runoff. This means that the surface structure due to the DEM has almost no influence on runoff routing. So the differences are in the processes of discharge formation. Despite the GTOPO30 DEM is much lower in spatial resolution than the reference DEM, it is still high enough or has the effect that the monthly mean temperature is below freezing until March and around 0°C (0.1°C) for April while it is -0.7°C in April in the reference. However, both, the reference and GTOPO30 setup, are clearly above freezing level in May at 4.2 °C and 5.1 °C, respectively. This means that the main melt, which is also temperature induced in early spring, starts almost at the same time in both setups. As Figure 7 shows, the melt induced runoff starts slightly earlier in the reference setup, because overall higher HRUs in the catchment, receive overall considerably more Qsi in the reference than it is the case in the GTOPO30 setup. The Qsi values of GTOPO30 are due to the coarse resolution rather the same over the catchment. Nonetheless, the major part of the RCZ receives in total more Qsi in the GTOPO30 setup. The longer the days last in later spring, the stronger the radiation induced melt effect becomes, which is why in the GTOPO30 setup, melt induced runoff becomes stronger in June than in the reference setup. Moreover, in summer the lower elevation has a greater effect on the temperature than in winter which leads to higher melt induced runoff in the GTOPO30 setup in June and July. This effect decreases in August since less snow is left to melt. We will add further discussions on this point to the text.

(22) L475: Monthly Qsi needs to be evaluated as it is directly involved in snowmelt.

Author's answer: We agree with you and will take a closer look at the monthly radiation and also at the radiation distribution. As mentioned in the answer to comment 21 there are HRUs, which receive significantly overall more Qsi in the reference than in the GTOPO30 setup, and therefore, show and earlier melt in the reference setup. However, there are overall also larger parts (upper, medium and lower parts) in the GTOPO30 setup, which receive in total more radiation than in the reference setup. This effect gains power with longer days and leads to a pronounced melt in June in the GTOPO30 setup.

(23) L506:507: This may be an overstatement for the CHRPS data. CHRPS yields good results for snow cover and runoff, but not in NSE and R2 of the daily snow depth.

Author's answer: The NSE and $R^2$ of the CHIRPS setup for daily snow depth is clearly the best among all presented globally available different meteorological setups. Despite the quality measures are lower than for the reference, they are still considerably high.

(24) 511-530: Can the inter-model difference also be related to their snow models? Do all of these models use the same snow model?

Author's answer: The inter-model difference can also be related to the used snow models. As explained in the paragraph, the studies cannot be compared one by one since they are conducted on different scales, in different (micro-) climatic regions and also with different models. However, these studies show that it is generally possible to obtain reasonable results with global data, if conditions are appropriate, e.g. data is available for downscaling or the investigation of larger scales.

(25) Please discuss the poor NSEs and R2s with the transferred and CHRP data.

Author's answer: As stated in the answer to comment 23, we consider the quality measures of the CHIRPS setup to be relatively good. According to Moriasi et al. (2007) and NSE > 0.5 can be considered as satisfactory in hydrological modelling. This threshold

is also used in alpine hydrological modelling (e.g. Rahman et al., 2013). The reason for the good performance of the CHIRPS setup is that the forcing data is the same as in the reference setup except for precipitation. As illustrated in Figure 4, CHIRPS precipitation well reflects the precipitation regime in RCZ. However, it is biased to higher values. Regarding the transferred data from Mt. Wendelstein, in particular the NSE values are worse. This can be traced back to the years 2002, 2005 and 2010 in which the peaks are strongly overestimated regarding the DWD_Wendelstein run. The NSE is particularly sensitive to large values. We will add a discussion on this.

(26) L550-551: This statement ignores the substantial differences in runoff between GTOPO30 and ALOS/SRTM.

Author's answer: We agree with you and will rewrite the sentence to account for the differences in runoff.

(27) L571: "the choice of the DEM has far less impact" Incorrect. DEMs have large impacts on the simulated streamflow annual cycle

Author's answer: We will clarify this sentence, because in comparison to the setups varying in meteorological driver data, the impact is less.

(28) L600:604: Misleading. The 'borrowed' data performed poorly in terms of NSE and R2.

Author's answer: We will discuss this to be clear that the 'borrowed data' from another station in a similar catchment performs worse than the CHIRPS setup.

(29) L510-512: There is a mystery. How can the forcing data sets of such a wide variation can produce such similar simulations? This needs answers from the authors.

Author's answer: We are not sure if you really mean L510-512 since we did not find a statement there that fits to your comment. We assume you perhaps could have meant L610-612, which is about differences due to DEMs. The answer why the different DEM setups performed quite similar is that on the one hand, the ALOS and SRTM DEM have

the same spatial resolution, which is high enough that the HRU surface characteristics are equally well represented as with the reference DEM. On the other hand, if results at the catchment scales are considered, in the coarser GTOPO30 setup, differences are leveled out. Moreover, as written in the answer to comment 21, the effect of altitude and enhanced radiation input is not as strong in winter, as in summer. Nonetheless, regarding the individual HRUs there are considerable differences as Figure 8 illustrates.

References

Freudiger, D., Kohn, I., Seibert, J., Stahl, K., and Weiler, M.: Snow redistribution for the hydrological modeling of alpine catchments, WIREs Water, 4, e1232, doi:10.1002/wat2.1232, 2017.

Grossi, G., Lendvai, A., Peretti, G., and Ranzi, R.: Snow Precipitation Measured by Gauges: Systematic Error Estimation and Data Series Correction in the Central Italian Alps, Water, 9, 461, doi:10.3390/w9070461, 2017.

Liston, G. E. and Elder, K.: A Meteorological Distribution System for High-Resolution Terrestrial Modeling (MicroMet), J. Hydrometeorol., 7, 217–234, doi:10.1175/JHM486.1, 2006.

Moriasi, D. N., Arnold, J. G., van Liew, M. W., Bingner, R. L., Harmel, R. D., and Veith, T. L.: Model Evaluation Guidelines for Systematic Quantification of Accuracy in Watershed Simulations, Transactions of the ASABE, 50, 885–900, doi:10.13031/2013.23153, 2007.

Rahman, K., Maringanti, C., Beniston, M., Widmer, F., Abbaspour, K., and Lehmann, A.: Streamflow Modeling in a Highly Managed Mountainous Glacier Watershed Using SWAT: The Upper Rhone River Watershed Case in Switzerland, Water Resour Manage, 27, 323–339, doi:10.1007/s11269-012-0188-9, 2013.

Weber, M., Feigl, M., Schulz, K., and Bernhardt, M.: On the Ability of LIDAR Snow Depth Measurements to Determine or Evaluate the HRU Discretization in a Land Surface Model, Hydrology, 7, 20, doi:10.3390/hydrology7020020, 2020.

WMO: Technical regulations: Basic documents no. 2, Volume I – General Meteorological Standards and Recommended Practices, 2010th ed., World Meteorological Organization, Geneva, 2011.

---

## Author Response (AR1)

**Response to Anonymous Referee #1**

This is a review of "The evaluation of the potential of global data products for snow hydrological modelling in ungauged high alpine catchments" by Weber et al. The paper investigates the impacts of using a series of climate data products to force a hydrological model and it's advanced snow modules and compare the behavior in terms of snow process representation and runoff. The authors also compare the impact of modifying the DEM resolution to investigate the impact of using coarser (but free) DEMS compared to more refined (but expensive) ones such as LiDAR. The study takes place on a small, 12km2 catchment in Bavaria near the summit of Zugspitze. The authors find that the choice of DEM is not as critical as first thought, and that the use of global climate products can yield reasonable results in hydrological modelling but that there is still improvements to be made. I have read the paper and found it very interesting and complete. The text flows generally well, although some expressions and sentences don't "sound" right and should be corrected by a native speaker. Scientifically, I have some issues with a few aspects of the work and I also have some suggestions to improve the work and make it more useful to the community. I will start by mentioning the more general points and end with smaller, more technical points.

**Author's answer**: We would like to thank the reviewer for carefully reading the manuscript and providing us with a very constructive and overall positive feedback. We have thoroughly considered all of his/her comments and address them point by point in the following.

**General comments:**

**1 -** The authors use a variety of climate data products to drive the hydrological model to simulate the snow accumulation and melt processes. There are two station datasets (local and from a somewhat distant but similar catchment), satellite products and the ERA20C product. I have a problem with the latter. It can be argued that the 0.05, and to some extent the 0.2 / 0.25 products can be "reasonable" in terms of spatial resolution to represent the 12km2 catchment. However, the 125km resolution ERA-20C has a resolution of 125x125 = 15625km2, or more than three orders of magnitude difference. The catchment represents less than 0.08% of the tile size. It seems unreasonable to me to include it in the analysis. I think no researcher would use this product for such a small catchment in real world applications. The authors talk about using ERA-20C because of the correction of Gao et al. 2012, but I am positive that using a product such as ERA5-Land (With a 0.1 resolution) would be a better proposition.

**Author's answer**: This is a very important comment and we are also convinced that showing results of the quite new ERA5-Land product will make our work more up to date. Thereafter, we absolutely agree with your suggestion and performed model runs with ERA5-Land (approx. 9 km) as well as ERA5 (approx. 31 km) data for the updated manuscript version. In our first response, we stated that we could obtain quite good simulation results with these data. Unfortunately, when we checked the ERA5 and ERA5-Land input data again, we found that they were wrong due to a rounding error in the latitude. Subsequently we ran the model again with the right input data. However, the results, we obtained are not satisfactory. Both data sets are too cold and too wet resulting in way to much snow with piling up to distinctive snow towers. We added statements for ERA5 and ERA5-Land in the following lines:

ll. 91-94: "To answer the first question, we investigated globally and publicly available meteorological driver data from the Climate Forecast System Reanalysis (CFSR), different versions of the Global Land Data Assimilation System (GLDAS), ERA-20C as well as ERA5 and ERA5-Land datasets and precipitation information from the Climate Hazards group Infrared Precipitation with Stations (CHIRPS)."

ll. 215-220: "Regarding the hybrid data, we forced CRHM with the well-known Climate Forecast System Reanalysis (CFSR) dataset pro-vided by the National Centers for Environmental Prediction (NCEP) (Saha et al., 2010). Moreover, we used NASA's Global Land Data Assimilation Systems (GLDAS) (Rodell et al., 2004), ERA-20C (Poli et al., 2016) and the new ERA5 (Hersbach et al., 2020) and ERA5-Land (Muñoz Sabater, 2019). CFSR, GLDAS, ERA-20C, ERA5 and ERA5-Land products combine in situ measurements, remote sensing and atmospheric modelling. ERA5 and ERA5-Land are the latest members of the ERA family and ERA5-Land has a relatively high spatial resolution with 9 km."

ll. 286-287: "The newer and higher resolved ERA5 and ERA5-Land setups are in a similar range with -1.8 °C and -2.5 °C."

l. 300: "Only precipitation data from ERA5 and CFSR are close to the reference data over the last 30 years."

ll. 308-309: "However, some curves, e.g. the ones of ERA5, ERA5-Land and DWD\_Wendelstein almost show a plateau from spring to summer in contrast to the reference."

ll. 487-488: "The same can be observed for the CFSR, ERA5 and ERA5-Land setups, but for those even at both measurement stations."

ll. 532-539: "Regarding the quite new and higher resolved ERA5 and ERA5-Land data, we were quite surprised that both data sets are not able to better capture the meteorological situation in the RCZ. In particular from the 9 km resolution ERA5-Land data, we expected better results. One reason for the bad performance is the direct vicinity of the RCZ to the surrounding much lower forelands. Therefore, even ERA5-Land might be too coarse to capture this situation. The comparably low pixel altitude of this product also supports this assumption. We assume that ERA5 and ERA5-Land could be applied with much more success in a central alpine catchment where the investigated catchment is surrounded by high-alpine regions, which might be more representative for the landscape domain."

ll. 628-631: "Another step in this direction is snow cover data from reanalysis products like ERA5 or ERA5-Land (Hersbach et al., 2020) that might provide information without modelling as well as for model evaluation or initialization purposes. However, its spatial resolution of 31 km and 9 km, respectively, might still be too coarse for heterogeneous high alpine terrain."

ll. 598-600: "To answer our first research question on the potential applicability of globally available meteorological data, we examined data from CFSR, different versions of GLDAS, CHIRPS, ERA5, ERA5-Land and ERA-20C including a specific downscaling approach, as well as a data transfer from another alpine station."

ll. 615-617: "The CFSR as well as ERA5 and ERA5-Land setups performed worst and produced so called 'snow towers' in all parts of the catchment due to too low temperatures, leading to the fact that the snowpack could not melt completely, even in summer."

**2 -** The GTOPO30 product seems to give reasonable results and the authors state this in several places in the paper. However, it seems that it performs well because it is biased and it is "counteracting" the bias of the meteorological products. Therefore it is better, but for the wrong reasons. I think it would be warranted to add a section (or sentence) in the discussion to clarify this to prevent readers from getting the wrong impression of the quality of GTOPO30. Again, I think users working on very small catchments with high gradients would never use such a coarse product, it was not designed for this.

**Author's answer**: We agree with you that the GTOPO30 product was not designed for applications in small and highly heterogeneous catchments. Nonetheless, such a product might be useful in large mountain domains and if computational power is limited. Moreover, we clarified why the GTOPO30 product performed so well at a first glance. The reasonable results derived with the GTOPO30 DEM are, however, just specious due to the coarse resolution, where small scale topographic effects are averaged out. This leads to the fact that the meteorological products are interpolated in a too coarse topographic description of the catchment, which might level out the meteorological data. Therefore, for some parts of the catchment the values are too high, while they are too low in other parts. Nevertheless, it might well capture the mean topographic/meteorological situation of the catchment as it is the case in our study. We added this to our discussion:

ll. 558-562: "However, the almost 100 m lower DEM-dependent average catchment altitude of GTOPO30 plays a minor role, since it is still high enough and temperatures are low enough that the snow cover can develop in the model as it was measured. Regarding the catchment scale, the differences among HRUs are averaged out. The GTOPO30 DEM was not created for applications in small basins with highly heterogeneous terrain and we generally would refrain from using it for such applications."

**3 -** I think the authors should have compared the snow modelling results they obtain with those from a reanalysis directly, such as ERA5. This could be a much simpler way than using reanalysis meteorological data to drive a hydrological model. It seems to me that this would be a much simpler alternative than using these convoluted methods? I think it could be appropriate to at least mention the possibility here as it fits the bill perfectly: using publicly available global datasets to model snow hydrological modelling in alpine catchments. Furthermore, it could be used to force the initial states of the hydrological model to simulate runoff.

**Author's answer**: Indeed, the quite new ERA5-Land reanalysis product includes many very interesting variables such as snow height and SWE as an average over the entire pixel size of the product. We also agree with your statement that integrating ERA5-Land snow would be an interesting addition. In this regard, we promised to make a comparison of the ERA5-Land SWE with specific snow hydrologic variables MSWE, DMSWE, snow cover duration, ablation shown in Table 6 for the entire RZC. However, we suggested this when we had the old, false

data as explained in our response to comment 1. As the right ERA5-Land data represents values of a pixel that is approx. 800 m lower such a comparison does not make sense in our opinion. This is supported by the temperature and precipitation data that are out of the climatological range. Therefore, we refrained from showing such a comparison. Nonetheless, we expect further developments regarding such data and stated that in Section 4.3 and specifically mentioned the ERA5 and ERA5-Land products.

ll. 586-589: "Another step in this direction is snow cover data from reanalysis products like ERA5 or ERA5-Land (Hersbach et al., 2020) that might provide information without modelling as well as for model evaluation or initialization purposes. However, its spatial resolution of 31 km and 9 km, respectively, might still be too coarse for heterogeneous high alpine terrain.

**4 -** I notice that there is no section on model calibration, as this model does not require calibration but is instead "parameterized" to the environment. I suggest adding this information as it is atypical for a model to not require calibration.

**Author's answer**: We followed your recommendation and added the following:

ll. 191-193: "Unlike most hydrological models, CRHM does not require calibration. The only parameters that we adjusted for modelling are the previously named catchment/HRU specific physiographic characteristics."

**Specific comments:**

**Lines 155-160**: Do I understand correctly that all precipitations were multiplied by 1.5? The SWE technically also includes the effects of ablation/sublimation/transport, so I think it is dangerous to correct precipitation in this manner as the actual real factor is probably different. Perhaps add some limitations in the text here.

**Author's answer**: We only corrected snow precipitation with this factor and agree with you that the real factor might be different for the named effects. However, the factor was determined in a way that minimizes these effects. To minimize the influence of sublimation, data with a 10 minute temporal resolution were used for the comparison of SWE and precipitation. Before the comparison, the effect of wind on the measured snow water equivalent was investigated. Therefore, time periods without snow fall, temperatures below freezing and strong wind were checked. This revealed no wind induced snow redistribution that could be captured by the snow gauge. Nevertheless, there are effects which could not be taken into account such as sublimation from the snowflakes during the snow fall event due to strong winds. As you suggested, we added a respective paragraph.

ll. 142-147: "Therefore, we used the factor of 1.5 to correct snow precipitation measured at the DWD station which is used for model forcing. The factor was determined using 10-minute values to minimize the effect of sublimation from the snow cover. However, the potential effect of snow redistribution by wind on this factor could not be detected by analysing the SWE in periods with temperatures below the freezing point, strong winds and without snowfall. Besides, it has to be noted that strong winds rarely occur without precipitation in RCZ, pointing to some limitations of this under catch factor determination."

**Lines 296-301**: This section is a bit confusing. Also, there are 2 peaks of runoff caused by snow accumulation periods? There are two in the year?

**Author's answer**: We agree with you that this section was a bit confusing and thus we rewrote it. We just wanted to hint at the fact, that the two in situ measured DWD station datasets (the reference and the Mt. Wendelstein setups) both have their precipitation maximum during the snow accumulation period in March. Furthermore, the reference has a second small peak in November. On the other hand, all other data sets have their maximum in summer.

ll. 302-305: "In general, not all precipitation regimes follow the same pattern. The reference and DWD_Wendelstein data sets follow a regime with predominant precipitation during the snow accumulation period. All other meteorological datasets reach their maximum precipitation in summer but also show a local maximum in March. The CHIRPS precipitation is similar to the reference, however, with a more pronounced maximum in August."

**Line 322**: missing year for reference "Danielson and Gesch".

**Author's answer**: We added the year.

**Line 336**: "Adjusted it to the ALOS…:" : Should this be altitude-corrected? Please clarify

**Author's answer**: We clarified the sentence. ll. 325-326: "…adjusted it to the ALOS, SRTM and GTOPO30 DEM specific HRU altitude, slope and aspect (Table 2) as explained in Section 2.3"

**Line 378**: "shorter" should be "less"

**Author's answer**: We corrected that.

**Line 394:** "snow towers" needs to be defined better.

**Author's answer**: We added the following. ll. 383-386: "Snow towers are an effect in snow hydrological modelling that occurs mainly at higher altitudes and describes the unrealistically high accumulation of snow over several years. Reasons can be the insufficient description of redistribution processes in the model or unrealistic meteorological forcing data (Freudiger et al., 2017) as the latter will be true in our case."

**Line 413-422:** This section is not clear upon reading. I needed to read more of the paper before coming back and understanding this section. Please simplfy and/or clarify.

**Author's answer**: We made some changes to the text to clarify this section. Ll. 394-403: "Regarding the snow depth results simulated by using the model setups with different topographic characteristics on basis of the 2.5 m DEM for the reference run and the three different globally and publicly available DEMs ALOS, SRTM and GTOPO30, the snow depth development was simulated realistically at both measurement sites (Fig. 7g, h, Table 5). This results in R2valuesabove 0.8, NSE values above 0.77 and MAE values below 0.33 for all three global data setups at the LWD site being close to the statistical values of the reference. For GTOPO30, the quality measures for the DWD site are even slightly better, whereas the ALOS and SRTM setups show slightly weaker NSE, R2and MAE values at the DWD site (Table 5). The two 30 m DEMs SRTM and ALOS show slight differences in the NSE values at the measurement stations, which can be attributed to differences in the sensing angles and post processing algorithms leading to topographic differences and as a consequence also to differences especially in Qsi as mentioned in Section 3.2."

**Response to Anonymous Referee #2**

Review of "The evaluation of the potential of global data products for snow hydrological modelling in ungauged high alpine catchments" Weber et al.

**General overview**

The authors present an evaluation of snowpack and hydrological modeling outcomes in a well-characterized and gauged catchment using parameterizations derived from regional and global datasets. They also evaluate the influence of three different global digital elevation models on the derivation of model inputs such as slope, aspect, and solar insolation as well as the impact on model results. The goal was to illustrate and quantify the impacts of using these products to estimate snowpack and runoff in ungauged catchments in snow-dominated mountainous regions.

The purpose and need for the study are reasonably clear and such work is important for the advancement of snow and mountain hydrology, in general. While it appears that the work and results are technically sound, I found the manuscript very difficult to follow and as such am unable to fully evaluate their results. I recommend a major revision. Specifically, I recommend: …

**Author's answer:** We would like to thank the reviewer for taking his/her time to read the manuscript and providing us with a very constructive feedback. Furthermore, we thank him/her for mentioning the importance of this work regarding advancements in snow/mountain hydrology. We have thoroughly considered all of his/her comments and address them point by point in the following.

**General comments:**

**1 –** I recommend reorganizing the methods and results to more concisely lay out the study (methods then results), including the use of additional tables or figures as recommended in the "Specific comments" section

**Author's answer:** We followed your recommendation and restructured the manuscript. Please see also our answers to the respective points to the specific comments.

**2 –** I recommend rewriting the manuscript with an eye toward brevity and clarity, including working with an English speaking editor. The manuscript is overly long, repetitive, and full of awkward and difficult to follow sentences.

**Author's answer**: We followed you recommendation and restructured the paper. We also took care to avoid repetitions and gave the paper to a native speakers for prove reading.

**Specific comments:**

**Introduction**: At some point in the introduction, there needs to be a clear statement or bulleted list of the research questions. These questions should then guide the organization of the rest of the paper.

**Author's answer**: In the last paragraph of the introduction (ll. 102-114), we now present the structure of the paper, as we agree that this improves readability. Moreover, we formulated our two main goals as research questions with bullet points in the introduction, regarding the overall question on how far it is possible to use globally available input data for snow-hydrologic modelling in ungauged high alpine catchments. ll. 87-90:

- "Is it possible to use globally available meteorological data products to reliably simulate snow depth, specific snow hydrological parameters and runoff in high alpine catchments?
- What impact have DEM products with different spatial resolutions on snow hydrological simulations in complex topography?"

**Section 2, 3, 4**: These appear to be all methods, yet there are partial results mixed in. It would be much clearer, if it were a single methods section with subsections on the catchment, the model method, and the input

datasets. Then the first subsection under results could compare the inputs of the various datasets, followed by the rest of Section 5. Otherwise, it is very difficult to follow.

**Author's answer**: As noted in the general comments, we restructured the manuscript according to your suggestions. The method section now contains the catchment description, the model description and simulation methods, as well as the input datasets. The former Sections 4.2 and 4.4 were moved as subsections to the results section. The structure now is

1    Introduction
2    Methods
2.1  The Research Catchment Zugspitze
2.2  Structure of the Cold Regions Hydrological Model
2.3  Meteorological input data and their preparation for model input
2.4  Applied DEMs to describe the land surface characteristics
3    Results
3.1  Comparison of the different meteorological input data sets
3.2  Influence of the DEMs and associated land-surface characteristics on meteorological conditions
3.3  Evaluation of simulated snow depth with measured data
3.4  Comparison of snow hydrologically relevant indices and the simulated runoff regime
4    Discussion
4.1  Application of global meteorological data
4.2  Application of global DEM products
4.3  General remarks and potential future developments
5    Conclusions

**Page 5, line 138**: The term "gradient" should be replaced with "relief". "Relief" refers to the absolute difference in elevation of a region, whereas "gradient" refers to the slope.

**Author's answer**: Thank you for pointing out this linguistic subtlety to us. We changed "gradient" to "relief".

**Page 5, line 142**: The term "knee wood" should be replaced by "krummholz", which is the more internationally recognized term for dwarf woody alpine vegetation.

**Author's answer**: We replaced "knee wood" by "krummholz".

**Introduction, lines 112-124**: Much of this should be moved to the methods section as it is repetitive.

**Author's answer**: We shortened this paragraph. Some aspects are still mentioned in ll. 102-114. The definition end explanation of HRUs is now in Section 2.2 ll. 153 ff.

**Page 5, line 160-199**: A figure or table would greatly help the reader understand the model structure and underlying component calculations.

**Author's answer**:  CHRM is a widely used model for investigating snow hydrological questions. The modules and underlying component calculations are described in detail in Pomeroy et al. (2007) and the other papers cited in the paragraph. We included a reference and pointed out more clearly the structure and description of the model in the updated version. In addition, a very general structure description for the application of the model in the RCZ is presented in Weber et al. (2016) – we included the reference to this paper in this context as well. The paragraph can be found in ll. 175 ff.

**Page 8, line 201-202**: Why is the reference referred to as representing the "gauged basin mode"? It would help if the methods section started out with statement about how the study is organized, such as "The approach we use is to model snow and runoff in the well-characterized RCZ using the CRHM and in-situ measurements. Then we repeat the modeling process using alternative datasets. In order to do this, we first describe the watershed, the model, the derivation of the datasets…." Moreover, the derivation of the reference model needs to be more clearly explained. Are you using a single meteorological station's data (Mt. Zugspitze DWD station) to drive the catchment model and compare the results at two other stations in the catchment (LWD, DWD)?

Use of standard three-letter station abbreviations for all stations in the study would help clarify this, too.

**Author's answer**: We followed your suggestion to shortly outline the method steps undertaken at the beginning of the methods section, which should now increase readability (ll. 117-118). The reference setup is referred to as 'gauged basin mode' since it uses in situ measured driver data directly measured within the gauged RCZ.

Indeed, we are using data from a single meteorological station. We pointed this out more clearly in the updated version along with a better description the reference data (ll. 195-196). The reason why we 'just' used the DWD dataset as meteorological input is now clarified in more detail in the manuscript: For the reference setup, we used meteorological data measured at the DWD station on Mt. Zugspitze since it is the best dataset in terms of continuity and data quality measured directly within the catchment. Due to some longer data gaps, especially at the end of the winter season and the summer time, we did not use the meteorological LWD data as driver data. We will add in that besides the gaps in summer, there are also some gaps during winter, mainly at the end of season (ll. 196-198).

In general, we agree that three-letter abbreviations for all stations would be good. For the two in situ stations LWD and DWD in the RCZ we used three-letter abbreviations. To avoid confusion, we named the DWD station situated at Mt. Wendelstein 'DWD_Wendelstein'. We think that for this study it is ok to extend the latter station name, as we also do not use three letters to describe the results of the global meteorological and DEM datasets, which have in general longer abbreviations and whereof it would not make sense to shorten them.

**Page 10, line 257**: What does PUB stand for?

**Author's answer**: The acronym PUB stands for predictions in ungauged basins and is defined in l. 67.

**Figures 3 and 4**: These figures are introduced in the text and appear before section 4.3 which describes the DEM parameterizations. It was unclear, at first, why the DEM datasets were part of the graphs. It would be clearer to present methods in one full section and results in the next. As it is, some results are mixed in with each method section.

**Author's answer**: We followed your recommendation and restructured the manuscript to clearly distinguish between methods and results. Regarding the two graphs you mentioned (Fig. 3 & 4) we think that in order to exceed a not too high number of graphs in total, and not to be repetitive that adding the information related to the DEMs is definitely ok. However, we mentioned more clearly in the caption that the graphs contain information of the meteorological and DEM setups.

**Page 18, line 432**: What does 'thunderstruck' mean? Do you mean that there are data gaps to power outages caused by lightning strikes?

**Author's answer**: We changed it to lightning strikes.

References

Pomeroy, J. W., Gray, D. M., Brown, T., Hedstrom, N. R., Quinton, W. L., Granger, R. J., and Carey, S. K.: The cold regions hydrological model: a platform for basing process representation and model structure on physical evidence, Hydrol. Process., 21, 2650–2667, doi:10.1002/hyp.6787, 2007.

Weber, M., Bernhardt, M., Pomeroy, J. W., Fang, X., Härer, S., and Schulz, K.: Description of current and future snow processes in a small basin in the Bavarian Alps, Environ Earth Sci, 75, 962, doi:10.1007/s12665-016-6027-1, 2016.

**Response to Anonymous Referee #3**

Comments on the manuscript: "The evaluation of the potential of global data products for snow hydrological modeling in ungauged high alpine catchments" by Weber et al.

This manuscript examines the sensitivity of the simulated snow characteristics in a small alpine basin related to meteorological input data and DEMs using a hydrological model with a sophisticated snow module. Studies like this are of a great importance as high elevation snowpack critically affects alpine ecosystems and water resources in a number of regions in the world. The experiment is comprehensively designed and well executed. The paper is informative and interesting.

**Author's answer:** We would like to thank the reviewer for thoroughly reading the manuscript and for providing us with valuable suggestions for improvement. Moreover, we thank him/her for mentioning the importance of this work regarding improvements in the investigation of high alpine water resources which is relevant for numerous (ungauged) catchments worldwide. We have carefully considered all of his/her comments and address them point by point in the following.

**General comments:**

**(1)** The analysis of the experimental results and the organization as well as the presentation of the analysis output need improvements,

**Author's answer**: In general, we reorganized the paper structure, which was also mentioned as an issue by reviewer 2 and we better streamlined and improved the presentation and analysis of the results as well as the discussion. We also had an eye on avoiding repetitions. As you make some clear suggestions on these points in the specific comments below, please also consider our answers to those points in the following. The general structure of the paper now is:

| | |
|---|---|
| 1 | Introduction |
| 2 | Methods |
| 2.1 | The Research Catchment Zugspitze |
| 2.2 | Structure of the Cold Regions Hydrological Model |
| 2.3 | Meteorological input data and their preparation for model input |
| 2.4 | Applied DEMs to describe the land surface parameters |
| 3 | Results |
| 3.1 | Comparison of the different meteorological input data sets |
| 3.2 | Influence of the DEMs and associated land surface parameters on meteorological conditions |
| 3.3 | Evaluation of simulated snow depth with measured data |
| 3.4 | Comparison of snow hydrologically relevant indices and the simulated runoff regime |
| 4 | Discussion |
| 4.1 | Application of global meteorological data |
| 4.2 | Application of global DEM products |
| 4.3 | General remarks and potential future developments |
| 5 | Conclusions |

**(2)** Insufficient evaluation of the reference simulation across entire HRUs is also problematic. Because of the large elevation range within the test basin (RCZ), the snow field within the basin is expected to vary widely. Without the evaluation of the elevation dependent model performance with the reference data, the accuracy of the reference run cannot be well established.

**Author's answer**: We agree with you that the snow cover in the basin can vary widely as it is the case in any high alpine region with complex topography and that it would be valuable if it was possible to evaluate the accuracy of the elevation dependent snow cover variation in more detail.

We discussed this point in more detail in the updated version and added the paragraph in ll. 486-491: "It could be argued that the evaluation of the simulation results on an HRU basis is insufficient due to the large heterogeneity of the catchment. However, the chosen HRU delineation partly accounts for this heterogeneity since it directly relies on the LIDAR measured dominant snow depth distribution (Weber et al., 2020). Moreover, dominant snow depth patterns in high alpine regions are largely persistent over the years as e.g.

Grünewald et al. (2013) show for various high alpine catchments. We are thereafter confident that the spatial distribution of the snow cover is quite well simulated."

Moreover, we think that the availability of two snow depth gauges in such a small catchment is already a lot and we are quite lucky with the situation in this well gauged high alpine catchment. Of course, it would be nice if more or even each of the ten HRUs would be represented with a snow depth gauge; however, this is not realistic and will most probably not be found in such a potential station density anywhere in the world. Thereafter, the only possible accuracy analysis for this study period is the investigation of snow depth with data measured at the DWD and LWD station which are situated at least at different altitudes. Moreover the locations of these stations were chosen from the two operating institutions DWD and LWD to be as representative as possible for the respective altitude, e.g. no shadowing against wind but also no particular exposure, relatively flat terrain, etc.

**(3)** The lack of the analysis and evaluation on monthly time scales (i.e., annual-cycle resolving analysis). This is important because the forcing and snow fields undergo strong seasonal cycle, hence, seasonal cycle-resolving analysis & evaluation can be useful in assessing key sources of the simulation errors. This will also help to link the seasonal cycle of the runoff to monthly snow ablation.

**Author's answer**: In addition to the already presented mean monthly precipitation (Fig. 4) and the mean monthly runoff (Fig. 8), we present the mean monthly SWE in Fig. 8 and the mean monthly $Q_{si}$ in Fig. 6 in the revised version of the manuscript.

Regarding $Q_{si}$, we wrote in ll. 307-310: "The 10-year monthly mean of incoming shortwave radiation ($Q_{si}$) for the RCZ, relevant for snow cover depletion, is presented in Fig. 6. All setups show very similar data. However, some curves, e.g. the ones of ERA5, ERA5-Land and DWD_Wendelstein almost show a plateau from spring to summer in contrast to the reference. A possible reason might be that convective clouds form during this period while Mt. Zugspitze is frequently above the condensation level."

Among the setups that showed plausible results, the accumulated amount of snow, which is a result of the precipitation regime, is the most important factor for the magnitude of runoff. Moreover, it also determines the period in which snow is available for melt and thus the occurrence of the peak runoff. Regarding the topographically different setups, the individual radiation budget of the HRU influences the timing and magnitude of runoff. We discussed this in ll. 450-469.
The precipitation regimes presented in Fig. 4 show strong differences. The reference's regime has its peak precipitation during the snow accumulation period whereas others, like ERA5, have their peak precipitation during summer. We presented this in ll. 300-306.

**(4)** The terminology "topography parameterization" is confusing. The true meaning of "topography parameterization" in the manuscript is "characteristics of topographic parameters" such as slope and azimuth that vary according to the DEM resolutions. "Parameterization" typically means representing unresolved properties using resolved values or representing a property using a related variable(s).

**Author's answer**: You are right, this was misleading. We changed the wording in the text where necessary.

**(5)** The writing is OK, but I found occasional awkward/unfamiliar sentences. As a speaker of American English as the second language, I don't like to suggest any specific changes in grammar and writing. I strongly recommend the authors to consult native English speakers to go through the entire writing.

**Author's answer**: We gave the manuscript to a native speaker for prove reading.

**Specific comments:**

**(1)** The authors present results from elaborate model runs and analyses, but the key messages are not clearly presented and are sometimes confusing.

**Author's answer**: We revised the entire paper and streamlined the text so that the key messages become clear. In the introduction, we added our two main research questions as bullet points, which are both dealing with the overall question of how far it is possible to use globally available input data for snow-hydrologic modelling in ungauged high alpine catchments.

ll. 87-90:

- "Is it possible to use globally available meteorological data products to reliably simulate snow depth, specific snow hydrological parameters and runoff in high alpine catchments?
- What impact have DEM products with different spatial resolutions on snow hydrological simulations in complex topography?"

The answers to our questions and thus our key messages are now clearly presented in the abstract (see comment below) and the conclusions part.

ll. 625-636: "Based on our results, the answer to our first research question is that it is not possible to exclusively use the tested global meteorological data products to reliably simulate snow depth and further snow hydrological parameters as well as runoff in the high alpine RCZ. One reason is that up to now, these global products are neither able to describe the meteorological heterogeneity of such complex catchments with steep terrain, nor its average conditions. This is reflected in a range of 3.5°C in the catchment mean decadal temperature and 1510 mm in the catchment mean decadal annual precipitation sum over all input data for the RCZ. However, we assume that results could be different if the investigated catchment was not a topographic outlier in the landscape region as it is the case with RCZ with its adjacent lower forelands. The answer to our second question is that compared to the influence of the different meteorological forcings on simulated snow hydrological parameters, the influence due to different characteristics of topographic parameters like slope, aspect and altitude due to different DEM products is smaller, even in complex terrain. Nonetheless, there are considerable differences mainly due to DEM product dependent variations in the radiation balance and due to mean HRU altitude induced variations in temperature and precipitation."

Moreover, we summarized in a few sentences the more specific key findings of the results part in each of the four sub-sections regarding

- the comparison of the different meteorological input data sets (see ll. 311-317 (Section 3.1)),
- the influence of the DEMs and associated land surface parameters on meteorological conditions (see ll. 345-349 (Section 3.2)),
- the evaluation of simulated snow depth with measured data (ll. 404-412 (Section 3.3)),
- and the comparison of snow hydrologically relevant indices and the simulated runoff regime (ll. 492-496 (Section 3.4)).

At the end of the conclusions section, we gave the following outlook (see ll. 636-641): "Despite the weak performance of the global meteorological products, we assume that they might produce better results if the analyzed catchment is more representative for the surrounding larger scale landscape region. Furthermore, we expect a growing importance of such data in future snow hydrological modelling, also in ungauged basins, due to their constant and rapid evolution including temporal and spatial refinements. In order to generalize the findings from our study and to intensively test newly developed meteorological and snow hydrological products, we suggest to conduct further investigations in the well monitored catchments from INARCH."

**(2)** Please improve Abstract so that the key findings in the experiment are presented more concisely clearly. Separating the sensitivity to DEMs and associated orographic parameters from the sensitivity to the forcing data may help organizing with more clarity.

**Author's answer**: We improved the abstract according to your suggestions. Key questions and the main findings are now clearly pointed out:

Abstract: "For many ungauged mountain regions, global datasets of different meteorological and land surface parameters are the only data sources available. However, their applicability in modelling high alpine regions has been insufficiently investigated so far. Therefore, we tested a suite of globally available datasets by applying the physically-based Cold Regions Hydrological Model (CRHM) for a 10-year period in the gauged high alpine Research Catchment Zugspitze (RCZ), which is 12 km² and located in the European Alps. Besides meteorological data, snow depth is measured at two stations. We ran CRHM with a reference run with in situ measured meteorological data and a 2.5 m high-resolution digital elevation model (DEM) for the parameterization of the surface characteristics. Regarding different meteorological setups, we used ten different globally available datasets (including versions of ERA, GLDAS, CFSR, CHIRPS) and additionally one transferred dataset from a similar station in the vicinity. Regarding the different DEMs, we used ALOS and SRTM (both 30 m) as well as GTOPO30 (1 km). The following two main goals were investigated: a) the reliability of simulations of snow depth, specific snow hydrological parameters and runoff with global meteorological products and b) the influence of different global DEMs on snow hydrological simulations in such a topographically complex terrain. The range between all setups in mean decadal temperature is high at 3.5°C

and for the mean decadal precipitation sum at 1510 mm, which subsequently leads to large offsets in the snow hydrological results. Only three meteorological setups, the reference, the transferred in situ dataset and the CHIRPS dataset, substituting precipitation only, showed agreeable results when comparing modelled to measured snow depth. Nevertheless, those setups showed obvious differences in the catchment's runoff regime and in snow depth, snow cover, ablation period, the date and quantity of maximum snow water equivalent in the entire catchment and in specific parts. All other globally available meteorological datasets performed worse. In contrast, all globally available DEM setups reproduced snow depth, the snow hydrological parameters and runoff quite well. Differences occurred mainly due to differences in radiation model input due to different spatial realizations. Even though SRTM and ALOS have the same spatial resolution, they showed considerable differences due to their different product origin. Despite the fact that the very coarse GTOPO30 DEM performed relatively well on the catchment mean, we advise against using this product in such heterogeneous high alpine terrain since small-scale topographic characteristics cannot be captured. While global meteorological data is not suitable for sound snow hydrological modelling in the RCZ, the choice of the DEM with resolutions in decameter-level is less critical. Nevertheless, global meteorological data can be a valuable source to substitute single missing variables. For the future, however, we expect an increasing role of global data in modelling ungauged high alpine basins due to further product improvements, spatial refinements and further steps regarding assimilation with remote sensing data."

**(3)** The statement in Abstract, L23:24, contradicts the statement in Conclusion L625-634.

**Author's answer**: We rewrote the entire abstract including this statement to avoid contradiction.

**(4)** L149:151: How the mean snow field (e.g., SWE, SCA) over the entire RCZ is calculated? This is among the key evaluation variables.

**Author's answer**: CRHM calculates for each HRU and each time step values of SWE and snow depth. For these calculations, CRHM requires a meteorological input for each HRU. Since such data is not available for each HRU, we used the method developed by Liston and Elder (2006) to generate it for each HRU. We use the modelled values of SWE and snow depth to calculate the area weighted mean value for the entire RCZ. We are confident, that the chosen HRU delineation is suitable for realistic snow cover simulations since it is the same as in Weber et al. (2020) who established these HRUs and evaluated them with spatially distributed LIDAR snow depth measurements. We described this better in the revised manuscript (ll. 155-167) and hope that this is the answer to your question since L149:151 do not deal with the calculation of the mean snow field over the entire RCZ. The snow indices as they are analyzed in Section 3.4 are HRU-area weighted catchment means.

**(5)** L156:157: The met data at LWD are not used as the forcing. Why the snow precipitation at LWD is corrected for undercatch?

**Author's answer**: The meteorological data at the LWD station, as already stated in the manuscript, has some longer data gaps (especially at the end of the winter seasons and during summer time in the study period of 2000-2010), which is why we refrained from using it as meteorological forcing data. Since 2014, a snow scale was installed at the LWD station, which enabled us to use measured SWE for more recent years as a good possibility to investigate the precipitation undercatch. Similar to the literature reported undercatch of snow precipitation of up to 50% (WMO, 2011; Grossi et al., 2017), we found out in our former study (Weber et al., 2020) that we also can expect an undercatch of 50% for the RCZ. Assuming that undercatch variations in precipitation over the entire catchment are rather negligible, the DWD snow precipitation has been corrected with the factor derived at the LWD station.

We clarified this in the revised manuscript. ll. 142-147: "Therefore, we used the factor of 1.5 to correct snow precipitation measured at the DWD station which is used for model forcing. The factor was determined using 10-minute values to minimize the effect of sublimation from the snow cover. However, the potential effect of snow redistribution by wind on this factor could not be detected by analysing the SWE in periods with temperatures below the freezing point, strong winds and without snowfall. Besides, it has to be noted that strong winds rarely occur without precipitation in RCZ, pointing to some limitations of this under catch factor determination."

**(6)** Spell out HRU at its first appearance.

**Author's answer**: We spelled it out at its first appearance

**(7)** L227: Check the spatial resolution of the CFSv2 data. The finest resolution I could find is T382 that correspond to approximately 0.313degree.

**Author's answer**: We checked the spatial resolution again and thank you for the hint! In course of this we realized that we misunderstood the data description of the database where we downloaded the data. As you wrote, the spatial resolution of CFSv2 is approx. 0.313 degree. However, CFSv2 only exists from 2011 onwards. All data before and thus the data we used is data from the Climate Forecast System Reanalysis (CFSR) which has a 0.2 degree resolution. We changed this throughout the manuscript and renamed this setup CFSR.

**(8)** L265: Section 3 → Section 4

**Author's answer**: We now refer to the right section.

**(9)** L288-291: I don't understand what "contrary development" indicates. This sentence is ambiguous. Please provide more explanations.

**Author's answer**: We referred to the wrong figure and corrected that. It is Figure 3b instead of 3a. This shows that the precipitation situation in the reference and at the DWD_Wendelstein setup is contrary over the years 2005 – 2010.

**(10)** L293-295: This sentence is ambiguous. Please rewrite.

**Author's answer**: We rewrote the paragraph. ll. 302-306: "In general, not all precipitation regimes follow the same pattern. The reference and DWD_Wendelstein data sets follow a regime with predominant precipitation during the snow accumulation period. All other meteorological datasets reach their maximum precipitation in summer but also show a local maximum in March. The CHIRPS precipitation is similar to the reference, however, with a more pronounced maximum in August. As we focus on the seasonal snow cover, we assume that precipitation amount and regime is of major importance for snow cover simulations."

**(11)** L288:289: This sentence cannot explain the peak in March.

**Author's answer**: We guess you mean L.298-298 and thank you for pointing this out. Indeed, this sentence made no sense and was deleted.

**(12)** May change the title of Section 4.3. I suggest "Landsurface parameterization on basis of DEMs" → "Sensitivity of the land-surface parameters to DEMs (or DEM resolutions)"

**Author's answer**: We changed the title of the respective section to "Applied DEMs to describe the land surface parameters" since we think this fits even better to the content.

**(13)** Also suggest a new title for Section 4.4: "Influence of the DEMs and associated land-surface parameters on meteorological conditions"

**Author's answer**: We changed the title of the respective section according to your suggestion.

**(14)** L339: 100 m → 200 m (195 m). (200m elevation difference also corresponds to a lapse rate of 5K/km, slightly more stable than the standard atmosphere which is understandable over a cold surface like snow/ice.)

**Author's answer**: The difference in catchment mean altitude is 105 m which is roughly 100 m. We wrote the exact value (l. 327) instead of roughly 100 m to avoid confusion and also added the catchment mean values in Table 2.

**(15)** Section 5: The evaluation based only on NSE and R2 is insufficient. Need more metrics, at least the 'mean bias' and RMSE. Also provide additional evaluations for each month (i.e., annual-cycle resolving model evaluations)

**Author's answer**: We additionally provided the mean average error (MAE) in Table 5. For the analysis of the results in Section 3.3, we fully took this statistical measure into account (see ll. 357-404).

**(16)** Section 5.1: If there are problems in evaluating the snow simulations at DWD and DLW as stated in Section 5.2, how can you justify evaluating the daily snowdepth against the observations at these sites?

**Author's answer**: In Section 3.4 (formerly 5.2) we stated that we do not validate the snow cover duration and the number of ablation days with measured values. This is mainly due to data gaps at the LWD station in spring, which makes it impossible to determine the melt out day as well as the exact length of snow cover duration for some years. In addition, at the DWD station, the melt out day may be delayed due to the fact that the snow stake is installed directly on the small glacier 'Nördlicher Schneeferner'. This might bias the definition of snow cover duration and ablation days.

Nonetheless, we can very well quantitatively validate the daily modelled snow depth with the measurements for the entire season when data is available (this is true for most days except the days, where we had data gaps at the LWD station at the end of the season). Of course, there might be years regarding the DWD station, in which the model is not able to capture the exact melt out date of the snow cover due to the mentioned effect (ice is conserving the snow cover). However this can almost be neglected in a day based comparison since this snow cover is usually very thin and hardly affects the total seasonally accumulated snow volume. We clarified the above mentioned points in more detail in the updated version.

Ll. 362-366: "The ice has a cooling effect on the snow cover which leads to a delayed melt out of a thin snow cover compared to a faster melt of snow directly on top of rocks. This effect is particularly strong in spring and autumn, and is not considered in the model. However, this effect is negligible in case larger amounts of snow are piled up on top of the glacier, as it is the case for most of the time during the winter season. Thus, modelled snow cover at the DWD station lasts on average 20 days less than the measured one."

**(17)** L395: "snow towers pile up" → Is this due to the forcing errors or model errors or combined focing-model errors?

**Author's answer**: This is a well-known effect, which might occur if the meteorological input is not realistic, which is the case for some input setups we showed. The model structure, however, is always the same in each of our setups and such snow towers could never be observed when forcing the model with in situ measured meteorological data. The reason for this snow towers is mentioned in the following sentences in ll. 386-387: "These simulated snow towers are caused by much lower temperatures and much higher precipitation compared to the reference (Section 3.1)."

To clarify this in more detail we also added the following definition in ll 383-386: "Snow towers are an effect in snow hydrological modelling that occurs mainly at higher altitudes and describes the unrealistically high accumulation of snow over several years. Reasons can be the insufficient description of redistribution processes in the model or unrealistic meteorological forcing data (Freudiger et al., 2017) as the latter will be true in our case."

**(18)** L395: 'downscaled temperature' → 'temperature downscaling'

**Author's answer**: We guess the reviewer means L399. We changed it.

**(19)** L406:421: This paragraph repeats the statements in the previous paragraph. May be removed and present a summary of this paragraph in Conclusion.

**Author's answer**: We removed this paragraph and we revised the manuscript with an eye on deleting repetitions in general.

**(20)** Section 5.2

A)   Statements in this section is too qualitative without solid supports from acceptable level of evaluations.

**Author's answer**: We rewrote parts of this section (now Section 3.4) including more 'hard' numbers.

See ll. 430-437: "For the catchment mean and varying meteorological products, Table 6 shows an overestimation of MSWE in the CHRIPS and DWD_Wendelstein setup of 15% and 20%, respectively, compared to the reference setup. This overestimation is is related to the 378 mm and 534 mm higher mean annual precipitation sum in the CHIRPS and the DWD_Wendelstein setups. For the DWD_Wendelstein setup the combination with the lower mean temperature leads to a 12 days later occurrence of the DMSWE, a 22 days longer snow cover duration and a two weeks longer ablation period on average for the entire catchment. In contrast, the later DMSWE, the longer snow cover duration and the longer melting period of the CHIRPS setup is solely due to higher winter precipitation and the resulting higher MSWE. As a consequence, the higher MSWE

values in these two meteorologically different setups result in a higher runoff of 29 and 30%, respectively (Table 6)."

See also ll. 441-448: "Regarding the simulated snow hydrological results using the setups with different DEMs, the mean catchment MSWE corresponding to the GTOPO30 DEM is closest to the reference with a negligible deviation of 1% (Table 6). This good performance goes in line with the comparison of the measured and modelled snow depth data in Section 3.3 (Table 5). Both 30 m resolution DEMs result in a considerably higher catchment mean MSWE than the reference setup at +18% in the ALOS setup and +17%in the SRTM setup. The GTOPO30 setup shows for the DMSWE and the length of the snow cover as well as the ablation periods also the smallest deviations compared to the reference setup. The mean catchment DMSWE of the ALOS and SRTM setups occur eleven and 12 days later, respectively, when compared with the reference setup. Moreover, the snow cover duration in these setups is almost one month longer, which is reflected in their higher number of ablation days, too."

B) If missing data and site characteristics at LWD and DWD, respectively, prevent using these data for evaluation, how can we trust the reference data are accurate enough for evaluating model data?

**Author's answer**: This concern is reasonable, probably for all measurement stations worldwide. In the following, we provide you some specific information to the stations we used.

DWD station: this data is quality controlled by the DWD, the German Weather Service. Measurement errors are filtered and there are virtually no gaps.

LWD station: The instruments of this station are regularly maintained by the LWD, the Bavarian Avalanche Warning Service, since they rely on good quality data. All, the large gaps at this station used to occur in late spring when the avalanche season was over and the LWD no longer produces an avalanche warning bulletin. So they simply had no need for data in this time of the year. Consequently, the data recorded at that location is very accurate and reliable.

Regarding the locations of the two measurement stations, both institutions chose locations, which are representative for the respective altitude and general site conditions. Further restrictions particular concerning the use of DWD data for snow cover duration evaluations are explained in the answer to your comment 16.

C) If DMSWE cannot be validated, how can MSWE can be validated? MSWE is supposed to occur on DMSWE.

**Author's answer**: We never validated the MSWE and never stated that in our manuscript. We do not have measured SWE in the investigated time period. We only have measured snow depth at the LWD and DWD stations to validate our model runs. In our analysis in Section 3.4 (former 5.2) we compared all SWE related indices resulting from the three best meteorological setups and all DEM setups to the reference run that was forced with in situ measured driver data and parameterized with the high resolution 2.5 m DEM.

**(21)** L457:470: How can the data of largest warm bias, precipitation underestimation and insolation overestimation perform best in simulating the amount and timing of runoff? This may indicate major flaw in the model physics and/or forcing combinations. Need further discussions. This also indicates the need for a more rigorous evaluation of the reference data and other model data.

**Author's answer**: We rewrote the respective passage in the manuscript to clarify:

ll. 452-465: "The fact that the snow cover indices of the GTOPO30 setup are so close to the reference entails a very similar runoff behavior despite it has the largest warm bias, precipitation underestimation and insolation overestimation (Table 4). To explain this, the differences in the process of discharge formation have to be examined. Although the GTOPO30 DEM is lower in altitude than the reference DEM, it is still high enough that monthly mean temperature is below freezing until March and only at 0.1 °C for April while it is at -0.7 °C April in the reference. Both, the reference and GTOPO30 setup are clearly above freezing with 4.2 °C and 5.1 °C, respectively, in May. The main melt, which is temperature induced in spring starts almost at the same time in both setups illustrated by the SWE decline in Fig. 8. Melt induced runoff starts slightly earlier in the reference setup (Fig. 8), because there are HRUs which receive considerably more Qsi than in the GTOPO30 setup (Fig. 4). Nonetheless, the major part of the RCZ receives more Qsi in the GTOPO30 setup. The longer the days last in spring, the stronger becomes the radiation induced melt effect which is why in the results of the GTOPO30 setup, melt induced runoff becomes stronger in June than in the reference setup. In summer the lower elevation has a greater effect on the temperature than in winter which leads to higher melt induced runoff in the GTOPO30 setup in June and July. This is illustrated in the faster decline of SWE in June and July in Fig. 8. This effect decreases in August since less snow is left to melt."

**(22)** L475: Monthly Qsi needs to be evaluated as it is directly involved in snowmelt.

**Author's answer:** As we explained in our answer to comment 3, we added Fig. 6 to present the mean monthly Qsi for the RCZ. Moreover, we added in ll. 307-310 the following: "The 10-year monthly mean of incoming shortwave radiation (Qsi) for the RCZ, relevant for snow cover depletion, is presented in Fig. 6. All setups show very similar data. However, some curves, e.g. the ones of ERA5, ERA5-Land and DWD_Wendelstein almost show a plateau from spring to summer in contrast to the reference. A possible reason are convective clouds that form during this period while Mt. Zugspitze frequently is above the condensation level."

Among the setups that show plausible results, the accumulated amount of snow, which is a result of the precipitation amount and regime, is the most important factor for the magnitude of runoff. Moreover, it also determines the period in which snow is available for melt and thus the occurrence of the peak runoff.

**(23)** L506:507: This may be an overstatement for the CHRPS data. CHRPS yields good results for snow cover and runoff, but not in NSE and R2 of the daily snow depth.

**Author's answer**: The NSE and $R²$ of the CHIRPS setup for daily snow depth is clearly the best among all presented globally available meteorologically different setups. Despite the quality measures are lower than for the reference, they are still considerably high. Studies from Moriasi et al. (2007) or Rahman et al. (2013) come to the conclusions that NSE values greater 0.5 can be considered as good in hydrological modelling.

**(24)** 511-530: Can the inter-model difference also be related to their snow models? Do all of these models use the same snow model?

**Author's answer**: The inter-model difference can also be related to the used snow models. As explained in the paragraph, the studies cannot be compared one by one since they are conducted on different scales, in different (micro-)climatic regions and also with different models. However, these studies show that it is generally possible to obtain reasonable results with global data, if conditions are appropriate, e.g., data is available for downscaling or the investigation of larger scales. We tried to improve the respective passage in the manuscript.

**(25)** Please discuss the poor NSEs and R2s with the transferred and CHRP data.

**Author's answer**: As stated in our answer for comment 23, we consider the quality measures of the CHIRPS setup to be good. According to Moriasi et al. (2007) an NSE > 0.5 can be considered as satisfactory in hydrological modelling. This threshold is also used in alpine hydrological modelling (e.g. Rahman et al., 2013). We added the following in ll. 370-379 : "The NSE values of the CHIRPS setup range between 0.54 and 0.55; the $R²$ values range between 0.66 and 0.79 (Table 5). The reason for the good performance of the CHIRPS setup is that the forcing data is the same as in the reference setup except from precipitation. The overestimation of snow depth in some years, especially at the DWD site, is due to the higher precipitation values as described in Section 3.1. For the snow depth simulations based on the transferred data from the alpine DWD AWS at Mt. Wendelstein, we also obtained reasonable but weaker results, e.g., with an $R²$ of 0.59 and 0.66 at LWD and DWD, respectively but higher MAE values. The NSE values at the stations are 0.33 and 0.11, respectively. The statistical values are rather moderate, but still hint at a plausible representation of the temporal development compared to the results of the other global data. The low NSE values can be traced back to the years 2002, 2005and 2010 in which the peaks are strongly overestimated in the DWD_Wendelstein run at both stations. Both, the CHIRPS and DWD_Wendelstein show temperature and precipitation data which are within the total range of climatology."

**(26)** L550-551: This statement ignores the substantial differences in runoff between GTOPO30 and ALOS/SRTM.

**Author's answer**:  The statement you mention here is about the modelled snow cover development and not the runoff. Regarding the snow cover, the difference between the different DEM setups is indeed rather negligible.

**(27)** L571: "the choice of the DEM has far less impact" Incorrect. DEMs have large impacts on the simulated streamflow annual cycle

**Author's answer**: We rewrote the paragraph (see ll. 367-369): "The presented results show that the right choice of the meteorological forcing data still is the biggest challenge for snow hydrological modelling in

ungauged alpine headwater catchments. The choice of the DEM has far less impact on the snow cover modelling but can still result in considerable differences in snow cover as well as runoff generation."

**(28)** L600:604: Misleading. The 'borrowed' data performed poorly in terms of NSE and R2.

**Author's answer**: We rewrote the paragraph (see ll. 608-610): "Those two plausible setups are the CHIRPS setup in which precipitation only is substituted whilst taking all other meteorological data from the reference in situ data set, and the meteorological dataset which was 'borrowed' from another alpine AWS in the catchment's greater vicinity at Mt. Wendelstein."

Regarding the quality measures, please refer to our answer to comment 25, too.

**(29)** L510-512: There is a mystery. How can the forcing data sets of such a wide variation can produce such similar simulations? This needs answers from the authors.

**Author's answer**: We are not sure if you really mean L510-512 since we did not find a statement there that fits to your comment. We assume you perhaps could have meant L610-612, which is about differences due to DEMs. Our answer why the different DEM setups performed quite similar is that on the one hand, the ALOS and SRTM DEMs have the same spatial resolution, which is high enough that the HRU surface characteristics are relatively well represented as with the reference DEM. On the other hand, if results at the catchment scale are considered, in the coarser GTOPO30 setup, differences are leveled out. Moreover, as written in our answer to comment 21, the effect of altitude and enhanced radiation input is not as strong in winter, as in summer. Nonetheless, regarding the individual HRUs there are considerable differences as Figure 9 illustrates.

**References**

Grossi, G., Lendvai, A., Peretti, G., and Ranzi, R.: Snow Precipitation Measured by Gauges: Systematic Error Estimation and Data Series Correction in the Central Italian Alps, Water, 9, 461, doi:10.3390/w9070461, 2017.

Liston, G. E. and Elder, K.: A Meteorological Distribution System for High-Resolution Terrestrial Modeling (MicroMet), J. Hydrometeorol., 7, 217–234, doi:10.1175/JHM486.1, 2006.

Moriasi, D. N., Arnold, J. G., van Liew, M. W., Bingner, R. L., Harmel, R. D., and Veith, T. L.: Model Evaluation Guidelines for Systematic Quantification of Accuracy in Watershed Simulations, Transactions of the ASABE, 50, 885–900, doi:10.13031/2013.23153, 2007.

Rahman, K., Maringanti, C., Beniston, M., Widmer, F., Abbaspour, K., and Lehmann, A.: Streamflow Modeling in a Highly Managed Mountainous Glacier Watershed Using SWAT: The Upper Rhone River Watershed Case in Switzerland, Water Resour Manage, 27, 323–339, doi:10.1007/s11269-012-0188-9, 2013.

Weber, M., Feigl, M., Schulz, K., and Bernhardt, M.: On the Ability of LIDAR Snow Depth Measurements to Determine or Evaluate the HRU Discretization in a Land Surface Model, Hydrology, 7, 20, doi:10.3390/hydrology7020020, 2020.

WMO: Technical regulations: Basic documents no. 2, Volume I – General Meteorological Standards and Recommended Practices, 2010th ed., World Meteorological Organization, Geneva, 2011.

---

## Referee Report (RR1)

Review of "The evaluation of the potential of global data products for snow hydrological modelling in ungauged high alpine catchments"

Michael Weber et al.

This article seeks to quantify the magnitude of snow hydrological parameter uncertainty when modelling snowpack using global climate and topographic datasets relative to using local climate and high-resolution topographic measurements. The work generally demonstrates that use of global climate data sets is inappropriate for a basin of this small size, high elevation, and rugged topography. Use of nearby datasets or precipitation products in combination with local climate data may produce acceptable results. Use of mid-resolution global topographic products (30m) also yielded acceptable results and in this case, coarser topography (1 km) yielded results consistent with reference runs, though for the wrong reasons.

General comments: The manuscript is much improved from the initial review and easier to follow. Besides the specific comments to follow, I would recommend focusing on key findings in the results, perhaps generalizing by category, and then discussing the significance of the findings with respect to model choices. While I realize that a full discussion of consideration of basin size, roughness, elevation, etc relative to available datasets in beyond the scope of this paper, the community would benefit from an organized discussion of these factors. This is hinted at when discussing the results of other research but could be made much more explicit. For example, rather than start with "Study A" exhibited better results, consider starting with the theme "basin size and elevation matter" and then demonstrate this with case studies including results from this work. This kind of organized discussion would make this a much stronger and more interesting paper.

Specific comments:

Table 1. Are all measured parameters at DWD really available since 1900? Qsi and Qli were probably added more recently?

Figure 1. What is UFS? Adding locations of glaciers would help readers see where snow depth is measurement on and off glacial surfaces.

Figure 2. What is the large blank area? This seems significant as this appears to be where one of the snow depth measurements is taken.

Line 196. What does it mean to run the model in "gauged basin mode"? Does this mean that parameters are adjusted to match outflow? There is mention of a stream gauge, but it is not clear how or if these data are used. Furthermore, how well does the model work? Summarize findings of Weber et al 2020.

Line 205: Similarly, what does "ungauged basin mode" mean?

Section 2.4: Are the different DEM's used with the reference simulation climatological data? If so, what parameters are adjusted as a result. Section 3.2 states that the reference simulation was used and that it was explained in section 2.3. This however is not the case.

Line 324-326. There is no explanation of how reference climate data are adjusted using ALOS, SRTM, and GTOPO30 DEMs. I believe the referenced section (2.3) is incorrect. It should be section 2.4. Please clarify.

Figure 5. What is "(d) setup"? Is this the reference simulation and Lidar topography? Please clarify.

Discussion: Many of the findings with respect to climate products are not too surprising given their coarse resolution. This has been demonstrated several times in mountainous terrain. What would make this discussion much more interesting is if the authors examined factors that limited their use by topic such as basin size, basin homogeneity, basin elevation, area climatological variability, etc and then supported statements with their findings as well as those of other researchers.

Line 545. How will one know if transferring data from a catchment within 100km will provide the best results? Again, if this were addressed in a framework mentioned in the previous comment, it would be more helpful.

Line 559. Again using a framework to consider the use of alternate topographic products would be helpful. For example, use of GTOPO30 might be ok if the area under consideration is well above the current snowline, but would be problematic in basins where the much of the snow accumulation area lies close to the freezing line (where small errors in Ta result in the wrong precipitation phase).

---

## Referee Report (RR2)

Comments on the manuscript: "The evaluation of the potential of global data products for snow hydrological modeling in ungauged high alpine catchments" by Weber et al.

The authors have address much of my major concerns in the revised manuscript. As a consequence, I don't see any critical issues that require major revision of this manuscript. However, the revised version still needs further improvements, especially in writing and organization. I recommend this manuscript to be accepted with minor revision. Specific suggestions are presented in the below.

(1) Need thorough improvements in writing, especially to fix grammatical errors.

(2) Line 4 in Abstract: Specify the 10-year period (Sept 2000 – Aug 2010)

(3) Line 126: convert 'ha' into 'square kilometers'. I'm not sure if 'ha' is an MKS unit.

(4) Line 178: 'specifically' may be better than 'especially'.

(5) Line 308: 'data' may be better than 'curves'.

(6) Line 320: What is 'internal' variance? Please explain 'internal'.

(7) Lines 367-403: This is a huge paragraph that contains discussions on two separate subtopics, the sensitivity to the met data (367-395) and the sensitivity to the topo data (395-403). Dividing the paragraph into two paragraphs (for the met data and for the topo data) will it make easier to capture the key statements in this paragraph.

(8) Related to (7), Figure 6 may also be split into two like; Fig. 6a to present the met data sensitivity and Fig. 6b to present the topo data sensitivity.

(9) L149:151: Figure 9 may also be split into two like; Fig. 9a for the met data sensitivity and Fig. 9b for the topo data sensitivity.

(10) L495: Comparison of the topo data sensitivity to the met data sensitivity is largely irrelevant. May change 'all other setups' to 'all other DEM setups' to clarify that this statement refers to the topo data sensitivity experiment.

(11) Section 5. Conclusion is another huge paragraph with mixed subtopics. I suggest to split it into 4 paragraphs; Lines 593-605, 605-617, 617-631, and 631-641.

---

## Author Response (AR2)

**Response to Anonymous Referee #1**

The authors have done a commendable job responding to the reviewer comments. I think the addition of the ERA5 family reanalyses, as well as the more in-depth analysis of the various product resolutions and their impacts, helps understand the added-value (or not) of the global datasets. The text is also more nuanced regarding strengths and weaknesses of various products. Overall, I am satisfied with the quality of this new and improved version of the paper.

One point still bothers me somewhat, however. The authors acknowledge that the observed reference precipitation can be affected by undercatch, and a correction factor is applied. However, this method is pretty coarse and brings lots of uncertainty. Then, in the text around figure 3, the reference dataset is taken as the "truth" to which the other products are compared. I think it would be wise to restate that the results are relative and that it is not known (at this point in the paper) which ones are closer to the reality.

I think this can be easily resolved with a sentence or two in the text, which is why I recommend minor revision which can be handled at the editorial board level next.

Two additional sentences in ll. 283-284: "Although we consider the measured data to be the reference, it is not sure at this point of the paper that it performs best in modelling the snow cover. Therefore, the presented results are relative."

Good work by the authors and an overall very interesting paper.

**Authors' answer:** We once again thank the reviewer for his/her constructive feedback and very much appreciate the effort he/she took.

Regarding his/her comment: We agree, and added the sentences as suggested.

**Response to Anonymous Referee #2**

**General comments:**

The manuscript is much improved from the initial review and easier to follow. Besides the specific comments to follow, I would recommend focusing on key findings in the results, perhaps generalizing by category, and then discussing the significance of the findings with respect to model choices. While I realize that a full discussion of consideration of basin size, roughness, elevation, etc relative to available datasets in beyond the scope of this paper, the community would benefit from an organized discussion of these factors. This is hinted at when discussing the results of other research but could be made much more explicit. For example, rather than start with "Study A" exhibited better results, consider starting with the theme "basin size and elevation matter" and then demonstrate this with case studies including results from this work. This kind of organized discussion would make this a much stronger and more interesting paper.

I suggest that the authors place their findings into a framework of considerations others may find useful when selecting meteorological or topographic datasets. This would be one of the more valuable contributions of this paper.

**Authors' answer:** We once again thank the reviewer for his/her constructive feedback and very much appreciate the effort he/she took.

In our opinion, it is currently not possible to implement an encompassing profound framework in the discussion part including findings of other studies, as similar studies explicitly for high-alpine areas in various other regions are still lacking in literature. However, we definitely agree with the reviewer that such a framework would be highly valuable and should be established as soon as more studies on comparing various global products explicitly in different high-alpine regions come up. This goes also in line with our general remarks and suggestions for next steps in Section 4.3, where we suggest to set up such comparisons at further similar study sites, e.g. of the INARCH network. We added several points in the discussion in this regard - please consider our answers to your last three specific comments for more details.

**Specific comments:**

Table 1. Are all measured parameters at DWD really available since 1900? Qsi and Qli were probably added more recently?

**Authors' answer:** You are right, radiation data was recorded since 2009. We changed that in the Table.

Figure 1. What is UFS? Adding locations of glaciers would help readers see where snow depth is measurement on and off glacial surfaces.

**Authors' answer:** We added the explanation of UFS, which is the Environmental Research Station Schneefernerhaus to the figure caption. We also added the glacier surfaces as suggested to the map.

Figure 2. What is the large blank area? This seems significant as this appears to be where one of the snow depth measurements is taken.

**Authors' answer:** The white area is HRU10, as it is explained in the legend of this figure. HRU10 represents the glaciated areas on which DWD snow depth is measured. The added glaciated area in Figure 1 should help to clarify this as well now.

Line 196. What does it mean to run the model in "gauged basin mode"? Does this mean that parameters are adjusted to match outflow? There is mention of a stream gauge, but it is not clear how or if these data are used. Furthermore, how well does the model work? Summarize findings of Weber et al 2020.

**Authors' answer:** "Gauged basin mode" refers to the model run with in situ measured data. It has not been adjusted to match the outflow. In terms of IAHS, the term "gauged" does not only refer to a stream gauge but to any measurement device in a basin such as temperature sensors, snow gauges, and precipitation gauges etc. As written in section 3.4, we do not use measured runoff in our study due to data gaps and the general poor data quality. We added a sentence in ll. 199-200 to prove the achieved good model accuracy of Weber et al. 2020: "Measured snow depth could be modelled with an accuracy of NSE > 0.7 (Nash-Sutcliffe-Efficiency, (Nash and Sutcliffe, 1970))."

Line 205: Similarly, what does "ungauged basin mode" mean?

**Authors' answer:** Analogues to the previous point, in the "ungauged basin mode" no in situ measured data are available for modelling. Therefore, other forcing data have been used, as explained in this sentence. We refer to the first sentence in the introduction in ll. 29-30, where we explain the meaning of ungauged. For clarification we added the subordinate clause to l. 209: "in which we assume to have no measured model forcing data".

Section 2.4: Are the different DEM's used with the reference simulation climatological data? If so, what parameters are adjusted as a result. Section 3.2 states that the reference simulation was used and that it was explained in section 2.3. This however is not the case.

**Authors' answer:** All different DEMs are used with the reference meteorological data as explained in Section 3.2 (ll. 330-332). Also as explained in Section 3.2, as well as in Section 2.4, the adjusted parameters are altitude, slope and aspect, which in turn influence the meteorological data. Section 3.2 states that the reference meteorological data, which means the in situ measured data, are used and that these data are adjusted to the DEM specific HRU altitude, aspect and slope as explained in Section 2.3. In Section 2.3 it is described how the meteorological data is adjusted to the HRUs. We think this should be clear enough to the reader.

Line 324-326. There is no explanation of how reference climate data are adjusted using ALOS, SRTM, and GTOPO30 DEMs. I believe the referenced section (2.3) is incorrect. It should be section 2.4. Please clarify.

**Authors' answer:** Section 2.3 is the correct reference in this case. In this Section, we explain how the meteorological data is adjusted to the HRUs and thus the DEM specific values of altitude, aspect and slope angle. The same method is also applied for the non-reference meteorological and DEM input data which is presented in the following.

Please consider ll.203-206: "The corrected data were then transferred to the HRUs following the method of Liston and Elder (2006) as well. This method uses monthly variable lapse rates for temperature (Kunkel, 1989) and a monthly variable precipitation adjustment factor (Thornton et al., 1997) for the altitude-dependent adjustment of temperature and precipitation. Relative humidity was adjusted via dew point temperature which has a relatively linear dependence on elevation (Liston and Elder, 2006)."

The radiation data is treated by CRHM internally which is explained in Section 2.2.

Figure 5. What is "(d) setup"? Is this the reference simulation and Lidar topography? Please clarify.

**Authors' answer:** We changed the figure caption so this is clear now.

Discussion: Many of the findings with respect to climate products are not too surprising given their coarse resolution. This has been demonstrated several times in mountainous terrain. What would make this discussion much more interesting is if the authors examined factors that limited their use by topic such as basin size, basin homogeneity, basin elevation, area climatological variability, etc and then supported statements with their findings as well as those of other researchers.

**Authors' answer:** We agree that several studies applied such global products; however, we are not aware of other studies, which explicitly compare numerous global products specifically for high-alpine regions as we did. In the discussion, we tried to mention existing studies, which point somehow in this direction, but which might not be directly comparable as we discussed. The majority of these studies consider alpine regions including their forelands and not explicitly only high alpine sites, e.g. above the tree line and highly characterized by complex terrain.

In our opinion, it is currently not possible to compare and categorize the findings of our study with others in general for a sound framework for high alpine catchments. As this might have been unclear in the discussion, we added the following in ll. 515-518: "According to the knowledge of the authors, studies comparing different global meteorological input data explicitly for high-alpine regions are very sparse and not available in the extent of this study. Therefore, a framework-based comparison of different studies in this regard is not possible at current stage. In the following, however, we discuss further studies, which point to a certain extent in this direction."

Moreover, at the beginning we included that we focused exemplarily on the RCZ (see ll. 505-506 "In the following, we discuss the potential applicability of global products for snow hydrological modelling in high alpine regions as exemplarily demonstrated for the RCZ in this study.").

As already stated concerning the general comment of the reviewer, we see great need for, e.g., a follow-up study including several high-alpine catchment of similar category than the RCZ to be able to make such statements. This would of course be helpful, as it would provide a clear guidance to action for the use of model forcing data in ungauged basins. Therefore, we added in ll.596-601: "This would enable to create a comprehensive framework of how to use globally available data for model forcing in ungauged high alpine basins. Regarding globally available meteorological setups, a framework categorization could include statements on catchment size and homogeneity, climatological variability as well as its topographic characteristics."

Line 545. How will one know if transferring data from a catchment within 100km will provide the best results? Again, if this were addressed in a framework mentioned in the previous comment, it would be more helpful.

**Authors' answer:** We agree that 100 km might have been a too speculative number although it was the case for the transfer of the meteorological data of Mt. Wendelstein. We deleted this number, but kept the statement that this is true for stations in the closer vicinity in the same mountain range with similar climatology.

Line 559. Again using a framework to consider the use of alternate topographic products would be helpful. For example, use of GTOPO30 might be ok if the area under consideration is well above the current snowline, but would be problematic in basins where the much of the snow accumulation area lies close to the freezing line (where small errors in Ta result in the wrong precipitation phase).

**Authors' answer:** We agree with the reviewer that also a framework regarding global DEM products for high-alpine catchments would be of great interest. However, similar to the global meteorological products, we think that for a comprehensive and profound framework regarding the topographic products it is not sufficient to rely only on the results obtained only in RCZ and as findings in other sites are still missing or very sparse. We included in Section 4.2, ll. 563-570 some literature findings for other than explicitly high alpine regions on different DEM resolutions in hydrological modelling as well as one study on glacier melt. The added references are the following:

- Hopkinson, C., Chasmer, L., Munro, S., and Demuth, M. N.: The influence of DEM resolution on simulated solar radiation-induced glacier melt, 24, 775–788, https://doi.org/10.1002/hyp.7531, 2010.
- Nagaveni, C., Kumar, K. P., and Ravibabu, M. V.: Evaluation of TanDEMx and SRTM DEM on watershed simulated runoff estimation, Journal of Earth System Science, 128, 73, https://doi.org/10.1007/s12040-018-1035-z, 2019.
- Sørensen, R. and Seibert, J.: Effects of DEM resolution on the calculation of topographical indices: TWI and its components, Journal of Hydrology, 347, 79–89, https://doi.org/10.1016/j.jhydrol.2007.09.001, 2007.
- Vaze, J., Teng, J., and Spencer, G.: Impact of DEM accuracy and resolution on topographic indices, Environmental Modelling & Software, 25, 1086–1098, https://doi.org/10.1016/j.envsoft.2010.03.014, 2010.

For a proper comparison in high alpine regions, it would require, similar to the meteorological products, a much broader choice of high alpine test sites with different topographic characteristics. Such studies should especially consider the degree of topographic complexity. Therefore, we added the following suggestion regarding the DEM products in ll. 599-601: "Regarding the usage of global DEM products an investigation especially on the degree of topographic complexity and especially the steepness of terrain would be most interesting."

**Response to Anonymous Referee #3**

We once again thank the reviewer for his/her constructive feedback and very much appreciate the effort he/she took. In the following, we give an individual answer to each comment.

1) Need thorough improvements in writing, especially to fix grammatical errors.
**Authors' answer:** We gave the manuscript to a native speaker for proofreading of this revised version and mentioned explicitly to have also an eye on grammar issues.

2) Line 4 in Abstract: Specify the 10-year period (Sept 2000 – Aug 2010)
**Authors' answer:** We added the time period (l.4).

3) Line 126: convert 'ha' into 'square kilometers'. I'm not sure if 'ha' is an MKS unit.
**Authors' answer:** We changed it to km² (l.126).

4) Line 178: 'specifically' may be better than 'especially'.
**Authors' answer:** We changed it to 'specifically' (l.179).

5) Line 308: 'data' may be better than 'curves'.
**Authors' answer:** We changed it to 'data' (l.306).

6) Line 320: What is 'internal' variance? Please explain 'internal'.
**Authors' answer:** We rewrote the sentence (ll.326-327): "The variance of slope and aspect angles in the coarse GTOPO30 DEM setup is very low."

7) Lines 367-403: This is a huge paragraph that contains discussions on two separate subtopics, the sensitivity to the met data (367-395) and the sensitivity to the topo data (395-403). Dividing the paragraph into two paragraphs (for the met data and for the topo data) will it make easier to capture the key statements in this paragraph.
**Authors' answer:** We agree and we divided it into two paragraphs according to the suggestion (ll.373-410).

8) Related to (7), Figure 6 may also be split into two like; Fig. 6a to present the met data sensitivity and Fig. 6b to present the topo data sensitivity.
**Authors' answer:** We split Figure 6 into a) and b) regarding meteorological and DEM data, respectively, as the lines were obviously too close together. We also added a) and b) in this regard in the Figure capture.

9) L149:151: Figure 9 may also be split into two like; Fig. 9a for the met data sensitivity and Fig. 9b for the topo data sensitivity.
**Authors' answer:** For Figure 9, however, we do not see the necessity to split it, since the signature of the bar plots clearly separates the met and topo data.

10 ) L495: Comparison of the topo data sensitivity to the met data sensitivity is largely irrelevant. May change 'all other setups' to 'all other DEM setups' to clarify that this statement refers to the topo data sensitivity experiment.
**Authors' answer:** We changed it to 'all other DEM setups' (l.502).

11) Section 5. Conclusion is another huge paragraph with mixed subtopics. I suggest to split it into 4 paragraphs; Lines 593-605, 605-617, 617-631, and 631-641.
**Authors' answer:** We agree that the readability of the conclusion improves and split this paragraph according to the suggestions of the reviewer (ll.620-669).